# MARKOVIAN COMPRESSION: LOOKING TO THE PAST HELPS ACCELERATE THE FUTURE

## ABSTRACT

This paper deals with distributed optimization problems that use compressed communication to achieve efficient performance and mitigate the communication bottleneck. We propose a family of compression schemes in which operators transform vectors fed to their input according to a Markov chain, i.e., the stochasticity of the compressors depends on previous iterations. Intuitively, this should accelerate the convergence of optimization methods, as considering previous iterations seems more natural and robust. The compressors are implemented in the vanilla Quantized Stochastic Gradient Descent (QSGD) algorithm. To further improve efficiency and convergence rate, we apply the momentum acceleration method. We prove convergence results for our algorithms with Markovian compressors and show theoretically that the accelerated method converges faster than the basic version. The analysis covers non-convex, Polyak-Lojasiewicz (PL), and strongly convex cases. Experiments are conducted to demonstrate the applicability of the results to distributed data-parallel optimization problems. Practical results demonstrate the superiority of methods utilizing our compressors design over several existing optimization algorithms.

## 1 INTRODUCTION

The optimization problem is currently a key issue in many practical applications, such as optimization in neural network training, resource allocation in computational systems, and parameter tuning in algorithmic trading strategies.

In addition, a variety of algorithms for optimization on a single device, such as SGD Robbins & Monro (1951), Adam Kingma & Ba (2014), Lion Yazdani & Jolai (2016), have emerged and been subjected to theoretical analysis. However, in the contemporary landscape of deep learning, there is an increasing trend towards adopting intricate and expansive models that pose significant training challenges. Prominent among these challenges are advanced deep learning frameworks for image analysis, sophisticated natural language processing structures akin to transformers Vaswani et al. (2017), and complex reinforcement learning methodologies designed for autonomous system operations Kiran et al. (2021). As a result, the training of such models has become impractical for execution on a single device due to their requirement for extensive data sets for training, which are unfeasible to store on a single device. Consequently, optimization algorithms have been specifically developed for distributed training Verbraeken et al. (2020); Chen et al. (2021). These methods utilize a large number of devices, with each one processing distinct data subsets and participating in an effective data exchange mechanism, thereby aiding in the training of these computationally intensive models. Thus, the problem of classical optimization evolves into a distributed optimization form:

$$\min_{x \in \mathbb{R}^d} \left\{ f(x) := \frac{1}{n} \sum_{i=1}^{n} f_i(x) \right\}, \tag{1}$$

where $f_i$ is a function, located on a device $i$. This formulation encompasses not only distributed learning, where data is dispersed across multiple devices to expedite training and facilitate the storage of large amounts of data, but also extends to federated learning Konečný et al. (2016); Li et al. (2020); Kairouz et al. (2021), where data distribution is motivated by the architecture of the system itself, allowing for decentralized model training while maintaining data privacy and integrity across diverse devices.

A downside of this approach manifests as the complexity associated with the transmission of large-scale data, a phenomenon often referred to as the "communication bottleneck" Gupta et al. (2021).

This bottleneck can significantly impede the efficiency of the system, particularly in scenarios involving extensive data exchange across distributed networks. The challenge intensifies in environments where the bandwidth is limited, requiring solutions to mitigate the impact of data transmission delays and ensure seamless data flow.

The primary solution at present is the compression of transmitted information Bekkerman et al. (2011); Chilimbi et al. (2014); Alistarh et al. (2017), wherein not a whole package is sent, but rather a selected subset. This method involves strategically selecting and compressing the most informative segments of data for transmission. By doing this way, it significantly reduces the volume of data that needs to be communicated across the network, thereby alleviating the communication bottleneck.

In recent times, a number of methods employing compression have been conceived and scrutinized Mishchenko et al. (2019); Gorbunov et al. (2021a); Richtárik et al. (2021). However, a lot of studies have utilized unbiased compression operators due to their simplicity and amenability to theoretical analysis. Such compression techniques, including methods as random sparsification and value rounding Nesterov (2012a); Alistarh et al. (2017); Horvath et al. (2022); Beznosikov et al. (2023a), fail to consider the integration of information conveyed in prior iterations. We hence highlight a potential research gap regarding the usage of previously transmitted data in compression operators and optimization algorithms.

This omission raises the following research questions that we address in the paper:

- *Is it possible to design compression operators that take into account information about what and how we forwarded in previous iterations?*
- *What methods can we integrate this kind of compression operators into? How does it affect the convergence rate of the methods, both in theory and in practice?*
- *Can the methods be made even more efficient, e.g., by using additional momentum acceleration techniques?*

In our paper, we focus on compression-based methods that take into account information collected across multiple preceding iterations, employing what are termed as Markovian compression operators. To the best of our knowledge, this approach emerges as novel and unexplored in the existing literature.

## 1.1 OUR CONTRIBUTIONS

**New type of compression operators.** We introduce a novel type of compressors that utilizes stochasticity transmitted over several previous iterations. We refer to this type of compressors as Markovian, because the states of these compressors can be viewed as a Markov chain. We examine two invented examples of such compressors: $\texttt{BanLast}(K, m)$ (Definition 5) and $\texttt{KAWASAKI}(K, b, \pi_\Delta, m)$ (Definition 6). The first new compressor operates on a more intuitive basis: it works as random sparsification, but prohibits the transmission of coordinates that were sent in the previous $K$ iterations. The latter functions in terms of probabilities: it reduces the likelihood of transmitting coordinates that appeared in previous iterations. The $\texttt{KAWASAKI}(K, b, \pi_\Delta, m)$ compressor is more flexible and, in fact, modify the idea $\texttt{BanLast}(K, m)$, but it introduces two hyperparameters that will be discussed later in Section 2.1.

**New algorithms.** The compression operators described above give rise to new methods that utilize them. In this context, our paper outlines a general framework based on Alistarh et al. (2017) for distributed gradient descent algorithms that employ Markovian compression operators ($\texttt{MQSGD}$, see Algorithm 1). Subsequently, to make this basic algorithm faster we apply the multiple momentum technique Nesterov (2012a) and obtain the accelerated method $\texttt{AMQSGD}$. The formulation of such an algorithm is detailed in Algorithm 2. The basic and accelerated methods are explored both theoretically and experimentally throughout the paper. Furthermore, experiments utilizing Markovian operators in the $\texttt{DIANA}$ Mishchenko et al. (2019) and SGD with momentum algorithms are conducted in Section 3.

**Strongly convex and non-convex cases.** Motivated by various applications primarily from machine learning, we provide the theoretical analysis in the strongly convex (Theorem 3) and non-convex / PL-condition (Theorem 2) cases of the target function $f$. Notably, we provide proper analysis for both setups with specific cases, which is rarely present in the field.

**Numerical experiments.** We conduct experiments with Markovian compressors in a data-parallel setup for several optimization problems and datasets. In particular, we analyze the proposed $\texttt{MQSGD}$ and $\texttt{AMQSGD}$, as well as the $\texttt{DIANA}$ and SGD optimizers for distributed optimization. In all setups, we observe an acceleration of convergence for methods employing the $\texttt{BanLast}$ and $\texttt{KAWASAKI}$ compressors compared to the baseline random sparsification.

## 1.2 RELATED WORK

**Compressed communications.** The use of compressed communications is a fairly well-known idea in distributed learning Seide et al. (2014). As soon as the main property of compressed messages is that they are much easier to transfer, it can be reached in different ways, such as by quantizing the entries of the input vector Alistarh et al. (2017); Mayekar & Tyagi (2019); Gandikota et al. (2020); Horvath et al. (2022), or by sparsifying it Richtárik & Takáč (2016); Alistarh et al. (2018), or even by combining these ideas Albasyoni et al. (2020); Beznosikov et al. (2023a). However, all of the compression operators could be roughly Condat et al. (2023) separated into two large groups: *unbiased* and *biased*.

The first group is much easier to analyze and is therefore more broadly represented in the literature. The basic method with unbiased compression was presented in Alistarh et al. (2017). Later this algorithms were modified using variance reduction technique with compression of gradient differences Mishchenko et al. (2019); Horváth et al. (2019); Gorbunov et al. (2021a) in order to improve the theoretical convergence guarantees. One can also note the works Gorbunov et al. (2019) and Khaled et al. (2020), where the authors developed a general theory for SGD-type methods with unbiased compression.

On the other hand, our understanding of distributed optimization with biased compressors is more complicated. In particular, biased compression implies the use of error compensation techniques Stich et al. (2018). Distributed SGD with biased compression and linear rate of convergence in a multi-node setting was first introduced in Beznosikov et al. (2023a). In the meantime, other error compensation techniques are being actively developed, Lin et al. (2022); Richtárik et al. (2021). The last approach called `EF21` was later studied in Fatkhullin et al. (2021), Gruntkowska et al. (2023).

**Markovian stochasticity.** Another recent trend in the literature is to design algorithms that use Markovian stochastic processes instead of $i.i.d.$ random variables in various ways. For instance, Duchi et al. (2012) introduced a version of the Mirror Descent algorithm that yields optimal convergence rates for non-smooth and convex problems. Later, Doan et al. (2020a); Dorfman & Levy (2023); Beznosikov et al. (2023b) studied first-order methods in the Markovian noise setting. Alternatively, token algorithms Hendrikx (2022); Ayache et al. (2022) are also a popular area of research in Markovian stochasticity. In particular, Even (2023) obtained optimal rates of convergence, and Sun et al. (2022); Mao et al. (2019); Doan et al. (2020b) looked at the token algorithm from the angle of the Lagrangian duality and from variants of the ADMM method. At the same time, there exist particular results, e.g., Bresler et al. (2020), which provide a lower bound for the particular finite sum problems in the Markovian setting.

Despite all of the above, to the best of our knowledge, there are currently no works that combine compressed data communications and Markovian stochasticity of the compressors.

## 1.3 TECHNICAL PRELIMINARIES

**Notations.** We use $\langle x, y \rangle := \sum_{i=1}^{d} x_i y_i$ to denote standard inner product of vectors $x, y \in \mathbb{R}^d$ and $(x \odot y)_i = x_i y_i$ to denote Hadamard product of vectors $x, y \in \mathbb{R}^d$. We introduce $l_2$-norm of vector $x \in \mathbb{R}^d$ as $\|x\| := \sqrt{\langle x, x \rangle}$. We define $x^* \in \mathbb{R}^d$ as a point, where we reach the minimum in the problem (1). We also denote $f^* > -\infty$ as a global (potentially not unique) minimum of $f$. We use a standard notation for $(d-1)$-dimensional simplex $\Delta_d := \left\{ p \in \mathbb{R}^d \mid p_j \geq 0 \text{ and } \sum_{j=1}^{d} p_j = 1 \right\}$ and for a set of natural numbers $\overline{1, n} := \{1, 2, \ldots, n\}$. We denote $C_m^k$ as the binomial coefficient $\binom{m}{k}$.

Throughout the paper, we assume that the objective functions $f_i$ and the function $f$ from (1) satisfy the following assumptions.

**Assumption 1** ($L_i$-smooth). *Every function $f_i$ is $L_i$-smooth on $\mathbb{R}^d$ with $L_i > 0$, i.e. it is differentiable and there exists a constant $L_i > 0$ such that for all $x, y \in \mathbb{R}^d$ it holds that $\|\nabla f_i(x) - \nabla f_i(y)\|^2 \leq L_i^2 \|x - y\|^2$. We define $L^2 := \frac{1}{n} \sum_{i=1}^{n} L_i^2$.*

**Assumption 2** ($\mu$-strongly convex). *The function $f$ is $\mu$-strongly convex on $\mathbb{R}^d$, i.e., it is differentiable and there is a constant $\mu > 0$ such that for all $x, y \in \mathbb{R}^d$ it holds that $(\mu/2) \|x - y\|^2 \leq f(x) - f(y) - \langle \nabla f(y), x - y \rangle$.*

**Assumption 3** (PL-condition). *The function $f$ satisfies the PL-condition, i.e., it is differentiable and there is a constant $\mu > 0$ such that for all $x \in \mathbb{R}^d$ it holds that $\|\nabla f(x)\|^2 \geq 2\mu (f(x) - f^*)$.*

**Assumption 4** (Data similarity). *The functions $f_i$ are similar on $\mathbb{R}^d$, i.e., there are constants $\delta, \sigma \geq 0$, such that the following inequality holds for all $x \in \mathbb{R}^d$: $\|\nabla f_i(x) - \nabla f(x)\|^2 \leq \delta^2 \|\nabla f(x)\|^2 + \sigma^2$.*

The equation above implies that the data stored at each worker does not differ significantly. This Assumption is quite standard in the literature Shamir et al. (2014); Arjevani & Shamir (2015); Khaled et al. (2020); Woodworth et al. (2020); Gorbunov et al. (2021b); Beznosikov et al. (2022; 2023b).

Now we introduce important definitions related to the theory of Markov processes.

**Definition 1** (Markov chain). *Markov chain with a finite state space $\{\nu_n\}_{n=0}^N$ is a stochastic process $\{X_t\}_{t \geq 0}$, that satisfies Markov property, i.e. $\mathbb{P}\{X_t = \nu_t \mid X_{t-1} = \nu_{t-1}, X_{t-2} = \nu_{t-2}, ..., X_0 = \nu_0\} = \mathbb{P}\{X_t = \nu_t \mid X_{t-1} = \nu_{t-1}\}$.*

**Definition 2** (Ergodicity of Markov chain). *Markov chain $\{X_t\}_{t \geq 0}$ with a finite state space $\{\nu_n\}_{n=0}^N$ is referred to be ergodic if for any $n \in \overline{1, N}$ there exists $\lim_{t \to \infty} \mathbb{P}\{X_t = \nu_n \mid X_0 = \nu_0\} = p_n$, where $0 \leq p_n \leq 1$ does not depend on the $\nu_0$. If Markov chain is ergodic, then $\{p_n\}_{n=0}^N \in \Delta_N$ and there exist $0 < \rho < 1, C > 0$, such that $|\mathbb{P}\{X_t = \nu_n \mid X_0 = \nu_0\} - p_n| \leq C\rho^t$.*

**Definition 3** (Mixing time of the discrete Markov chain). *We say that $\tau_{mix}(\varepsilon)$ is the mixing time of the ergodic Markov chain $\{X_t\}_{t \geq 0}$ with stationary distribution $\{p_n\}_{n=0}^N$, if $\forall \varepsilon > 0, \forall t \geq \tau_{mix}(\varepsilon) \hookrightarrow \max_{n \in \overline{0,N}} \{|\mathbb{P}\{X_t = \nu_n \mid X_0 = \nu_0\} - p_n|\} \leq \varepsilon \cdot p_{\min}$, where $p_{\min} := \min_{n \in \overline{0,N}} \{p_n\}$. From the Definition 2, it follows that $\tau_{mix}(\varepsilon) \geq \frac{\log(C/p_{\min}\varepsilon)}{\log(1/\rho)}$.*

These definitions are extremely important for further analysis of the Markovian compressors, which are presented in the next section.

## 2 MAIN RESULTS

### 2.1 MARKOVIAN COMPRESSORS

In this section, we introduce Markovian compressors that take into account the information transmitted in previous $K$ operations. It is assumed that these compressors function within an iterative algorithm aimed at minimizing the problem (1), wherein a distinct discrete variable, denoted as the step $t$, is involved. Consequently, due to the dependence of the compressors on previous states, they exhibit a reliance on the step $t$. Let us narrow down the class of compressors to be discussed in this paper.

**Definition 4** (Random sparsification). *$Q_t(x)$ is a random sparsification compressor, if it operates on the vector $x \in \mathbb{R}^d$ as $Q_t(x) = \frac{d}{m} x \odot \mathbb{1}(\nu_t)$, where $\nu_t$ is a set of $m$ coordinates : $\nu_t \subseteq \overline{1, d}$.*

The classical Rand$m$ operator fits Definition 4, in particular, for this compressor subsets $\nu_t$ are generated uniformly at each step $t$, therefore it is unbiased, i.e., $\mathbb{E}_t[Q_t(x)] = x$ for all $t$. In this paper, we do not generate $\nu_t$ independently, but according to some Markov chain, i.e., compressors start to take into account past iterations. We formulate this idea as an assumption.

**Assumption 5** (Asymptotic unbiasedness of Markovian compressors). *We assume that operator $Q_t$ is a random sparsification compressor (Definition 4) and $\{\nu_t\}_{t \geq 0}$ are realizations of some ergodic Markov chain with uniform stationary distribution.*

Assumption 5 implies that in the limit as $t \to \infty$, the compressor $Q_t$ is unbiased, i.e., $\mathbb{E}[Q_t(x)] \to x$ as $t \to \infty$, because the stationary distribution of the Markov chain is uniform. We are now ready to introduce two compressors that adhere to Assumption 5. The first compressor is called BanLast$(K, m)$, it prohibits sending coordinates that have been sent at least once in the last $K$ iterations.

**Definition 5** (BanLast$(K, m)$ compressor). *Let $Q_t(x)$ be a random sparsification compressor (Definition 4). The $j \in \nu_t$ are chosen according to the distribution $p^t \in \Delta_d$ and $p^t$ is given by the formula:*
$$p_j^t = \begin{cases} 0, & \text{if } j \in \bigcup_{s=t-K}^{t-1} \nu_s, \\ \frac{1}{d - Km}, & \text{otherwise.} \end{cases}$$

The BanLast$(K, m)$ compressor exhibits a limitation in its utility due to an application restriction: $d \geq (K + 1)m$, since we need at least $m$ coordinates to have a non-zero probability at each step $t$. In order to avoid these limitations, we introduce a more flexible Markovian compressor KAWASAKI$(K, b, \pi_\Delta, m)$.

**Definition 6** (KAWASAKI($K, b, \pi_\Delta, m$) compressor). *Let $Q_t(x)$ be a random sparsification compressor (Definition 4). The $j \in \nu_t$ are chosen according to the distribution $p^t \in \Delta_d$, which is given by the formula:*

$$\widetilde{p}_j^{\,t} = \frac{1/d}{b^{\text{\# of choices } j \text{ for the last } K \text{ iterations}}}, \quad j \in \overline{1, d}; \qquad p^t = \pi_\Delta\left(\widetilde{p}^{\,t}\right),$$

*where $b > 1$ is a forgetting rate and $\pi_\Delta : \mathbb{R}^d \to \Delta_d$ is an activation function.*

The KAWASAKI($K, b, \pi_\Delta, m$) compressor is now applicable for arbitrary values of $d \geq m$, and $K$. However, it introduces two additional hyperparameters in comparison with BanLast($K, m$), namely $b$ and $\pi_\Delta$. The parameter $b$ is responsible for the how strongly we penalize a coordinate if it was selected in previous iterations, the larger $b$ is, the less likely we are to select a coordinate in step $t$ if it was selected in steps $t - K$ to $t - 1$. The function $\pi_\Delta$ is required in order to obtain the probability vector $p^t$ from the vector $\widetilde{p}^{\,t}$, the necessary conditions for this function will be introduced later. The following examples illustrate potential selections for $\pi_\Delta$:

$$\left(\pi_\Delta\left(\widetilde{p}\right)\right)_j = |\widetilde{p}_j| / \|\widetilde{p}\|_1, \quad \pi_\Delta\left(\widetilde{p}\right) = \text{Softmax}\left(\widetilde{p}\right), \quad \pi_\Delta\left(\widetilde{p}\right) = \arg\min_{p \in \Delta_d}\{\|\widetilde{p} - p\|^2\}.$$

We now provide an example where using the Markovian compressor BanLast($K, m$) (Definition 5) speeds up the optimization process by a factor of three compared to the unbiased compressor Rand$m$.

**Example 1.** *Consider the QSGD algorithm (Algorithm 1), which solves the problem* (1) *in the case $n = 1$, of the form $x^{t+1} = x^t - \gamma Q(\nabla f(x^t))$. Assume that at some step $t$ we observe gradient of the form $(1, 0, ..., 0)^T \in \mathbb{R}^d$. In the QSGD algorithm, we compress the gradient at each step, therefore, we do not always send the first coordinate to the server, i.e. we do not move from the point $x^t$.*

*In the case of $m = 0.1 \cdot d$, i.e. we send 10% of all coordinates at each step, if we use the BanLast($K, m$) compressor, then the mathematical expectation of the number of steps to leave the point $x^t$ is approximately $3.4$ in the case of $K = 7$. For Rand10% this number is equal to $10$, i.e. we speed up the optimization process by a factor of three. For arbitrary values of $d$ and $m$, the formula for calculating the number of steps to leave the point $x^t$ is provided in Appendix B.*

Moreover, in Appendix B, we obtain more general results for an arbitrary value of $\alpha \in (0; 1]$ with $d = \alpha \cdot m$. In particular, we find the exact expression for the dependence of the number of steps to leave the point $x^t$. For each fixed $\alpha$ we can find the optimal value of $K^*(\alpha)$. It turns out that empirically this dependence is close to a linear one of the form $K^*(\alpha) \approx 0.73 \cdot \alpha$. Such a rule can be used as an automatic way of choosing $K$.

We now present a theorem demonstrating that our Markovian compressors from Definitions 5 and 6 satisfy the conditions outlined in Assumption 5.

**Theorem 1** (Asymptotic unbiasedness of BanLast($K, m$) and KAWASAKI($K, b, \pi_\Delta, m$)). *Compressors from Definitions 5 and 6 can be described using Markov chains with states $\{\nu_1, \nu_2, ..., \nu_K\}_{\nu_1, ..., \nu_K \in M}$, where $M$ is the set of all subsets of $\overline{1, d}$ of size $m$. Moreover,*

- *BanLast($K, m$) (Definition 5) is ergodic with a uniform stationary distribution, if $d > (K+1)m$.*

- *If $d > (2K+1)m$, then for BanLast($K, m$) we get*

$$\rho = \sqrt{1 - \left(\frac{C_{d-2Km}^m}{(C_{d-Km}^m)^2}\right)^K} \quad and \quad C = \left(1 - \left(\frac{C_{d-2Km}^m}{(C_{d-Km}^m)^2}\right)^K\right)^{-1}.$$

- *If for all permutations $\phi$ of the set $\overline{1, d}$ it holds that $\pi_\Delta\left(\phi\left(\widetilde{p}\right)\right) = \phi\left(\pi_\Delta\left(\widetilde{p}\right)\right)$, then KAWASAKI($K, b, \pi_\Delta, m$) (Definition 6) is ergodic with a uniform stationary distribution.*

- *If $\left(\pi_\Delta\left(\widetilde{p}\right)\right)_j = |\widetilde{p}_j| / \|\widetilde{p}\|_1$, then*

$$\rho = 1 - \left[db^K - m(b^K - 1)\right]^{-mK} \quad and \quad C = \left(1 - \left[db^K - m(b^K - 1)\right]^{-mK}\right)^{-1}. \tag{2}$$

The proof of Theorem 1 is provided in Appendix C. The outcomes of Theorem 1 hold significant importance for the subsequent investigation of algorithms aimed at solving problem (1) employing Markovian compressors. Note that the examples of activation functions $\pi_\Delta$ provided above satisfy the conditions of Theorem 1.

## 2.2 Distributed gradient descent with Markovian compressors

In this section, we propose a new algorithm `Markovian QSGD` (Algorithm 1). This algorithm is similar to the vanilla `QSGD` Alistarh et al. (2017), but in line 7 of Algorithm 1 we use Markovian compressor $Q_t^i$, that we introduced in Section 2.1, i.e., $Q_t^i$ can be either `BanLast(K, m)` (Definition 5) or `KAWASAKI(K, b, \pi_\Delta, m)` (Definition 6).

**Theorem 2** (Convergence of `MQSGD` (Algorithm 1)). *Consider Assumptions 1, 4 and 5. Let the problem* (1) *be solved by Algorithm 1.*

- *For any $\varepsilon, \gamma > 0$, $T > \tau > \tau_{mix}(\varepsilon)$ satisfying conditions, described in Appendix E.1, it holds that*

$$\mathbb{E}\left[\left\|\nabla f(\widehat{x}^T)\right\|^2\right] = \mathcal{O}\left(\frac{F_\tau}{\gamma T} + \frac{\gamma L \tau d^2}{m^2}\sigma^2\right),$$

*where $\widehat{x}^T$ is chosen uniformly from $\{x^t\}_{t=0}^T$.*

- *If $f$ additionally verifies the PL-condition (Assumption 3), then for any $\varepsilon > 0$, $\gamma > 0$, $\tau > \tau_{mix}(\varepsilon)$ and $T > \tau$ satisfying conditions, described in Appendix E.1, it holds that*

$$F_T = \mathcal{O}\left(\left(1 - \frac{\mu\gamma}{12}\right)^{T-\tau} F_\tau + \frac{\gamma d^2 L \tau}{\mu m^2}\sigma^2\right).$$

*Here we use* the *notations $F_t := \mathbb{E}\left[f(x^t) - f(x^*)\right]$* and $F_\tau := \mathbb{E}\left[f(x^\tau) - f(x^*)\right]$.

The proof of Theorem 2 is provided in Appendix E.3, E.4. If Assumption 4 does not hold we observe different results, which are provided in the Appendix F.

Usually in convergence evaluations of various methods, expressions with the term of $F_0$, i.e., something that depends on the initial choice, arise as constants, but in Theorem 2, a term of the form $F_\tau$ appears. This can be explained by the fact that at iterations from $t = 0 \to \tau$ the Markov chain has not yet been stabilized, and the initial state can be taken as $t = \tau$.

**Sketch proof of Theorem 2**. Let us write out a descent lemma of the form

---

**Algorithm 1** `Markovian QSGD (MQSGD)`

1: **Input:** starting point $x^0 \in \mathbb{R}^d$,
2: step size $\gamma > 0$,
3: number of iterations $T$
4: **for** $t = 0$ **to** $T$ **do**
5:     Broadcast $x^t$ to all workers
6:     **for** $i = 1$ **to** $n$ in parallel **do**
7:         Set $g_i^t = Q_t^i(\nabla f_i(x^t))$
8:         Send $g_i^t$ to the server
9:     **end for**
10:    Aggregate $g^t = \frac{1}{n}\sum\limits_{i=1}^n g_i^t$
11:    Update $x^{t+1} = x^t - \gamma g^t$
12: **end for**

---

$$\mathbb{E}\left[\left\|x^{t+1} - x^*\right\|^2\right] = \mathbb{E}\left[\left\|x^t - x^*\right\|^2\right] - 2\mathbb{E}\left[\gamma\left\langle\nabla f(x^t), x^t - x^*\right\rangle\right]$$

$$\underbrace{-\frac{2\gamma}{n}\sum_{i=1}^n\mathbb{E}\left[\left\langle Q_t^i(\nabla f(x^t)) - \nabla f_i(x^t), x^t - x^*\right\rangle\right]}_{①} + \gamma^2\mathbb{E}\left[\left\|\frac{1}{n}\sum_{i=1}^n Q_t^i(\nabla f_i(x^t))\right\|^2\right]. \quad (3)$$

The expression ① in (3) is zero if $Q_t^i$ are unbiased and independent from iteration $t$, because $\mathbb{E}\left[\left\langle Q_t^i(\nabla f(x^t)) - \nabla f_i(x^t), x^t - x^*\right\rangle\right] = \mathbb{E}\left[\left\langle\mathbb{E}_t\left[Q_t^i(\nabla f(x^t)) - \nabla f_i(x^t)\right], x^t - x^*\right\rangle\right] = 0$, where $\mathbb{E}_t[\cdot]$ is the conditional expectation at a step $t$. Therefore, the theory for such compressors is highly developed. In our case, $Q_t^i(x^s)$ are unbiased only if $t - s \to \infty$, which follows from asymptotic unbiasedness of our Markovian compressors obtained from Assumption 5. However, we can use some coarsening rather than unbiasedness when $t - s = \tau$, where $\tau > \tau_{\text{mix}}(\varepsilon)$, using the technique of "stepping back" as follows:

$$\mathbb{E}\left[\left\langle Q_t^i\left(a^{t-\tau}\right) - a^{t-\tau}, b^{t-\tau}\right\rangle\right] \leq \frac{\varepsilon d}{m}\mathbb{E}\left[\left\|a^{t-\tau}\right\|\left\|b^{t-\tau}\right\|\right]. \quad (4)$$

Importantly, we must apply the compressor $Q_t$ at step $t$ to the vector $a^{t-\tau}$ at step $t - \tau$, since if we apply it to the vector $a^t$ at step $t$, we will not be able to uncover the conditional expectation, since we will have randomness in $a^t$ (see details in Appendix D). As can be seen from (3) we need to apply the last inequality with $a^{t-\tau} = \nabla f_i(x^{t-\tau})$ and $b^{t-\tau} = x^{t-\tau} - x^*$, but in (3) we only obtain expression with variables at step $t$, therefore, it has to be handled in some way. In order to resolve this issue we use a straightforward algebra:

$$\mathbb{E}\left[\langle Q_t^i\left(\nabla f_i(x^t)\right) - \nabla f_i(x^t), x^t - x^*\rangle\right] = \mathbb{E}\left[\langle Q_t^i\left(\nabla f_i(x^{t-\tau})\right) - \nabla f_i(x^{t-\tau}), x^{t-\tau} - x^*\rangle\right]$$

$$- \mathbb{E}\left[\langle Q_t^i\left(\nabla f_i(x^t) - \nabla f_i(x^{t-\tau})\right) - \nabla f_i(x^t) + \nabla f_i(x^{t-\tau}), x^t - x^{t-\tau}\rangle\right]$$

$$+ \mathbb{E}\left[\langle Q_t^i\left(\nabla f_i(x^t) - \nabla f_i(x^{t-\tau})\right) - \nabla f_i(x^t) + \nabla f_i(x^{t-\tau}), x^t - x^*\rangle\right] \tag{5}$$

$$+ \mathbb{E}\left[\langle Q_t^i\left(\nabla f_i(x^t)\right) - \nabla f_i(x^t), x^t - x^{t-\tau}\rangle\right].$$

The first term in the last inequality (5) is solved with the $\varepsilon$-inequality (4), other scalar products are solved using the Fenchel-Young inequality. Terms with $\mathbb{E}\left\|x^t - x^{t-\tau}\right\|^2$ are evaluated using line 9 of Algorithm 1: $x^t - x^{t-\tau} = -\gamma\sum_{s=t-\tau}^{t-1} g^s$. Terms with $\mathbb{E}\left\|Q_t^i\left(\nabla f_i(x^t) - \nabla f_i(x^{t-\tau})\right)\right\|^2$ are obtained from the following inequalities (see details in Appendix E):

$$\left\|Q_t^i\left(\nabla f(x) - \nabla f(y)\right)\right\|^2 \le \frac{d^2}{m^2}\left\|\nabla f(x) - \nabla f(y)\right\|^2 \le \frac{d^2 L^2}{m^2}\left\|x - y\right\|^2,$$

Since the evaluation of $\mathbb{E}\left\|x^{t+1} - x^*\right\|^2$ raises the terms of the form $\mathbb{E}\left\|x^{t-\tau} - x^*\right\|^2$, we have to do a summation of $\mathbb{E}\left\|x^{t+1} - x^*\right\|^2$ from $t = \tau$ to $t = T$. These terms greatly complicate the proof of Theorem 2 compared to the unbiased compressors. The results of Theorem 2 can be rewritten as an upper complexity bound on a number of iterations $T$ of the Algorithm 1 by carefully tuning the step size $\gamma$.

**Corollary 1** (Step tuning for Theorem 2).

• *Under the conditions of Theorem 2 in the non-convex case, choosing $\gamma$ as in Appendix E.2, in order to achieve the $\epsilon$-approximate solution (in terms of $\mathbb{E}\left[\left\|\nabla f(x^T)\right\|^2\right] \le \epsilon^2$), it takes*

$$\mathcal{O}\left(\frac{L\tau d^2}{m^2}F_\tau\left(\frac{\delta^2 + 1}{\epsilon^2} + \frac{\sigma^2}{\epsilon^4}\right)\right) \text{ iterations of Algorithm 1.}$$

• *Under the conditions of Theorem 2 in the PL-condition (Assumption 3) case, choosing $\gamma$ as in Appendix E.2 in order to achieve the $\epsilon$-approximate solution (in terms of $\mathbb{E}\left[f(x^t) - f(x^*)\right] \le \epsilon$), it takes*

$$\mathcal{O}\left(\frac{d^2 L\tau}{m^2\mu}\left((\delta^2 + 1)\log\left(\frac{1}{\epsilon}\right) + \frac{\sigma^2}{\mu\epsilon}\right)\right) \text{ iterations of Algorithm 1.}$$

### 2.3 ACCELERATED METHOD

After giving the convergence result for the vanilla distributed SGD with Markovian compression operator, we now move on to the accelerated scheme. Since we do not assume boundedness of the gradient variance, the classical Nesterov acceleration Nesterov (2014) does not produce the expected effect, and therefore an additional momentum has to be introduced Nesterov (2012b); Vaswani et al. (2019). By applying a multistep strategy partially similar to Beznosikov et al. (2023b), we obtain our Algorithm 2.

---

**Algorithm 2** `Accelerated Markovian QSGD (AMQSGD)`

1: **Input:** starting point $x^0 \in \mathbb{R}^d$, step size $\gamma > 0$, momentums $\theta, \eta, \beta, p$, number of iterations $T$
2: **for** $t = 0$ **to** $T$ **do**
3:     Update $x_g^t = \theta x_f^t + (1 - \theta)x^t$
4:     Broadcast $x_g^t$ to all workers
5:     **for** $i = 1$ **to** $n$ in parallel **do**
6:         Set $g_i^t = Q_t^i\left(\nabla f_i(x_g^t)\right)$
7:         Send $g_i^t$ to the server
8:     **end for**
9:     Aggregate $g^t = \frac{1}{n}\sum_{i=1}^{n} g_i^t$
10:     Update $x_f^{t+1} = x_g^t - p\gamma g^t$
11:     Update $x^{t+1} = \eta x_f^{t+1} + (p - \eta)x_f^t$
12:     $+ (1 - p)(1 - \beta)x^t + (1 - p)\beta x_g^t$
13: **end for**

---

**Theorem 3** (Convergence of `AMQSGD` (Algorithm 2)). *Consider Assumptions 1, 2, 4. Let the problem (1) be solved by Algorithm 2. Then for any $\gamma, \varepsilon > 0$, $T > \tau > \tau_{mix}(\varepsilon)$, $\beta, \theta, \eta, p$ satisfying conditions, described in Appendix G.1, it holds that*

$$F_{T+1} = \mathcal{O}\left( \exp\left[ -(T-\tau)\sqrt{\frac{p^2\mu\gamma}{3}} \right] F_\tau + \exp\left[ -T\sqrt{\frac{p^2\mu\gamma}{3}} \right] \Delta_\tau + \frac{\gamma}{\mu}\sigma^2 \right).$$

*Here we use the notations:* $F_t := \mathbb{E}[\|x^t - x^*\|^2 + 3/\mu(f(x_f^t) - f(x^*))]$ *and* $\Delta_\tau \leq \gamma^{1/2}\tau^{-4/3}\mu^{-1/3}\sum_{t=0}^{\tau}\left( \mathbb{E}\|\nabla f(x_g^t)\|^2 + \mathbb{E}\|x^t - x^*\|^2 + \mathbb{E}[f(x_f^t) - f(x^*)] \right).$

The above theorem shows that in the strongly convex case Accelerated Markov QSGD with constant step-size can attain sublinear convergence. In terms of dealing with Markovian stochasticity, its proof follows quite similar ideas as the proof of Theorem 2: here again we use the technique of *stepping back* for mixing time, which allows us to effectively deal with the bias of the gradient estimator. The full proof is provided in Appendix G.3. The results of Theorem 3 can be rewritten as an upper complexity bound on a number of iterations $T$ of the Algorithm 2 by carefully tuning the step size $\gamma$.

**Corollary 2** (Step tuning for Theorem 3). *Under the conditions of Theorem 3, choosing $\gamma$ as in Appendix G.2 in order to achieve the $\epsilon$-approximate solution (in terms of $\mathbb{E}\left[\left\|x^T - x^*\right\|^2\right] \leq \epsilon^2$), it takes*

$$\mathcal{O}\left( \frac{d^2 L^{\frac{2}{3}}\tau^{\frac{4}{3}}}{m^2\mu^{\frac{2}{3}}}\left( (\delta^2+1)\log\left(\frac{1}{\epsilon}\right) + \frac{\sigma^2}{\mu\epsilon} \right) \right) \text{ iterations of Algorithm 2.}$$

## 2.4 DISCUSSION

Our Example 1 and the numerical experiments in Section 3 show that the using of Markovian compressors could lead to a better performance quite well, however, the theoretical guarantees turn out to be poorer than in the unbiased case. In particular, if we use Rand$m$ in the QSGD algorithm, then we observe the following estimates Beznosikov et al. (2023a):

$$X_T = \mathcal{O}\left( (1-\mu\gamma)^T X_0 + \gamma\frac{d}{m}\frac{\sigma^2}{\mu n} \right),$$

where $X_t = \mathbb{E}\left[\|x^t - x^*\|^2\right]$ and $\gamma \lesssim \frac{1}{L(1+d/mn)}$. However, Theorem 2 gives us such estimates:

$$F_T = \mathcal{O}\left( \left(1 - \frac{\mu\gamma}{12}\right)^T F_\tau + \gamma\frac{d^2}{m^2}\frac{\tau L\sigma^2}{\mu} \right),$$

where $F_t := \mathbb{E}\left[f(x^T) - f(x^*)\right]$ and $\gamma \lesssim \frac{m^2}{Ld^2\tau(\delta^2+1)}$. It is important to note that not only has the theory for Markovian compressors not yet been studied, but also dealing with the Markovian stochasticity itself implies quite strict limitations. For instance,

**d/m vs d²/m².** We are forced to uniformly bound the noise of the compressor (linearity in the compression constant is prevented by this) due to the impossibility of using the expectation trick, in contrast to the unbiased case Beznosikov et al. (2023a), where the authors estimated the variance of the compressor noise. The assumption of uniformly bounded noise cannot be rejected by any authors who work with Markovian stochasticity Beznosikov et al. (2023b); Dorfman & Levy (2023); Doan et al. (2020a); Sun et al. (2018); Even (2023), therefore, there is no possibility to achieve linearity in the compression rate in our theoretical guaranties, according to the current theoretical advances.

**Mixing time.** Furthermore, it is imperative to emphasize that it follows from Theorems 2 and 3 that the convergence rate is improved as $\tau$ (and, consequently, $K$) diminishes. In other words, the distribution of the compressor's underlying Markov chain has to converge to a uniform distribution as fast as possible, but empirically one wants the choice of coordinates to depend on previous iterations rather than be random (e.g. for Rand$m$ compressor $\tau = 1, K = 0$). This causes a logical contradiction: while using a large $K$ will theoretically give poorer convergence, in practice algorithms with non-zero values of $K$ perform better (see Section 3). It is also worth mentioning that when Markovian stochasticity is employed, we can never avoid $\tau$ in our estimates, since it appears in the lower bounds on the convergence rate of methods that involve Markovian properties Bresler et al. (2020). Thus, our Algorithms 1 and 2 have a reasonably good polynomial dependence on mixing time (Theorem 2 shows an optimal estimation in terms of $\tau$), considering the fact there are several works Doan et al. (2020b) whose bounds include terms that are even *exponential* in the mixing time.

**L/μ.** In spite of the difficulties listed above, we still can observe that the momentums implementation in Algorithm 2 gives an acceleration in terms of $L/\mu$ compared to vanilla QSGD (Algorithm 1). In

the classical version of accelerated Gradient Descent, one can achieve an acceleration of the form $\sqrt{L/\mu}$ Nesterov (1983), but our analysis allows only to achieve $(L/\mu)^{2/3}$ in Theorem 3. When Markovian stochasticity is employed, it is also possible to achieve estimation of the form $\sqrt{L/\mu}$ Beznosikov et al. (2023b), but it is obtained by using batches with size scaled as $2^j$, where $j$ is drawn from a truncated geometric distribution. Unfortunately, this specific batching technique cannot be applied in our paper, as we consider compressors that act as random sparsification (Definition 4), which necessitates that the gradient be compressed only once at each iteration.

**Variance reduction.** In our paper, we focus on the QSGD method and its accelerated version (Algorithms 1 and 2). However, in modern studies on distributed optimization, techniques of variance reduction are of a great interest (DIANA Mishchenko et al. (2019), MARINA Gorbunov et al. (2021a), DASHA Tyurin & Richtárik (2022)), because these methods converge linearly to the exact solution of the problem (1), while QSGD (Algorithms 1 and 2) converges only to the $\sigma^2$-neighborhood of the solution. We implement Markovian compressors (Definitions 5 and 6) in these methods in our experiments, but we do not provide theoretical guarantees for such algorithms since we have just developed a theoretical baseline for the study of Markovian compressors. This represents a promising direction for future research.

Even though it is not entirely clear whether it is possible to achieve significant improvements in the theoretical results, due to the peculiarities of dealing with Markovian randomness, for now we could only highlight a significantly better performance of Algorithms 1 and 2 compared to a similar algorithms using a vanilla unbiased compressor Rand$m$ (see Section 3).

## 3 EXPERIMENTS

In order to justify the practical usage of the proposed methods and analyze their behavior, we conduct a series of experiments using Markovian compression on distributed optimization problems, specifically logistic regression and neural network-based image classification. We observe that Markovian compressors, when used with `MQSGD` and `AMQSGD`, as well as with classical SGD and DIANA Mishchenko et al. (2019), improve convergence on several benchmarks. Appendix H provides a description of the technical setup, extended experiments with hyperparameters analysis, and an application of Markovian compressors to model-parallel neural network training.

### 3.1 LOGISTIC REGRESSION

Firstly, we experiment on a classification task using a logistic regression model with $L_2$ regularization of the form:

$$\min_{w \in \mathbb{R}^d} \left\{ f(w) = \frac{1}{n} \sum_{i=1}^{n} \log(1 + \exp(-y_s w^T x_s)) + \lambda \|w\|^2 \right\},$$

with a regularization term $\lambda = 0.05$. The dataset is split among $n = 10$ clients. We use `Mushrooms`, `A9A`, and `W8A` datasets from LibSVM Chang & Lin (2011) and `MNIST` Deng (2012). Experiments are conducted using Python 3.10 and PyTorch, and a distributed environment is simulated. We experiment with `MQSGD`, `AMQSGD`, and `DIANA` optimizers, employing Rand-10% as a sparsification compressor. Markovian compressors were utilized independently on each client, with normalization activation function, and with all hyperparameters being fine-tuned.

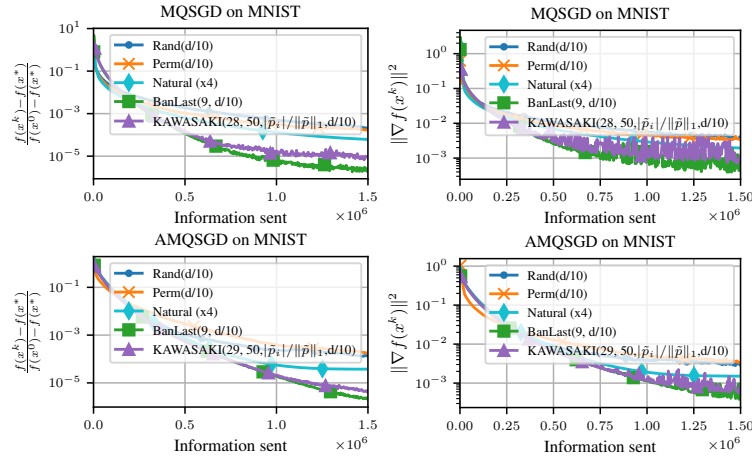

Figure 1: Logistic Regression on MNIST experiments results. All hyperparameters are fine-tuned, and best runs are selected.

Figure 1 shows the convergence of the Rand-10% baseline and Markovian compressors on the `MQSGD` and `AMQSGD` algorithms on MNIST dataset. Both Markovian compressors achieve faster convergence

than the baseline and more complex compressors like PermK Szlendak et al. (2021) and Natural compressors Horvath et al. (2022). In most of our results, `BanLast` and `KAWASAKI` show similar performance with fine-tuned hyperparameters. Experiments on other datasets, and tuning history size $K$ tuning analysis appear in Appendix H.2. Additionally, as our compressors are fully compatible with classical compressors, we conduct experiments on combination with Natural compression in Appendix H.5.

## 3.2 NEURAL NETWORKS

We also apply Markovian compressors in more complex optimization tasks, such as image classification on `CIFAR-10` Krizhevsky et al. (2009) dataset with `ResNet-18` convolutional neural network He et al. (2016). Formally, we solve optimization problem:

$$\min_{w \in \mathbb{R}^d} \left\{ f(w) = \frac{1}{n} \sum_{i=1}^{n} l(\text{softmax}(f(x_i, w)), y_i) \right\},$$

where $x_i$ is a training image, $y_i$ is its respective class, and $l()$ is a cross-entropy loss function. Dataset is split equally between $n = 5$ clients. We use Rand-5% sparsification operator and SGD optimizer with cosine annealing LR schedule. Hyperparameters, such as the learning rate, batch size, and Markovian-specific ones are fine-tuned.

Figure 2 depicts the training loss and gradient norm, with the aggregate values shown in Table 1. As in the previous case, the application of the Markovian compressor favours faster convergence and better validation results. Note that for more complex optimization task, smoother history accumulation (as in `KAWASAKI`) is required.

Figure 3 presents comparison with Permutation and Natural compression, which confirm practical usefullness of Markovian compressors on more complex and non-convex optimization problems. Note that our compressors can be applied in combination with complex randomized compressor like Natural compression, making our method even more flexible.

Table 1: Numerical results of training ResNet-18 on CIFAR-10 with different compressors. Each cell represents mean ± standard deviation over 5 runs.

|  | Rand-5% | Banlast | KAWASAKI |
|---|---|---|---|
| Train Loss | $0.0743 \pm 0.003$ | $0.0734 \pm 0.003$ | $\mathbf{0.0305 \pm 0.001}$ |
| Gradient Norm | $1.403 \pm 0.029$ | $1.383 \pm 0.035$ | $\mathbf{0.745 \pm 0.015}$ |
| Test Accuracy | $87.9 \pm 0.179$ | $88.0 \pm 0.122$ | $\mathbf{89.05 \pm 0.294}$ |

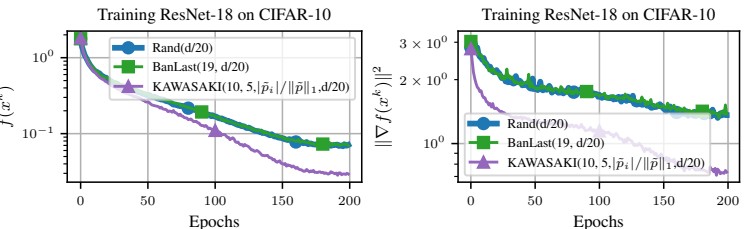

Figure 2: Image classification with ResNet-18 on CIFAR-10 experiments results. Best runs for each method are displayed.

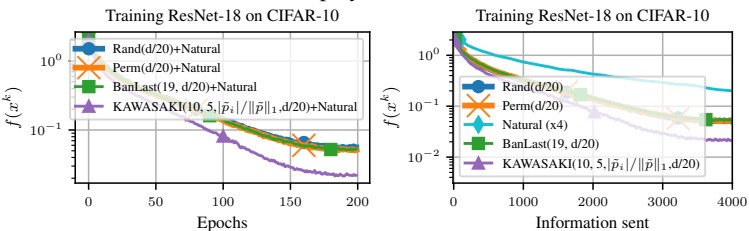

Figure 3: Comparison with other compressors on Resnet-18 training on CIFAR-10 dataset for Rand-5% sparsification on $N = 20$ clients. Natural compression factor is 4. Left figure is sequential combination with Natural compression. Right figure is comparison against PermK and Natural compressors independently, with information sent on x-axis.

## 4 CONCLUSION

In this paper, we propose a family of compression schemes, which takes into account previous iterations of algorithm and transform the input vector according to a Markov chain. We develop two sparsification methods `BanLast` (Definition 5) and `KAWASAKI` (Definition 6) based on this idea. These compressors are implemented in QSGD (Algorithm 1) and accelerated QSGD (Algorithm 2). We provide convergence rates under different assumptions on the objective function (Theorems 2 and 3). In experiments, we show that our compression methods outperform the baselines in the deep neural network optimisation problem. Future research may consider the implementation of our Markovian compressors in other optimization methods, e.g. using the variance reduction techniques.

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

# Supplementary Material

## CONTENTS

# A AUXILIARY LEMMAS AND FACTS

In this section we list auxiliary facts and our results that we use several times in our proofs.

## A.1 CAUCHY–SCHWARZ INEQUALITY

For all $x, y \in \mathbb{R}^d$

$$\langle x, y \rangle \leq \|x\| \, \|y\| \, .$$

## A.2 FENCHEL-YOUNG INEQUALITY

For all $x, y \in \mathbb{R}^d$ and $\beta > 0$

$$2 \langle x, y \rangle \leq \beta^{-1} \|x\|^2 + \beta \|y\|^2 .$$

# B MATHEMATICAL CALCULATIONS FROM EXAMPLE 1

By definition of the mathematical expectation of an integer positive random variable $Z$, we obtain that $\mathbb{E}[Z] = \sum_{s=1}^{\infty} s \cdot \mathbb{P}\{Z = s\}$. In our problem, $Z$ is the number of an iteration where we first selected the desired coordinate. For Rand$m$ compressor, we have $\mathbb{P}\{Z = s\} = \frac{m}{d} \cdot \left(1 - \frac{m}{d}\right)^{s-1}$. The first term is the probability of picking the desired coordinate at iteration $s$ and the second term is the probability of not picking the desired coordinate at iterations from $1$ to $s - 1$. Using this, the mathematical expectation of the number of steps to quit the point $x^t$ for Rand$m$ compressor is equal to

$$\sum_{s=1}^{\infty} s \left(1 - \frac{m}{d}\right)^{s-1} \frac{m}{d} = \frac{d}{m}. \tag{6}$$

Now we calculate the expectation for `BanLast(K, m)` compressor (Definition 5). If $s > K$, similarly to the Rand$m$ case, we obtain that $\mathbb{P}\{Z = s\} = \frac{m}{d - Km} \left(1 - \frac{m}{d - Km}\right)^{s-1}$, because we cannot choose $Km$ coordinates. If $s \leq K$, then the formula of $\mathbb{P}\{Z = s\}$ becomes a bit more complicated, because the probability of not picking the desired coordinate at iterations from $1$ to $s - 1$ is different at each iteration and is equal to $\prod_{h=0}^{s-2} \left(1 - \frac{m}{d - hm}\right)$. If $s = 1$, then this probability is equal to one. Using this, we can calculate the mathematical expectation of the number of steps to leave the point $x^t$ for `BanLast(K, m)` compressor:

$$\sum_{s=1}^{K} \frac{sm}{d - (s-1)m} \prod_{h=0}^{s-2} \left(1 - \frac{m}{d - hm}\right) + \sum_{s=K+1}^{\infty} s \left(1 - \frac{m}{d - Km}\right)^{s-1} \frac{m}{d - Km}$$

$$= \sum_{s=1}^{K} \frac{sm}{d - (s-1)m} \prod_{h=0}^{s-2} \left(1 - \frac{m}{d - hm}\right) + \frac{d}{m} \left(1 - \frac{m}{d - Km}\right)^{K} \tag{7}$$

$$= \sum_{s=1}^{K} \frac{s}{\alpha - (s-1)} \prod_{h=0}^{s-2} \left(1 - \frac{1}{\alpha - h}\right) + \alpha \left(1 - \frac{1}{\alpha - K}\right)^{K},$$

where we used the notation $\alpha = d/m$ to show that (7) depends only on $d/m$, but not on $d$ and $m$ separately. We can consider (7) as an optimization problem with respect to $K$. Since $K$ is an integer and the objective function in (7) is complex, we numerically find the optimal $K$ for different $\alpha$. For the sake of clarity, we show the difference between formulas (6) and (7) on Figure 4(c).

We consider $\alpha \in [5.3, 6.7, 8.3, 10, 11.1, 12.5, 14.3, 16.7, 20]$ and find the optimal $K$ by a complete brute force search – see Figure 4 (a). Then, we perform a linear approximation and obtain the formula $K^*(\alpha) \approx 0.7323\alpha$ – see Figure 4 (b). Since the correlation coefficient between the points and the approximated line is equal to $0.73$, we can consider this formula to be accurate enough for practical applications.

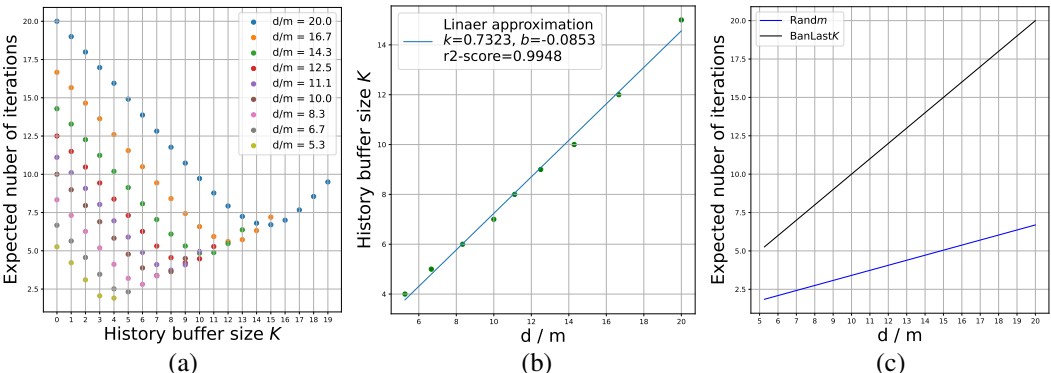

(a)              (b)              (c)

Figure 4: Theoretical estimate on dependence of history buffer size $K$ on parameter $\alpha = d/m$: (a) represents expected number of iterations required to transfer all coordinates to server on history buffer size $K$ for different $\alpha$, (b) represents scaling of optimal history buffer size $K^*$ on $\alpha$. (c) represents comparison of expected number of iterations required to transfer all coordinates to server on problems parameter $\alpha$ for Rand$m$ and BanLast$K$.

## C    PROOF OF THEOREM 1

**Lemma 1.** *If $P$ is a transition matrix of a finite homogeneous Markov chain, i.e.*

$$P := (p_{ij})_{i,j=1}^{n},$$

*where $p_{ij}$ is probability of moving from i to j in one time step. And the matrix $P$ is symmetric, i.e. $P^T = P$, then stationary distribution exists and it is uniformly distributed.*

*Proof of Lemma 1.* Let us look at uniform distribution

$$\pi := \left(\frac{1}{n}, \frac{1}{n}, \ldots, \frac{1}{n}\right).$$

We can easily obtain that $\pi$ is a stationary distribution, using symmetry and stochastic property of matrix $P$:

$$\pi P = \frac{1}{n}\mathbf{1}^T P = \frac{1}{n}(P\mathbf{1})^T = \frac{1}{n}\mathbf{1}^T = \pi.$$

$\square$

*Proof of Theorem 1.* We consider states of Markov chain as $s := \{\nu_1, \nu_2, ..., \nu_K\}_{\nu_1,...,\nu_K \in M}$, where $M$ is the set of all subsets of $\overline{1,d}$ of size $m$. We define $p(s, s', i)$ as the probability to move from state $s$ to state $s'$ for the number of steps $i$.

• For both compressors BanLast$(K, m)$ (Definition 5) and KAWASAKI$(K, b, \pi_\Delta, m)$ (Definition 6) corresponding Markov chain is finite and indecomposable.

The finiteness of the chain is apparent, as the number of states can be explicitly expressed as $|M| = (C_d^m)^K$. We show that both chains are indecomposable below. Then we deduce that the chain is ergodic based on the Ergodic Theorem Neumann (1932). Thus, we know that a stationary distribution exists. Than we show that the statinary distribution is uniform over the set of states using Lemma 1.

All that remains is to show that both chains are indecomposable and that transition matrixes for both chaines are symmetric.

We will start with `BanLast`$(K, m)$. Restriction on $K, m$ and $d$ is $d > (K + 1)m$. That makes obvious that any two states are communicated, i.e. for any $s, s'$ there exists way from $s$ to $s'$. Thus, the Markov chain is indecomposable.

For the compressor probability to move from $s$ to $s'$ in one time step can be explicitly expressed as:

$$p(s, s', 1) = \left( \frac{1}{C_{d-Km}^m} \right)^K,$$

where $C_{d-Km}^m = \frac{(d-Km)!}{m!(d-(K+1)m)!}$ is a binomial coefficient. And all these states are equal in probability. If $d = (K + 1)m$, then for $s$ there will be only one set $s'$, such that $p(s, s', 1) > 0$, in this case chain will not be ergodic. If $d > (K + 1)m$, then there are more then one state $s'$, for witch $p(s, s', 1) > 0$, therefore chain will be ergodic.

• According to the Ergodic Theorem, $\rho = (1 - \delta)^{1/N_0}$ and $C = (1 - \delta)^{-1}$, where $N_0$ is the minimal number of iterations through which is strictly greater then zero and $\delta := \min_{s,s'}\{p(s, s', N_0)\} > 0$. For `BanLast`$(K, m)$ in case of $d > (2K + 1)m$ it holds that

$$N_0 = 2 \text{ and } \delta = p(s, s, 2) = \left( \frac{C_{d-2Km}^m}{C_{d-Km}^m} \right)^K \cdot \left( \frac{1}{C_{d-Km}^m} \right)^K,$$

because the smallest probability is to return to state $s$ in two steps.

• For `KAWASAKI`$(K, b, \pi_\Delta, m)$ from any given state, there exists a path to any other state in just one iteration, because probabilities to choose any set of coordinates $\nu$ are non-zero. Thus, the corresponding markov chain is indecomposable.

We focus on the case where $K = 1$ and that generalize analysis to accommodate larger values of $K$. Let us look at probabilities to move from $\nu_i$ to $\nu_j$ and from $\nu_j$ to $\nu_i$. We show that both these probabilities correspond to random choice of the same indexes with the same distribution vector $p$, defined in 6, i.e. the probabilities are equal. For this case let us define $\nu$ as operator

$$\Psi_i(\overline{1, d}) := \nu_i,$$

i.e. operator chooses indexes that are in $\nu_i$ from $\overline{1, d}$. And

$$\Phi(p, \Psi_i) := \mathbb{P}\{\text{choose } \nu_i \text{ with distribution vector } p\}.$$

According to 6, probability to move from $\nu_i$ to $\nu_j$ equals a probability to choose indexes $\nu_j$ with distribution

$$p_i = \pi_\Delta(\widetilde{p}_i),$$

where

$$\widetilde{p}_i^k = \begin{cases} 1/bd & \text{if } k \in \nu_i \\ 1/d & \text{if } k \notin \nu_i \end{cases},$$

i.e.

$$p_{ij} = \Phi(p_i, \Psi_j).$$

By the definition of $\Phi$, for arbitrary permutation $\phi$ and index choice $\Psi$ holds

$$\Phi(\phi(p), \Psi \circ \phi) = \Phi(p, \Psi).$$

Now we point out that for arbitrary $\nu_i$ and $\nu_j$ exists permutation $\phi_{ij}$, such that

$$\Psi_j \circ \phi_{ij} = \Psi_i.$$

For such permutation holds $\phi_{ij}(\widetilde{p}_i) = \widetilde{p}_j$, i.e. the permutations moves indexes from $\nu_i$ to indexes from $\nu_j$. Then we need to use the property of $\pi_\Delta$ to get the same equality for $p_i, p_j$:

$$\phi_{ij}(p_i) = \phi_{ij}(\pi_\Delta(\widetilde{p}_i)) = \pi_\Delta \phi_{ij}((\widetilde{p}_i)) = \pi_\Delta(p_j).$$

This allows us to write

$$p_{ij} = \Phi(p_i, \Psi_j) = \Phi(\phi_{ij}(p_i), \Psi_j \circ \phi_{ij}) = \Phi(p_j, \Psi_i) = p_{ji}.$$

Thus we get equality of probabilities to move from $\nu_j$ to $\nu_i$ and to opposite way.

Now we can easily generalize the proof for arbitrary $K$. All that is required is to consider, instead of the sets of indices $\nu$, combinations of sets of indices that were chosen for transmission over the previous $K$ steps. In this way, the number of states is increased, but the logic of reasoning remains unchanged.

• As was mentioned above, for KAWASAKI$(K, b, \pi_\Delta, m)$ $N_0 = 1$. We now compute $\delta := p(s, s, 1)$, where $s = \{\nu, ..., \nu\}$, where $\nu$ occurs $K$ times. In this case probability to choose $\nu$ another $K$ times is equal to $\mathbb{P}\{j \in \nu\}^{mK}$. And

$$\mathbb{P}\{j \in \nu\} = \min\left\{\pi_\Delta\left[\widetilde{p} := \Big(\underbrace{\frac{1/d}{b^K}, ..., \frac{1/d}{b^K}}_{m}, \underbrace{\frac{1/d}{1}, ..., \frac{1/d}{1}}_{d-m}\Big)^T\right]\right\}.$$

If we consider $(\pi_\Delta(\widetilde{p}))_j = |\widetilde{p}_j|/\|\widetilde{p}\|_1$, then, since $\|\widetilde{p}\|_1 = \frac{1}{db^k}(db^K - m(b^K - 1))$, it hold that $\delta = (db^K - m(b^K - 1))^{-mK}$. This finishes the proof. $\qquad\square$

## D MAIN LEMMAS

**Lemma 2.** *For any $i \in \overline{1, n}$, $\varepsilon > 0$, $\tau > \tau_{mix}(\varepsilon)$, $t > \tau$, for any $a^{t-\tau}, b^{t-\tau} \in \mathbb{R}^d$, such that if we fix all randomness up to step $t - \tau$, $a^{t-\tau}$ and $b^{t-\tau}$ become non-random, it holds that*

$$\mathbb{E}\left[\langle Q_t^i(a^{t-\tau}) - a^{t-\tau}, b^{t-\tau}\rangle\right] \leq \frac{\varepsilon d}{m}\mathbb{E}\left[\|a^{t-\tau}\| \cdot \|b^{t-\tau}\|\right].$$

*Proof.* We begin by using tower property:

$$\mathbb{E}\left[\langle Q_t^i(a^{t-\tau}) - a^{t-\tau}, b^{t-\tau}\rangle\right] = \mathbb{E}\left[\langle \mathbb{E}_{t-\tau}\left[Q_t^i(a^{t-\tau}) - a^{t-\tau}\right], b^{t-\tau}\rangle\right], \qquad (8)$$

where $\mathbb{E}_{t-\tau}[\cdot]$ is the conditional expectation with fixed randomness of all steps up to $t - \tau$. Since on a step $t$ we compress vector $a^{t-\tau}$ according to distribution $\pi_t^i$ by the formula $Q_t^i(a^{t-\tau}) = d/ma^{t-\tau} \odot \mathbb{1}(\nu_t^i)$, where $\nu_t^i$ is some set of $m$ coordinates : $\nu_t^i \subset \overline{1, d}$ and $\mathbb{1}(\nu_t^i)$ is vector with 1 on coordinates $\nu_t^i$ on 0 otherwise. Using this we can obtain:

$$\mathbb{E}_{t-\tau}\left[Q_t^i(a^{t-\tau}) - a^{t-\tau}\right] = \sum_{\widetilde{\nu}_i \in M}\left(\mathbb{P}_{t-\tau}\{\nu_t^i = \widetilde{\nu}_i\} - \frac{1}{C_d^m}\right)a^{t-\tau} \odot \mathbb{1}(\widetilde{\nu}_i)\frac{d}{m},$$

where $M$ is set of all subsets of $\overline{1, d}$ of size $m$. This equality follows from the fact that $\sum_{\widetilde{\nu}_i \in M} a^{t-\tau} \odot \mathbb{1}(\widetilde{\nu}_i) = C_{d-1}^{m-1}a^{t-\tau}$ and $C_{d-1}^{m-1}/C_d^m = m/d$. Now with the help of Cauchy–Schwarz inequality A.1 we can estimate (8):

$$(8) \leq \mathbb{E}\left[\sum_{\widetilde{\nu}_i \in M}\left|\mathbb{P}_{t-\tau}\{\nu_t^i = \widetilde{\nu}_i\} - \frac{1}{C_d^m}\right|\|a^{t-\tau} \odot \mathbb{1}(\widetilde{\nu}_i)\|\frac{d}{m}\|b^{t-\tau}\|\right]. \qquad (9)$$

Since $t > \tau$ and $\tau > \tau_{\text{mix}}(\varepsilon)$ it holds that $\left|\mathbb{P}_{t-\tau}\{\nu_t^i = \widetilde{\nu}_i\} - 1/C_d^m\right| \leq \varepsilon \cdot 1/C_d^m$, because stationary distribution of our Markov chain is uniform. Using the fact that $\|a^{t-\tau} \odot \mathbb{1}(\widetilde{\nu}_i)\| \leq \|a^{t-\tau}\|$ we can obtain:

$$(9) \leq \mathbb{E}\left[\sum_{\widetilde{\nu}_i \in M}\varepsilon\frac{1}{C_d^m}\|a^{t-\tau}\|\frac{d}{m}\|b^{t-\tau}\|\right] = \frac{\varepsilon d}{m}\mathbb{E}\left[\|a^{t-\tau}\| \cdot \|b^{t-\tau}\|\right].$$

This finishes the proof.

$\qquad\square$

**Lemma 3.** *For any $i \in \overline{1,n}$, $\varepsilon > 0$, $\tau > \tau_{mix}(\varepsilon)$, $t > \tau$, for any $a^{t-\tau} \in \mathbb{R}^d$, such that if we fix all randomness up to step $t - \tau$, $a^{t-\tau}$ becomes non-random, it holds that*

$$\mathbb{E}\left[\left\|\mathbb{E}_{t-\tau}\left[Q_t^i(a^{t-\tau})\right] - a^{t-\tau}\right\|^2\right] \leq \frac{\varepsilon^2 d^2}{m^2}\mathbb{E}\left[\left\|a^{t-\tau}\right\|^2\right].$$

*Proof.* Using same notation as in the proof of Lemma 3 we obtain

$$\mathbb{E}\left[\left\|\mathbb{E}_{t-\tau}\left[Q_t^i(a^{t-\tau})\right] - a^{t-\tau}\right\|^2\right] = \mathbb{E}\left[\left\|\sum_{\widetilde{\nu}_i \in M}\left(\mathbb{P}_{t-\tau}\left\{\nu_t^i = \widetilde{\nu}_i\right\} - \frac{1}{C_d^m}\right)\frac{d}{m}a^{t-\tau} \odot \mathbb{1}(\widetilde{\nu}_i)\right\|^2\right]$$

$$\leq \mathbb{E}\left[\frac{d^2}{m^2}C_d^m\sum_{\widetilde{\nu}_i \in M}\left(\left|\mathbb{P}_{t-\tau}\left\{\nu_t^i = \widetilde{\nu}_i\right\} - \frac{1}{C_d^m}\right|^2\left\|a^{t-\tau} \odot \mathbb{1}(\widetilde{\nu}_i)\right\|^2\right)\right].$$

Since $t > \tau$ and $\tau > \tau_{mix}(\varepsilon)$ it holds that $\left|\mathbb{P}_{t-\tau}\left\{\nu_t^i = \widetilde{\nu}_i\right\} - 1/C_d^m\right| \leq \varepsilon \cdot 1/C_d^m$, because stationary distribution of our Markov chain is uniform. Using the fact that $\left\|a^{t-\tau} \odot \mathbb{1}(\widetilde{\nu}_i)\right\| \leq \left\|a^{t-\tau}\right\|$ we can obtain:

$$\mathbb{E}\left[\left\|\mathbb{E}_{t-\tau}\left[Q_t^i(a^{t-\tau})\right] - a^{t-\tau}\right\|^2\right] \leq \frac{\varepsilon^2 d^2}{m^2}\mathbb{E}\left[\left\|a^{t-\tau}\right\|^2\right].$$

This finishes the proof.

$\square$

**Lemma 4.** *For any $i \in \overline{1,n}$ and $a \in \mathbb{R}^d$ it holds that*

$$\left\|Q^i(a)\right\|^2 \leq \frac{d^2}{m^2}\|a\|^2 \quad and \quad \left\|Q^i(a) - a\right\|^2 \leq 4\frac{d^2}{m^2}\|a\|^2.$$

*Proof.* Consider the first inequality. Since $Q^i(a) = d/ma \odot \mathbb{1}(\nu^i)$, then $\left\|Q^i(a)\right\| \leq d/m\|a\|$, therefore

$$\left\|Q^i(a)\right\|^2 \leq \frac{d^2}{m^2}\|a\|^2.$$

Consider the second inequality. Using Fenchel-Young inequality A.2 with $\beta = 1$ we can estimate

$$\left\|Q^i(a) - a\right\|^2 \leq 2\left\|Q^i(a)\right\|^2 + 2\|a\|^2 \leq 2\left(\frac{d^2}{m^2} + 1\right)\|a\|^2 \leq 4\frac{d^2}{m^2}\|a\|^2.$$

This finishes the proof. $\square$

**Corollary 3.** *For any $i \in \overline{1,n}$, $\varepsilon > 0$, $\tau > \tau_{mix}(\varepsilon)$, $t > \tau$, for any $a^t, b^t \in \mathbb{R}^d$, such that if we fix all randomness up to step $t$, $a^t$ and $b^t$ become non-random. And for any $\hat{a}^{t-\tau}, \hat{b}^{t-\tau}$, such that if we fix all randomness up to step $t - \tau$, $\hat{a}^{t-\tau}$ and $\hat{b}^{t-\tau}$ become non-random, it holds that*

$$2\left|\mathbb{E}\left[\left\langle Q_t^i\left(a^t\right) - a^t, b^t\right\rangle\right]\right| \leq \frac{\varepsilon d}{m\beta_0}\mathbb{E}\left[\left\|\hat{a}^{t-\tau}\right\|^2\right] + \frac{\varepsilon d\beta_0}{m}\mathbb{E}\left[\left\|\hat{b}^{t-\tau}\right\|^2\right] + \frac{1}{\beta_2}\mathbb{E}\left[\left\|b^t\right\|^2\right],$$

$$+ \left(\frac{1}{\beta_1} + \frac{1}{\beta_3}\right)\mathbb{E}\left[\left\|b^t - \hat{b}^{t-\tau}\right\|^2\right] + 4\frac{d^2}{m^2}\beta_3\mathbb{E}\left[\left\|a^t\right\|^2\right] + 4\frac{d^2(\beta_1 + \beta_2)}{m^2}\mathbb{E}\left[\left\|a^t - \hat{a}^{t-\tau}\right\|^2\right]$$

*where $\beta_0, \beta_1, \beta_2, \beta_3 > 0$.*

*Proof.* Using straightforward algebra we obtain

$$
\begin{aligned}
\mathbb{E}\left[\left\langle Q_t^i\left(a^t\right)-a^t, b^t\right\rangle\right] &= \mathbb{E}\left[\left\langle Q_t^i\left(\hat{a}^{t-\tau}\right)-\hat{a}^{t-\tau}, \hat{b}^{t-\tau}\right\rangle\right] \\
&\quad - \mathbb{E}\left[\left\langle Q_t^i\left(a^t-\hat{a}^{t-\tau}\right)-a^t+\hat{a}^{t-\tau}, b^t-\hat{b}^{t-\tau}\right\rangle\right] \\
&\quad + \mathbb{E}\left[\left\langle Q_t^i\left(a^t-\hat{a}^{t-\tau}\right)-a^t+\hat{a}^{t-\tau}, b^t\right\rangle\right] \\
&\quad + \mathbb{E}\left[\left\langle Q_t^i\left(a^t\right)-a^t, b^t-\hat{b}^{t-\tau}\right\rangle\right].
\end{aligned}
$$

Using Lemma 2 with $a^{t-\tau} = \hat{a}^{t-\tau}, b^{t-\tau} = \hat{b}^{t-\tau}$ and Fenchel-Young inequality A.2 with $\beta_1, \beta_2, \beta_3 > 0$ we obtain:

$$
\begin{aligned}
2\left|\mathbb{E}\left[\left\langle Q_t^i\left(a^t\right)-a^t, b^t\right\rangle\right]\right| &\leq 2\frac{\varepsilon d}{m}\mathbb{E}\left[\left\|\hat{a}^{t-\tau}\right\| \cdot \left\|\hat{b}^{t-\tau}\right\|\right] \\
&\quad + \beta_1\mathbb{E}\left[\left\|Q_t^i\left(a^t-\hat{a}^{t-\tau}\right)-a^t+\hat{a}^{t-\tau}\right\|^2\right] + \frac{1}{\beta_1}\mathbb{E}\left[\left\|b^t-\hat{b}^{t-\tau}\right\|^2\right] \\
&\quad + \beta_2\mathbb{E}\left[\left\|Q_t^i\left(a^t-\hat{a}^{t-\tau}\right)-a^t+\hat{a}^{t-\tau}\right\|^2\right] + \frac{1}{\beta_2}\mathbb{E}\left[\left\|b^t\right\|^2\right] \\
&\quad + \beta_3\mathbb{E}\left[\left\|Q_t^i\left(a^t\right)-a^t\right\|^2\right] + \frac{1}{\beta_3}\mathbb{E}\left[\left\|b^t-\hat{b}^{t-\tau}\right\|^2\right].
\end{aligned}
$$

Using Lemma 4 and Fenchel-Young inequality A.2 with $\beta_0 > 0$ we obtain

$$
\begin{aligned}
2\left|\mathbb{E}\left[\left\langle Q_t^i\left(a^t\right)-a^t, b^t\right\rangle\right]\right| &\leq \frac{\varepsilon d}{m\beta_0}\mathbb{E}\left[\left\|\hat{a}^{t-\tau}\right\|^2\right] + \frac{\varepsilon d\beta_0}{m}\mathbb{E}\left[\left\|\hat{b}^{t-\tau}\right\|^2\right] \\
&\quad + 4\frac{d^2}{m^2}\left(\beta_1+\beta_2\right)\mathbb{E}\left[\left\|a^t-\hat{a}^{t-\tau}\right\|^2\right] + \left(\frac{1}{\beta_1}+\frac{1}{\beta_3}\right)\mathbb{E}\left[\left\|b^t-\hat{b}^{t-\tau}\right\|^2\right] \\
&\quad + 4\frac{d^2}{m^2}\beta_3\mathbb{E}\left[\left\|a^t\right\|^2\right] + \frac{1}{\beta_2}\mathbb{E}\left[\left\|b^t\right\|^2\right].
\end{aligned}
$$

This finishes the proof. $\square$

**Lemma 5.** *Assume 4, then for any $x \in \mathbb{R}^d$ it holds that*

$$
\frac{1}{n}\sum_{i=1}^n \|\nabla f_i(x)\|^2 \leq 2(\delta^2+1)\|\nabla f(x)\|^2 + 2\sigma^2.
$$

*Proof.* Using straightforward algebra and Fenchel-Young inequality A.2 with $\beta = 1$ we obtain

$$
\begin{aligned}
\frac{1}{n}\sum_{i=1}^n \|\nabla f_i(x)\|^2 &\leq \frac{2}{n}\sum_{i=1}^n \|\nabla f_i(x) - \nabla f(x)\|^2 + 2\|\nabla f(x)\|^2 \\
&\leq 2(\delta^2+1)\|\nabla f(x)\|^2 + 2\sigma^2.
\end{aligned}
$$

The last inequity follows from 4. This finishes the proof. $\square$

# E    EXTENSIONS FOR THEOREM 2

## E.1    FULL VERSION OF THEOREM 2

**Theorem 4** (Convergence of MQSGD (Algorithm 1), extension of 2)**.** *Consider Assumptions 1, 4 and 5. Let problem* (1) *be solved by Algorithm 1.*

- *For any $\varepsilon > 0$, $\gamma > 0$, $\tau > \tau_{mix}(\varepsilon)$ and $T > \tau$ satisfying*

$$\gamma \lesssim \frac{m^2}{d^2 L(\delta^2 + 1)\tau} \quad and \quad \varepsilon \lesssim \frac{m^2}{d^2(\delta^2 + 1)},$$

*it holds that*

$$\mathbb{E}\left[\left\|\nabla f(\widehat{x}^T)\right\|^2\right] = \mathcal{O}\left(\frac{F_\tau}{\gamma T} + \frac{\gamma L \tau d^2}{m^2}\sigma^2\right),$$

*where $\widehat{x}^T$ is chosen uniformly from $\{x^t\}_{t=0}^T$.*

- *If $f$ additionally verifies the PL-condition (Assumption 3), then for any $\varepsilon > 0$, $\gamma > 0$, $\tau > \tau_{mix}(\varepsilon)$ and $T > \tau$ satisfying*

$$\gamma \lesssim \frac{m^2}{Ld^2\tau(\delta^2 + 1)} \quad and \quad \varepsilon = \sqrt{\gamma L\tau} \lesssim \frac{m}{d\sqrt{\delta^2 + 1}},$$

*it holds that*

$$F_T = \mathcal{O}\left(\left(1 - \frac{\mu\gamma}{12}\right)^{T-\tau}F_\tau + \frac{\gamma d^2 L\tau}{\mu m^2}\sigma^2\right).$$

*Here we use a notation $F_t := \mathbb{E}\left[f(x^t) - f(x^*)\right].$*

## E.2 FULL VERSION OF COROLLARY 1

**Corollary 4** (Step tuning for Theorem 2, extension of Corollary 1)**.**
- *Under the conditions of Theorem 2 in the non-convex case, choosing $\gamma$ as*

$$\gamma \lesssim \frac{m}{d\sqrt{L\tau}}\min\left\{\frac{m}{d(\delta^2 + 1)\sqrt{L\tau}} \; ; \; \sqrt{\frac{F_\tau}{T\sigma^2}},\right\},$$

*in order to achieve $\epsilon$-approximate solution (in terms of $\mathbb{E}\left[\left\|\nabla f(x^T)\right\|^2\right] \le \epsilon^2$) it takes*

$$\mathcal{O}\left(\frac{L\tau d^2}{m^2}F_\tau\left(\frac{\delta^2 + 1}{\epsilon^2} + \frac{\sigma^2}{\epsilon^4}\right)\right) \text{ iterations of Algorithm 1.}$$

- *Under the conditions of Theorem 2 in the PL-condition (Assumption 3) case, choosing $\gamma$ as*

$$\gamma \lesssim \min\left\{\frac{m^2}{Ld^2\tau(\delta^2 + 1)} \; ; \; \frac{\log\left(\max\left\{2; \frac{\mu^2 m^2 F_\tau T}{d^2 L\tau\sigma^2}\right\}\right)}{\mu T}\right\},$$

*in order to achieve $\epsilon$-approximate solution (in terms of $\mathbb{E}\left[f(x^t) - f(x^*)\right] \le \epsilon$) it takes*

$$\mathcal{O}\left(\frac{d^2 L\tau}{m^2\mu}\left((\delta^2 + 1)\log\left(\frac{1}{\epsilon}\right) + \frac{\sigma^2}{\mu\epsilon}\right)\right) \text{ iterations of Algorithm 1.}$$

## E.3 PROOF OF THEOREM 2, NON-CONVEX CASE

*Proof.* Denoting $F_t := \mathbb{E}\left[f(x^t) - f(x^*)\right]$, we have using $L$-smoothness:

$$F_{t+1} - F_t \le -\gamma\mathbb{E}\left[\left\langle\frac{1}{n}\sum_{i=1}^n Q_t^i(\nabla f_i(x^t)), \nabla f(x^t)\right\rangle\right] + \frac{\gamma^2 L}{2}\mathbb{E}\left[\left\|\frac{1}{n}\sum_{i=1}^n Q_t^i(\nabla f_i(x^t))\right\|^2\right].$$

$$(10)$$

Consider $-\gamma\mathbb{E}\left[\left\langle\frac{1}{n}\sum_{i=1}^{n}Q_t^i(\nabla f_i(x^t)),\nabla f(x^t)\right\rangle\right]$. Using straightforward algebra: $\pm\nabla f_i(x^{t-\tau})$ and $\pm\nabla f(x^{t-\tau})$ we can re-write this term:

$$-\gamma\,\mathbb{E}\left[\left\langle\frac{1}{n}\sum_{i=1}^{n}Q_t^i(\nabla f_i(x^t)),\nabla f(x^t)\right\rangle\right]$$

$$=\underbrace{-\gamma\mathbb{E}\left[\left\langle\frac{1}{n}\sum_{i=1}^{n}Q_t^i(\nabla f_i(x^{t-\tau})),\nabla f(x^{t-\tau})\right\rangle\right]}_{①}$$

$$\underbrace{-\gamma\mathbb{E}\left[\left\langle\frac{1}{n}\sum_{i=1}^{n}Q_t^i(\nabla f_i(x^t)),\nabla f(x^t)-\nabla f(x^{t-\tau})\right\rangle\right]}_{②}$$

$$\underbrace{-\gamma\mathbb{E}\left[\left\langle\frac{1}{n}\sum_{i=1}^{n}Q_t^i(\nabla f_i(x^t)-\nabla f_i(x^{t-\tau})),\nabla f(x^{t-\tau})\right\rangle\right]}_{③}.$$

Consider ①. Using straightforward algebra, tower property, Lemmas 3 and 5 we obtain

$$① = -\gamma\mathbb{E}\left[\left\langle\frac{1}{n}\sum_{i=1}^{n}\mathbb{E}_{t-\tau}\left[Q_t^i(\nabla f_i(x^{t-\tau}))\right],\nabla f(x^{t-\tau})\right\rangle\right]$$

$$=-\frac{\gamma}{2}\mathbb{E}\left[\left\|\frac{1}{n}\sum_{i=1}^{n}\mathbb{E}_{t-\tau}\left[Q_t^i(\nabla f_i(x^{t-\tau}))\right]\right\|^2\right]$$

$$+\frac{\gamma}{2}\mathbb{E}\left[\left\|\nabla f(x^{t-\tau})-\frac{1}{n}\sum_{i=1}^{n}\mathbb{E}_{t-\tau}\left[Q_t^i(\nabla f_i(x^{t-\tau}))\right]\right\|^2\right]-\frac{\gamma}{2}\mathbb{E}\left[\left\|\nabla f(x^{t-\tau})\right\|^2\right] \quad (11)$$

$$\leq\frac{\gamma}{2}\varepsilon^2\frac{d^2}{m^2}\frac{1}{n}\sum_{i=1}^{n}\mathbb{E}\left[\left\|\nabla f_i(x^{t-\tau})\right\|^2\right]-\frac{\gamma}{2}\mathbb{E}\left[\left\|\nabla f(x^{t-\tau})\right\|^2\right]$$

$$\leq\gamma\left(\varepsilon^2\frac{d^2}{m^2}(\delta^2+1)-\frac{1}{2}\right)\mathbb{E}\left[\left\|\nabla f(x^{t-\tau})\right\|^2\right]+\gamma\varepsilon^2\frac{d^2}{m^2}\sigma^2$$

$$\leq-\frac{\gamma}{4}\mathbb{E}\left[\left\|\nabla f(x^{t-\tau})\right\|^2\right]+\gamma\varepsilon^2\frac{d^2}{m^2}\sigma^2.$$

The last inequality follows from the fact, that

$$\varepsilon\leq\frac{m}{2d\sqrt{\delta^2+1}}.$$

Consider ②. Using Cauchy-Schwarz A.1 and Fenchel-Young A.2 with $\beta=1$ inequalities we obtain

$$
\begin{aligned}
② &\leq \mathbb{E}\left[\left\|-\frac{\gamma}{n}\sum_{i=1}^{n}Q_t^i(\nabla f_i(x^t))\right\|\left\|\nabla f(x^t)-\nabla f(x^{t-\tau})\right\|\right] \\
&\leq \gamma L\mathbb{E}\left[\left\|\frac{1}{n}\sum_{i=1}^{n}Q_t^i(\nabla f_i(x^t))\right\|\left\|x^t-x^{t-\tau}\right\|\right] \\
&= \gamma^2 L\mathbb{E}\left[\left\|\frac{1}{n}\sum_{i=1}^{n}Q_t^i(\nabla f_i(x^t))\right\|\left\|\sum_{s=t-\tau}^{t-1}\frac{1}{n}\sum_{i=1}^{n}Q_s^i(\nabla f_i(x^s))\right\|\right] \\
&\leq \frac{\gamma^2 L}{2}\left(\tau\mathbb{E}\left[\left\|\frac{1}{n}\sum_{i=1}^{n}Q_t^i(\nabla f_i(x^t))\right\|^2\right]+\sum_{s=t-\tau}^{t-1}\mathbb{E}\left[\left\|\frac{1}{n}\sum_{i=1}^{n}Q_s^i(\nabla f_i(x^s))\right\|^2\right]\right).
\end{aligned}
\tag{12}
$$

Third equality holds since $x^t - x^{t-\tau} = \gamma\sum_{s=t-\tau}^{t-1}\frac{1}{n}\sum_{i=1}^{n}Q_s^i(\nabla f_i(x^s))$. Consider ③. Using Cauchy-Schwarz A.1 and Fenchel-Young A.2 with $\beta = m/d$ inequalities we obtain

$$
\begin{aligned}
③ &\leq \mathbb{E}\left[\left\|-\frac{\gamma}{n}\sum_{i=1}^{n}Q_t^i(\nabla f_i(x^t)-\nabla f_i(x^{t-\tau}))\right\|\left\|\nabla f(x^{t-\tau})\right\|\right] \\
&\leq \gamma L\mathbb{E}\left[\left\|\frac{1}{n}\sum_{i=1}^{n}Q_t^i(\nabla f_i(x^t)-\nabla f_i(x^{t-\tau}))\right\|\left\|\nabla f(x^{t-\tau})\right\|\right] \\
&\leq \gamma^2 L\frac{d}{m}\mathbb{E}\left[\left\|\frac{1}{n}\sum_{i=1}^{n}Q_t^i(\nabla f_i(x^t))\right\|\left\|\sum_{s=t-\tau}^{t-1}\frac{1}{n}\sum_{i=1}^{n}Q_s^i(\nabla f_i(x^s))\right\|\right] \\
&\leq \frac{\gamma^2 L}{2}\left(\sum_{s=t-\tau}^{t-1}\mathbb{E}\left[\left\|\frac{1}{n}\sum_{i=1}^{n}Q_s^i(\nabla f_i(x^s))\right\|^2\right]+\frac{d^2\tau}{m^2}\mathbb{E}\left[\left\|\nabla f(x^{t-\tau})\right\|^2\right]\right).
\end{aligned}
\tag{13}
$$

Wrapping (10) - (13) up we obtain

$$
\begin{aligned}
F_{t+1}-F_t &\leq \frac{\gamma^2 L}{2}\mathbb{E}\left[\left\|\frac{1}{n}\sum_{i=1}^{n}Q_t^i(\nabla f_i(x^t))\right\|^2\right]-\frac{\gamma}{4}\mathbb{E}\left[\left\|\nabla f(x^{t-\tau})\right\|^2\right]+\gamma\varepsilon^2\frac{d^2}{m^2}\sigma^2 \\
&\quad +\frac{\gamma^2 L}{2}\left(\tau\mathbb{E}\left[\left\|\frac{1}{n}\sum_{i=1}^{n}Q_t^i(\nabla f_i(x^t))\right\|^2\right]+\sum_{s=t-\tau}^{t-1}\mathbb{E}\left[\left\|\frac{1}{n}\sum_{i=1}^{n}Q_s^i(\nabla f_i(x^s))\right\|^2\right]\right) \\
&\quad +\frac{\gamma^2 L}{2}\left(\sum_{s=t-\tau}^{t-1}\mathbb{E}\left[\left\|\frac{1}{n}\sum_{i=1}^{n}Q_s^i(\nabla f_i(x^s))\right\|^2\right]+\frac{d^2\tau}{m^2}\mathbb{E}\left[\left\|\nabla f(x^{t-\tau})\right\|^2\right]\right) \\
&\leq \gamma^2 L\tau\mathbb{E}\left[\left\|\frac{1}{n}\sum_{i=1}^{n}Q_t^i(\nabla f_i(x^t))\right\|^2\right]+\gamma\varepsilon^2\frac{d^2}{m^2}\sigma^2 \\
&\quad +\gamma^2 L\sum_{s=t-\tau}^{t-1}\mathbb{E}\left[\left\|\frac{1}{n}\sum_{i=1}^{n}Q_s^i(\nabla f_i(x^s))\right\|^2\right]+\left(\frac{\gamma^2 L\tau d^2}{2m^2}-\frac{\gamma}{4}\right)\mathbb{E}\left[\left\|\nabla f(x^{t-\tau})\right\|^2\right].
\end{aligned}
$$

Using Lemma 5 we obtain

$$F_{t+1} - F_t \leq \frac{2d^2\gamma^2 L\tau}{m^2}\left((\delta^2+1)\mathbb{E}\left[\left\|\nabla f(x^t)\right\|^2\right]+\sigma^2\right)+\left(\frac{\gamma^2 L\tau d^2}{2m^2}-\frac{\gamma}{4}\right)\mathbb{E}\left[\left\|\nabla f(x^{t-\tau})\right\|^2\right]$$

$$+\frac{2d^2\gamma^2 L}{m^2}\sum_{s=t-\tau}^{t-1}\left((\delta^2+1)\mathbb{E}\left[\left\|\nabla f(x^s)\right\|^2\right]+\sigma^2\right)+\gamma\varepsilon^2\frac{d^2}{m^2}\sigma^2$$

$$=\frac{2d^2\gamma^2 L(\delta^2+1)\tau}{m^2}\mathbb{E}\left[\left\|\nabla f(x^t)\right\|^2\right]+\frac{2d^2\gamma^2 L(\delta^2+1)}{m^2}\sum_{s=t-\tau}^{t-1}\mathbb{E}\left[\left\|\nabla f(x^s)\right\|^2\right]$$

$$+\left(\frac{\gamma^2 L\tau d^2}{2m^2}-\frac{\gamma}{4}\right)\mathbb{E}\left[\left\|\nabla f(x^{t-\tau})\right\|^2\right]+\frac{\gamma d^2}{m^2}\left(4\gamma L\tau+\varepsilon^2\right)\sigma^2.$$

$$(14)$$

Summing (14) from $t=\tau$ to $t=T$ and using the fact that $\varepsilon^2 \leq \gamma L\tau$ and $1+\delta^2 \geq 1$ we obtain

$$\sum_{t=\tau}^{T}\frac{\gamma}{4}\mathbb{E}\left[\left\|\nabla f(x^{t-\tau})\right\|^2\right]\leq F_\tau+\frac{2d^2\gamma^2 L(\delta^2+1)}{m^2}\left(\tau\sum_{t=\tau}^{T}\mathbb{E}\left[\left\|\nabla f(x^t)\right\|^2\right]\right.$$

$$\left.+\sum_{t=\tau}^{T}\sum_{s=t-\tau}^{t-1}\mathbb{E}\left[\left\|\nabla f(x^s)\right\|^2\right]+\tau\sum_{t=\tau}^{T}\mathbb{E}\left[\left\|\nabla f(x^{t-\tau})\right\|^2\right]\right)+\sum_{t=\tau}^{T}5\frac{\gamma^2 L\tau d^2}{m^2}\sigma^2.$$

Since $\sum_{t=\tau}^{T}\sum_{s=t-\tau}^{t-1}\mathbb{E}\left[\left\|\nabla f(x^s)\right\|^2\right]\leq\tau\sum_{t=0}^{T}\mathbb{E}\left[\left\|\nabla f(x^t)\right\|^2\right]$, we get

$$\gamma\sum_{t=0}^{T-\tau}\mathbb{E}\left[\left\|\nabla f(x^t)\right\|^2\right]\leq 4F_\tau+\frac{24d^2\gamma^2 L(\delta^2+1)\tau}{m^2}\sum_{t=0}^{T}\mathbb{E}\left[\left\|\nabla f(x^t)\right\|^2\right]+20\sum_{t=\tau}^{T}\frac{\gamma^2 L\tau d^2}{m^2}\sigma^2.$$

Taking

$$\gamma\leq\frac{m^2}{48d^2 L(\delta^2+1)\tau},$$

we obtain

$$\gamma\sum_{t=0}^{T-\tau}\mathbb{E}\left[\left\|\nabla f(x^t)\right\|^2\right]\leq 8F_\tau+\frac{48d^2\gamma^2 L(\delta^2+1)\tau}{m^2}\sum_{t=T-\tau}^{T}\mathbb{E}\left[\left\|\nabla f(x^t)\right\|^2\right]+40\sum_{t=\tau}^{T}\frac{\gamma^2 L\tau d^2}{m^2}\sigma^2.$$

$$(15)$$

We now prove that for any $t \geq 0$, we have

$$\sup_{t\leq s\leq t+\tau}\left\{\mathbb{E}\left[\left\|\nabla f(x^s)\right\|^2\right]\right\}\leq 4\mathbb{E}\left[\left\|\nabla f(x^t)\right\|^2\right]+8L^2\gamma^2\tau^2\frac{d^2}{m^2}\sigma^2.$$

For $t \leq s \leq t+\tau$ it holds that

$$\mathbb{E}\left[\|\nabla f(x^s)\|^2\right] \le 2\mathbb{E}\left[\|\nabla f(x^t)\|^2\right] + \mathbb{E}\left[\|\nabla f(x^s) - \nabla f(x^t)\|^2\right]$$

$$\le 2\mathbb{E}\left[\|\nabla f(x^t)\|^2\right] + 2L^2\gamma^2\mathbb{E}\left[\left\|\sum_{r=t}^{s-1}\frac{1}{n}\sum_{i=1}^n Q_r^i(\nabla f_i(x^r))\right\|^2\right]$$

$$\le 2\mathbb{E}\left[\|\nabla f(x^t)\|^2\right] + 2L^2\gamma^2\tau\frac{d^2}{m^2}\sum_{r=t}^{s-1}\frac{1}{n}\sum_{i=1}^n \mathbb{E}\left[\|\nabla f_i(x^r)\|^2\right]$$

$$\le 2\mathbb{E}\left[\|\nabla f(x^t)\|^2\right] + 4L^2\gamma^2\tau\frac{d^2}{m^2}\sum_{r=t}^{s-1}\left((\delta^2+1)\mathbb{E}\left[\|\nabla f(x^r)\|^2\right] + \sigma^2\right)$$

$$\le 2\mathbb{E}\left[\|\nabla f(x^t)\|^2\right] + 4L^2\gamma^2\tau^2\frac{d^2}{m^2}\left((\delta^2+1)\sup_{t\le s\le t+\tau}\left\{\mathbb{E}\left[\|\nabla f(x^s)\|^2\right]\right\} + \sigma^2\right).$$

Since
$$\gamma \le \frac{m}{\sqrt{8}dL\sqrt{\delta^2+1}\tau},$$

it holds that

$$\sup_{t\le s\le t+\tau}\left\{\mathbb{E}\left[\|\nabla f(x^s)\|^2\right]\right\} \le 4\mathbb{E}\left[\|\nabla f(x^t)\|^2\right] + 8L^2\gamma^2\tau^2\frac{d^2}{m^2}\sigma^2.$$

Using this (15) takes form

$$\gamma\sum_{t=0}^{T-\tau}\mathbb{E}\left[\|\nabla f(x^t)\|^2\right] \le 8F_\tau + \frac{192d^2\gamma^2 L(\delta^2+1)\tau}{m^2}\sum_{t=T-2\tau}^{T-\tau}\mathbb{E}\left[\|\nabla f(x^t)\|^2\right]$$

$$+ 384L^3\gamma^4\tau^3\frac{d^4}{m^4}(\delta^2+1)\sigma^2 + 40\sum_{t=\tau}^T\frac{\gamma^2 L\tau d^2}{m^2}\sigma^2.$$

Taking
$$\gamma \le \frac{m}{384dL\sqrt{\delta^2+1}\tau},$$

and dividing both sides of the inequality by $T - \tau$, we obtain

$$\frac{1}{T-\tau}\sum_{t=0}^{T-\tau}\mathbb{E}\left[\|\nabla f(x^t)\|^2\right] \le 16\frac{F_\tau}{\gamma(T-\tau)} + 80\frac{\gamma^2 L\tau d^2}{m^2}\sigma^2.$$

Therefore, if $\widehat{x}^T$ is chosen uniformly from $\{x^t\}_{t=0}^{T-1}$, then it holds that

$$\mathbb{E}\left[\|\nabla f(\widehat{x}^T)\|^2\right] \le 16\frac{F_\tau}{\gamma T} + 80\frac{\gamma^2 L\tau d^2}{m^2}\sigma^2.$$

This finishes the proof. $\qquad\square$

### E.4 PROOF OF THEOREM 2, UNDER PL-CONDITION

*Proof.* We start from (14):

$$F_{t+1} - F_t = \frac{2d^2\gamma^2 L(\delta^2+1)\tau}{m^2}\mathbb{E}\left[\left\|\nabla f(x^t)\right\|^2\right] + \frac{2d^2\gamma^2 L(\delta^2+1)}{m^2}\sum_{s=t-\tau}^{t-1}\mathbb{E}\left[\left\|\nabla f(x^s)\right\|^2\right]$$

$$+ \left(\frac{\gamma^2 L\tau d^2}{2m^2} - \frac{\gamma}{4}\right)\mathbb{E}\left[\left\|\nabla f(x^{t-\tau})\right\|^2\right] + \frac{\gamma d^2}{m^2}\left(4\gamma L\tau + \varepsilon^2\right)\sigma^2.$$

If $f$ satisfies PL-inequality (Assumption 3), then $-\mathbb{E}\left[\left\|\nabla f(x^{t-\tau})\right\|^2\right] \leq -2\mu F_{t-\tau}$, so that, for some $0 < \alpha < 1$ we obtain

$$F_{t+1} - F_t = \frac{2d^2\gamma^2 L(\delta^2+1)\tau}{m^2}\mathbb{E}\left[\left\|\nabla f(x^t)\right\|^2\right] + \frac{2d^2\gamma^2 L(\delta^2+1)}{m^2}\sum_{s=t-\tau}^{t-1}\mathbb{E}\left[\left\|\nabla f(x^s)\right\|^2\right]$$

$$+ \left(\frac{\gamma^2 L\tau d^2}{2m^2} - \frac{(1-\alpha)\gamma}{4}\right)\mathbb{E}\left[\left\|\nabla f(x^{t-\tau})\right\|^2\right] \qquad (16)$$

$$- \frac{\alpha\gamma\mu}{2}F_{t-\tau} + \frac{\gamma d^2}{m^2}\left(4\gamma L\tau + \varepsilon^2\right)\sigma^2.$$

For $t \geq 0$, let $p_t = p^t$ and $p = (1 - \alpha\mu\gamma/4)^{-1}$. We multiply the above expression by $p_t$ and sum for $t < T$, hoping for cancellations. Using PL-condition (Assumption 3), for $T \geq \tau$ we obtain

$$\sum_{t=\tau}^{T-1} p_{t+1}\left(F_t - F_{t+1} - \frac{\alpha\gamma\mu}{4}F_{t-\tau}\right) = \sum_{t=\tau}^{T-1} p_{t+1}\left[\left(1 - \frac{\alpha\gamma\mu}{4}\right)F_t - F_{t+1} + \frac{\alpha\gamma\mu}{4}(F_t - F_{t-\tau})\right]$$

$$= \sum_{t=\tau}^{T-1} p_t F_t - \sum_{t=\tau+1}^{T} p_t F_t + \frac{\alpha\gamma\mu}{4}\sum_{t=\tau}^{T-1} p_{t+1}(F_t - F_{t-\tau})$$

$$\leq p_\tau F_\tau - p_T F_T + \frac{\alpha\gamma\mu}{4}\sum_{t=\tau}^{T-1} p_{t+1}F_t$$

$$- \frac{\alpha\gamma\mu p_\tau}{4}\sum_{t=0}^{T-1-\tau} p_{t+1}F_t$$

$$\leq p_\tau F_\tau - p_T F_T + \frac{\alpha\gamma\mu}{4}\sum_{t=T-\tau}^{T-1} p_{t+1}F_t$$

$$\leq p_\tau F_\tau - p_T F_T + \frac{\alpha\gamma}{8}\sum_{t=T-\tau}^{T-1} p_{t+1}\mathbb{E}\left[\left\|\nabla f(x^t)\right\|^2\right].$$

For any $t \geq 0$ we use a notation $b_t := \mathbb{E}\left[\left\|\nabla f(x^t)\right\|^2\right]$. We now handle $b_t$ terms from (16).

$$-\sum_{t=\tau}^{T-1}\frac{(1-\alpha)\gamma}{4}p_{t+1}b_{t-\tau} + \gamma^2 L\frac{d^2}{m^2}\sum_{t=\tau}^{T-1} p_{t+1}\left(2\tau(\delta^2+1)b_t + 2(\delta^2+1)\sum_{s=t-\tau}^{t-1} b_s + \frac{\tau}{2}b_{t-\tau}\right).$$
$$(17)$$

If $p_t = p^t$, $p = (1 - \alpha\mu\gamma/2)^{-1}$ and $\gamma = \gamma_1/\tau$, then, using the fact that $(1 - a/x)^{-x} \leq 2e^a \leq 2e$ if $x \geq 2$ and $0 \leq a \leq 1$, we can get that $1 \geq p_\tau = (1 - \mu\gamma_1/(2\tau))^{-\tau} \leq 2e^{\mu\gamma_1/2} \leq 2e \leq 6$. Then

$$\sum_{t=\tau}^{T} p_{t+1}\sum_{s=t-\tau}^{t-1} b_s \leq p^\tau\sum_{t=\tau}^{T}\sum_{s=t-\tau}^{t-1} p_{s+1}b_s \leq 6\tau\sum_{t=0}^{T} p_{t+1}b_t.$$

Now we can estimate (17):

$$(17) \leq - \sum_{t=0}^{T-\tau-1} \frac{(1-\alpha)\gamma}{4} p_{t+1} b_t + \gamma^2 L \frac{d^2 \tau}{m^2} \left( 2(\delta^2 + 1) \sum_{t=\tau}^{T-1} b_t + 12(\delta^2 + 1) \sum_{t=0}^{T-1} b_t + 3 \sum_{t=0}^{T-\tau} b_t \right)$$

$$\leq - \sum_{t=0}^{T-\tau-1} p_{t+1} \gamma b_t \left( \frac{1-\alpha}{4} - 17\gamma L \frac{d^2 \tau (\delta^2 + 1)}{m^2} \right) + 14\gamma^2 L \frac{d^2 \tau (\delta^2 + 1)}{m^2} \sum_{t=T-\tau}^{T-1} p_{t+1} b_t.$$

Taking

$$\gamma \leq \frac{m^2 (1-\alpha)}{136 L d^2 \tau (\delta^2 + 1) \beta},$$

where $\beta \geq 1$, we obtain

$$(17) \leq - \frac{(1-\alpha)\gamma}{8} \sum_{t=0}^{T-\tau-1} p_{t+1} b_t + \frac{(1-\alpha)\gamma}{4\beta} \sum_{t=T-\tau}^{T-1} p_{t+1} b_t.$$

Now we can estimate (16):

$$0 \leq p_\tau F_\tau - p_T F_T + \left( \frac{\alpha\gamma}{8} + \frac{(1-\alpha)\gamma}{4\beta} \right) \sum_{t=T-\tau}^{T-1} p_{t+1} b_t - \frac{(1-\alpha)\gamma}{8} \sum_{t=0}^{T-\tau-1} p_{t+1} b_t$$

$$+ \sum_{t=\tau}^{T-1} p_{t+1} \frac{\gamma d^2}{m^2} \left( 4\gamma L \tau + \varepsilon^2 \right) \sigma^2. \tag{18}$$

Using that we proved in E.3 we have $b_t \leq 4 b_{t-\tau} + 8 L^2 \gamma^2 \tau^2 \frac{d^2}{m^2} \sigma^2$. Then, we can obtain

$$\gamma \left( \frac{\alpha}{8} + \frac{1-\alpha}{4\beta} \right) \sum_{t=T-\tau}^{T-1} p_{t+1} b_t \leq 24\gamma \left( \frac{\alpha}{8} + \frac{1-\alpha}{4\beta} \right) \sum_{t=T-2\tau}^{T-\tau-1} p_{t+1} b_t$$

$$+ 48 L^2 \gamma^3 \tau^3 \frac{d^2}{m^2} \left( \frac{\alpha}{8} + \frac{1-\alpha}{4\beta} \right) \sigma^2.$$

Taking $\alpha = 1/6$ and $\beta = 4$, we obtain

$$\frac{\alpha}{8} + \frac{1-\alpha}{4\beta} = \frac{1-\alpha}{8},$$

and (18) takes form

$$0 \leq p_\tau F_\tau - p_T F_T + 48 L^2 \gamma^3 \tau^3 \frac{d^2}{m^2} \sigma^2 + \sum_{t=\tau}^{T-1} p_{t+1} \frac{\gamma d^2}{m^2} \left( 4\gamma L \tau + \varepsilon^2 \right) \sigma^2. \tag{19}$$

Using the fact that

$$\sum_{t=\tau}^{T} \left( 1 - \frac{\alpha\mu\gamma}{2} \right)^{T-t} = \sum_{t=0}^{T-\tau} \left( 1 - \frac{\alpha\mu\gamma}{2} \right)^t \leq \sum_{t=0}^{+\infty} \left( 1 - \frac{\alpha\mu\gamma}{2} \right)^t = \frac{2}{\alpha\mu\gamma},$$

and taking

$$\gamma \leq \frac{m^2}{625 L d^2 \tau (\delta^2 + 1)} \quad \text{and} \quad \varepsilon = \sqrt{\gamma L \tau} \leq \frac{m}{25 d \sqrt{\delta^2 + 1}},$$

by dividing (19) by $p_\tau$, we obtain

$$\mathbb{E}\left[f(x^T) - f(x^*)\right] \leq \left(1 - \frac{\mu\gamma}{12}\right)^{T-\tau} \mathbb{E}\left[f(x^\tau) - f(x^*)\right] + 636 \frac{\gamma d^2 L \tau}{\mu m^2} \sigma^2.$$

This finishes the proof.

$\square$

## F    CONVERGENCE OF ALGORITHM 1 WITHOUT DATA SIMILARITY

**Theorem 5** (Convergence of GD Algorithm 1 without data similarity). *Consider Assumptions 1 and 2. Let problem* (1) *be solved by Algorithm 1. Then for any $\varepsilon > 0$, $\gamma > 0$, $\tau > \tau_{mix}(\varepsilon)$ and $T > \tau$ satisfying*

$$\gamma \leq \frac{m^2 \sqrt{\mu}}{24 d^2 L^{3/2} \tau} \quad \text{and} \quad \varepsilon \leq \frac{m\sqrt{\mu}}{24 d} \min\left\{\frac{1}{L^{3/2}}; \sqrt{\mu}\right\},$$

*it holds that*

$$\mathbb{E}\left[\left\|x^{T+1} - x^*\right\|^2\right] \leq \left(1 - \frac{\mu\gamma}{2}\right)^{T-\tau} \mathbb{E}\left[\left\|x^\tau - x^*\right\|^2\right] + \left(1 - \frac{\mu\gamma}{2}\right)^T \Delta_\tau + 26 \frac{\gamma d^2 \tau}{\mu m^2} \sigma_*^2,$$

*where*

$$\Delta_\tau = \mathcal{O}\left(\frac{\gamma^2 d^2}{m^2} \sqrt{\frac{\mu}{L}} \sum_{t=0}^{\tau} \left[\tau \mathbb{E}\left[\left\|x^t - x^*\right\|^2\right] + 4L\mathbb{E}\left[f(x^t) - f(x^*)\right]\right]\right).$$

*Proof of Theorem 5.* We start by writing out step of the Algorithm 1:

$$\mathbb{E}\left[\left\|x^{t+1} - x^*\right\|^2\right] = \mathbb{E}\left[\left\|x^t - x^*\right\|^2\right] - 2\gamma\mathbb{E}\left[\frac{1}{n}\sum_{i=1}^{d}\left\langle Q_t^i\left(\nabla f_i(x^t)\right) - \nabla f_i(x^t), x^t - x^*\right\rangle\right]$$

$$- 2\gamma\mathbb{E}\left[\left\langle \nabla f(x^t), x^t - x^*\right\rangle\right] + \gamma^2\mathbb{E}\left[\left\|\frac{1}{n}\sum_{i=1}^{n} Q_t^i\left(\nabla f_i(x^t)\right)\right\|^2\right]. \tag{20}$$

Consider $\mathbb{E}\left[\left\langle Q_t^i\left(\nabla f_i(x^t)\right) - \nabla f_i(x^t), x^t - x^*\right\rangle\right]$. Using Corollary 3 with $a^t = \nabla f_i(x^t)$, $b^t = x^t - x^*$, $\hat{a}^{t-\tau} = \nabla f_i(x^{t-\tau})$ and $\hat{b}^{t-tau} = x^{t-\tau} - x^*$ we obtain

$$2\mathbb{E}\left[\frac{1}{n}\sum_{i=1}^{n}\left|\left\langle Q_t^i\left(\nabla f_i(x^t)\right) - \nabla f_i(x^t), x^t - x^*\right\rangle\right|\right] \leq \frac{\varepsilon d}{m\beta_0}\frac{1}{n}\sum_{i=1}^{n}\mathbb{E}\left[\left\|\nabla f_i(x^{t-\tau})\right\|^2\right]$$

$$+ \frac{\varepsilon d\beta_0}{m}\mathbb{E}\left[\left\|x^{t-\tau} - x^*\right\|^2\right] + 4\frac{d^2 L^2}{m^2}(\beta_1 + \beta_2)\mathbb{E}\left[\left\|x^t - x^\tau\right\|^2\right] + \left(\frac{1}{\beta_1} + \frac{1}{\beta_3}\right)\mathbb{E}\left[\left\|x^t - x^\tau\right\|^2\right]$$

$$+ 4\frac{d^2}{m^2}\beta_3\frac{1}{n}\sum_{i=1}^{d}\mathbb{E}\left[\left\|\nabla f_i(x^t)\right\|^2\right] + \frac{1}{\beta_2}\mathbb{E}\left[\left\|x^t - x^*\right\|^2\right]. \tag{21}$$

Using the fact that $f_i$ are $L$-smooth, we can obtain:

$$
\begin{aligned}
\frac{1}{n}\sum_{i=1}^{n}\left\|\nabla f_i(x^t)\right\|^2 &= \frac{1}{n}\sum_{i=1}^{n}\left\|\nabla f_i(x^t) - \nabla f_i(x^*) + \nabla f_i(x^*)\right\|^2 \\
&\leq \frac{2}{n}\sum_{i=1}^{n}\left\|\nabla f_i(x^t) - \nabla f_i(x^*)\right\|^2 + \frac{2}{n}\sum_{i=1}^{n}\left\|\nabla f_i(x^*)\right\|^2 \\
&\leq \frac{4L}{n}\sum_{i=1}^{n}\left(f_i(x^t) - f_i(x^*) - \left\langle \nabla f_i(x^*), x^t - x^*\right\rangle\right) + 2\sigma_*^2 \\
&= 4L(f(x^t) - f(x^*)) + 2\sigma_*^2,
\end{aligned}
\tag{22}
$$

where we use a notation $\sigma_*^2 := \frac{1}{n}\sum_{i=1}^{n}\left\|\nabla f_i(x^*)\right\|^2$. Now we can estimate (21):

$$
\begin{aligned}
(21) \leq &\ \frac{2\varepsilon d}{m\beta_0}\left(2L\mathbb{E}\left[f(x^{t-\tau}) - f(x^*)\right] + \sigma_*^2\right) + \frac{\varepsilon d\beta_0}{m}\mathbb{E}\left[\left\|x^{t-\tau} - x^*\right\|^2\right] \\
&+ \left(4\frac{d^2 L^2}{m^2}(\beta_1 + \beta_2) + \frac{1}{\beta_1} + \frac{1}{\beta_3}\right)\mathbb{E}\left[\left\|-\gamma\sum_{s=t-\tau}^{t-1}\frac{1}{n}\sum_{i=1}^{n}Q_s^i\left(\nabla f_i(x^s)\right)\right\|^2\right] \\
&+ 8\frac{d^2}{m^2}\beta_3\left(2L\mathbb{E}\left[f(x^t) - f(x^*)\right] + \sigma_*^2\right) + \frac{1}{\beta_2}\mathbb{E}\left[\left\|x^t - x^*\right\|^2\right].
\end{aligned}
$$

Now we can estimate (20). Using Lemma 4 and Assumption 2 we can obtain

$$
\begin{aligned}
\mathbb{E}\left[\left\|x^{t+1} - x^*\right\|^2\right] \leq &\left(1 - \mu\gamma + \frac{\gamma}{\beta_2}\right)\mathbb{E}\left[\left\|x^t - x^*\right\|^2\right] + \frac{\varepsilon d\beta_0\gamma}{m}\mathbb{E}\left[\left\|x^{t-\tau} - x^*\right\|^2\right] \\
&+ 4L\mathbb{E}\left[\frac{\varepsilon d\gamma}{m\beta_0}(f(x^{t-\tau}) - f(x^*)) + 4\frac{d^2\beta_3\gamma}{m^2}(f(x^t) - f(x^*))\right. \\
&\left.+ \left(4\frac{d^2 L^2}{m^2}(\beta_1 + \beta_2) + \frac{1}{\beta_1} + \frac{1}{\beta_3}\right)\frac{\gamma^3\tau d^2}{m^2}\sum_{s=t-\tau}^{t-1}(f(x^s) - f(x^*))\right. \\
&\left.+ \frac{\gamma^2 d^2}{m^2}(f(x^t) - f(x^*)) - \frac{\gamma}{2L}(f(x^t) - f(x^*))\right] \\
&+ 2\left[\frac{\varepsilon d\gamma}{m\beta_0} + 4\frac{d^2\beta_3\gamma}{m^2} + \left(4\frac{d^2 L^2}{m^2}(\beta_1 + \beta_2) + \frac{1}{\beta_1} + \frac{1}{\beta_3}\right)\frac{\gamma^3\tau^2 d^2}{m^2} + \frac{\gamma^2 d^2}{m^2}\right]\sigma_*^2.
\end{aligned}
\tag{23}
$$

Taking $\beta_0 = \beta_1 = 1, \beta_3 = \gamma, \beta_2 = 4/\mu$ and using fact, that $\varepsilon \leq \gamma\tau d/m$ inequality (23) takes form

$$
\begin{aligned}
\mathbb{E}\left[\left\|x^{t+1} - x^*\right\|^2\right] \leq &\left(1 - \frac{3}{4\mu\gamma}\right)\mathbb{E}\left[\left\|x^t - x^*\right\|^2\right] + \frac{\varepsilon d\beta_0\gamma}{m}\mathbb{E}\left[\left\|x^{t-\tau} - x^*\right\|^2\right] \\
&+ 4L\mathbb{E}\left[\frac{\varepsilon d\gamma}{m\beta_0}(f(x^{t-\tau}) - f(x^*)) + 5\frac{d^2\gamma^2}{m^2}(f(x^t) - f(x^*))\right. \\
&\left.+ 20\frac{d^4 L^2}{m^4}\frac{\gamma^3\tau}{\mu}\sum_{s=t-\tau}^{t-1}(f(x^s) - f(x^*)) - \frac{\gamma}{2L}(f(x^t) - f(x^*))\right] \\
&+ 4\frac{d^2\gamma^2\tau}{m^2}\left[3 + 10\frac{d^2 L^2}{m^2}\frac{\gamma}{\mu}\right]\sigma_*^2.
\end{aligned}
\tag{24}
$$

Let us perform the summation from $t = \tau$ to $t = T > \tau$ of equations (24) with coefficients $p_k$:

$$\sum_{t=\tau}^{T} p_t \mathbb{E}\left[\left\|x^{t+1} - x^*\right\|^2\right] \le \sum_{t=\tau}^{T} p_t (1 - \frac{3\mu\gamma}{4}) \mathbb{E}\left[\left\|x^t - x^*\right\|^2\right]$$

$$+ \sum_{t=\tau}^{T} p_t \frac{\gamma\varepsilon d}{m} \mathbb{E}\left[\left\|x^{t-\tau} - x^*\right\|^2\right]$$

$$+ \sum_{t=\tau}^{T} p_t 4L \left(\frac{\gamma\varepsilon d}{m} + 5\frac{\gamma^2 d^2 \tau}{m^2} - \frac{\gamma}{2L}\right) \mathbb{E}\left[f(x^t) - f(x^*)\right] \quad (25)$$

$$+ 20 \sum_{t=\tau}^{T} p_t 4L \frac{d^4 L^2}{m^4} \frac{\gamma^3 \tau}{\mu} \sum_{s=t-\tau}^{t-1} \mathbb{E}\left[f(x^s) - f(x^*)\right]$$

$$+ \sum_{t=\tau}^{T} p_t 4 \frac{d^2 \gamma^2 \tau}{m^2} \left[3 + 10\frac{d^2 L^2}{m^2} \frac{\gamma}{\mu}\right] \sigma_*^2.$$

If $p_t = p^t$, $p = (1 - \mu\gamma/2)^{-1}$ and $\gamma = \gamma_1/\tau$, then, using the fact that $(1 - a/x)^{-x} \le 2e^a \le 2e$ if $x \ge 2$ and $0 \le a \le 1$, we can get that $p_\tau = (1 - \mu\gamma_1/(2\tau))^{-\tau} \le 2e^{\mu\gamma_1/2} \le 2e \le 6$.

$$\sum_{t=\tau}^{T} p_t \sum_{s=t-\tau}^{t-1} a_s \le p^\tau \sum_{t=\tau}^{T} \sum_{s=t-\tau}^{t-1} p_s a_s \le 6\tau \sum_{t=0}^{T} p_t a_t.$$

Using this we can estimate (25):

$$\sum_{t=\tau}^{T} p_t \mathbb{E}\left[\left\|x^{t+1} - x^*\right\|^2\right] \le \sum_{t=\tau}^{T} p_t \left(1 - \mu\gamma + 6\frac{\gamma\varepsilon d}{m}\right) \mathbb{E}\left[\left\|x^t - x^*\right\|^2\right]$$

$$+ \sum_{t=\tau}^{T} 4p_t L \left(\frac{\gamma\varepsilon d}{m} + 5\frac{\gamma^2 d^2 \tau}{m^2} + 120\frac{d^4 L^2}{m^4} \frac{\gamma^3 \tau^2}{\mu} - \frac{\gamma}{2L}\right) \mathbb{E}\left[f(x^t) - f(x^*)\right]$$

$$+ 4 \sum_{t=\tau}^{T} p_t \left[3 + 10\frac{d^2 L^2}{m^2} \frac{\gamma}{\mu}\right] \sigma_*^2 + \sum_{t=0}^{\tau} p_{t+\tau} \frac{\gamma\varepsilon d}{m} \mathbb{E}\left[\left\|x^t - x^*\right\|^2\right] \quad (26)$$

$$+ 80 \sum_{t=0}^{\tau} p_{t+\tau} L \frac{d^4 L^2}{m^4} \frac{\gamma^3 \tau}{\mu} \mathbb{E}\left[f(x^t) - f(x^*)\right].$$

Taking

$$\gamma \le \frac{m^2 \sqrt{\mu}}{24 d^2 L^{3/2} \tau} \quad \text{and} \quad \varepsilon = \min\left\{\frac{\gamma d\tau}{m}; \frac{\mu m}{24 d}\right\} \le \frac{m\sqrt{\mu}}{24 d} \min\left\{\frac{1}{L^{3/2}}; \sqrt{\mu}\right\}.$$

We get

$$\frac{\gamma\varepsilon d}{m} + 5\frac{\gamma^2 d^2 \tau}{m^2} + 120\frac{d^4 L^2}{m^4} \frac{\gamma^3 \tau^2}{\mu} - \frac{\gamma}{2L} \le 0 \quad \text{and} \quad 1 - \frac{3\mu\gamma}{4} + 6\frac{\gamma\varepsilon d}{m} = 1 - \frac{\mu\gamma}{2}.$$

Assume a notation

$$\Delta_\tau := \sum_{t=0}^{\tau} p_{t+\tau} \frac{\gamma\varepsilon d}{m} \mathbb{E}\left[\left\|x^t - x^*\right\|^2\right] + 80 \sum_{t=0}^{\tau} p_{t+\tau} L \frac{d^4 L^2}{m^4} \frac{\gamma^3 \tau}{\mu} \mathbb{E}\left[f(x^t) - f(x^*)\right]$$

$$\le 120 \frac{\gamma^2 d^2}{m^2} \sqrt{\frac{\mu}{L}} \sum_{t=0}^{\tau} \left(\tau \mathbb{E}\left[\left\|x^t - x^*\right\|^2\right] + 4L \mathbb{E}\left[f(x^t) - f(x^*)\right]\right).$$

Using the notation of $\Delta_\tau$, (26) takes form

$$\sum_{t=\tau}^{T} p_t \mathbb{E}\left[\left\|x^{t+1} - x^*\right\|^2\right] \le \sum_{t=\tau}^{T} p_t \left(1 - \frac{\mu\gamma}{2}\right) \mathbb{E}\left[\left\|x^t - x^*\right\|^2\right] + \sum_{t=\tau}^{T} 13 p_t \frac{\gamma^2 d^2 \tau}{m^2} \sigma_*^2 + \Delta_\tau.$$

Using $p_t = p^t$ and $p = (1 - \mu\gamma/2)^{-1}$ we can obtain:

$$\sum_{t=\tau}^{T} \left(1 - \frac{\mu\gamma}{2}\right)^{-t} \mathbb{E}\left[\left\|x^{t+1} - x^*\right\|^2\right] \le \sum_{t=\tau}^{T} \left(1 - \frac{\mu\gamma}{2}\right)^{-t+1} \mathbb{E}\left[\left\|x^t - x^*\right\|^2\right]$$
$$+ \sum_{t=\tau}^{T} 13 \left(1 - \frac{\mu\gamma}{2}\right)^{-t} \frac{\gamma^2 d^2 \tau}{m^2} \sigma_*^2 + \Delta_\tau.$$

The summed terms on the left and right sides are reduced, therefore this expression takes the form:

$$\left(1 - \frac{\mu\gamma}{2}\right)^{-T} \mathbb{E}\left[\left\|x^{T+1} - x^*\right\|^2\right] \le \left(1 - \frac{\mu\gamma}{2}\right)^{-\tau} \mathbb{E}\left[\left\|x^\tau - x^*\right\|^2\right]$$
$$+ \sum_{t=\tau}^{T} 13 \left(1 - \frac{\mu\gamma}{2}\right)^{-t} \frac{\gamma^2 d^2 \tau}{m^2} \sigma_*^2 + \Delta_\tau.$$

We can re-arrange this inequality:

$$\mathbb{E}\left[\left\|x^{T+1} - x^*\right\|^2\right] \le \left(1 - \frac{\mu\gamma}{2}\right)^{T-\tau} \mathbb{E}\left[\left\|x^\tau - x^*\right\|^2\right]$$
$$+ \sum_{t=\tau}^{T} 13 \left(1 - \frac{\mu\gamma}{2}\right)^{T-t} \frac{\gamma^2 d^2 \tau}{m^2} \sigma_*^2 + \left(1 - \frac{\mu\gamma}{2}\right)^{T} \Delta_\tau.$$

Using the fact that

$$\sum_{t=\tau}^{T} \left(1 - \frac{\mu\gamma}{2}\right)^{T-t} = \sum_{t=0}^{T-\tau} \left(1 - \frac{\mu\gamma}{2}\right)^{t} \le \sum_{t=0}^{+\infty} \left(1 - \frac{\mu\gamma}{2}\right)^{t} = \frac{2}{\mu\gamma}.$$

We can estimate:

$$\mathbb{E}\left[\left\|x^{T+1} - x^*\right\|^2\right] \le \left(1 - \frac{\mu\gamma}{2}\right)^{T-\tau} \mathbb{E}\left[\left\|x^\tau - x^*\right\|^2\right] + \left(1 - \frac{\mu\gamma}{2}\right)^{T} \Delta_\tau + 26 \frac{\gamma d^2 \tau}{\mu m^2} \sigma_*^2.$$

This finishes the proof.

$\square$

## G    EXTENSIONS FOR THEOREM 3

### G.1    FULL VERSION OF THEOREM 3

**Theorem 6** (Convergence of AMQSGD Algorithm 2, full version). *Consider Assumptions 1, 2 and 4. Let problem* (1) *be solved by Algorithm 2. Then for any* $\gamma > 0, \varepsilon > 0, \tau > \tau_{mix}(\varepsilon), T > \tau$ *and* $\beta, \theta, \eta, p$ *satisfying*

$$\gamma \lesssim \frac{\mu^{\frac{1}{3}} m^{\frac{1}{2}}}{\tau L^{\frac{4}{3}} d^{\frac{1}{2}}}, \quad p \lesssim \frac{m^2}{\tau^2 d^2 (\delta^2 + 1)}, \quad \varepsilon \lesssim \min \left\{ \frac{m^{\frac{7}{4}}}{d^{\frac{7}{4}} \tau^{\frac{5}{4}} L(\delta^2 + 1)}; \frac{m^{\frac{15}{4}}}{d^{\frac{15}{4}} \tau^{\frac{13}{4}} (\delta^2 + 1)^2} \right\}$$

$$\beta = \sqrt{\frac{2p^2 \mu \gamma}{3}}, \quad \eta = \sqrt{\frac{3}{2\mu\gamma}}, \quad \theta = \frac{p\eta^{-1} - 1}{\beta p \eta^{-1} - 1}$$

*it holds that*

$$F_{T+1} = \mathcal{O} \left( \exp\left[ -(T - \tau) \sqrt{\frac{p^2 \mu \gamma}{3}} \right] F_\tau + \exp\left[ -T \sqrt{\frac{p^2 \mu \gamma}{3}} \right] \Delta_\tau + \frac{\gamma}{\mu} \sigma^2 \right).$$

*Here we use notations:* $F_t := \mathbb{E}[\|x^t - x^*\|^2 + \frac{3}{\mu}(f(x_f^t) - f(x^*))]$ *and* $\Delta_\tau \leq \frac{\sqrt{\gamma}}{\tau^{\frac{4}{3}} \mu^{\frac{1}{3}}} \sum\limits_{t=0}^{\tau} \left( \mathbb{E}\left\| \nabla f(x_g^t) \right\| + \mathbb{E}\left\| x^t - x^* \right\|^2 + \mathbb{E}[f(x_f^t) - f(x^*)] \right).$

### G.2 FULL VERSION OF COROLLARY 2

**Corollary 5** (Step tuning for Theorem 3, full version of Corollary 2). *Under the conditions of Theorem 3, choosing $\gamma$ as*

$$\gamma \lesssim \min \left\{ \frac{\mu^{\frac{1}{3}}}{L^{\frac{4}{3}} \tau^{\frac{8}{3}}}; \frac{\log\left( \max\left\{ 2; \frac{\mu^{\frac{2}{3}} (F_\tau + \Delta_\tau) T}{\tau^{\frac{4}{3}} L^{\frac{2}{3}} \sigma^2} \right\} \right)}{\mu p^2 T^2} \right\},$$

*in order to achieve $\epsilon$-approximate solution (in terms of $\mathbb{E}\left[ \left\| x^T - x^* \right\|^2 \right] \leq \epsilon^2$) it takes*

$$\mathcal{O} \left( \frac{d^2 L^{\frac{2}{3}} \tau^{\frac{4}{3}}}{m^2 \mu^{\frac{2}{3}}} \left( (\delta^2 + 1) \log\left( \frac{1}{\epsilon} \right) + \frac{\sigma^2}{\mu \epsilon} \right) \right) \text{ iterations.}$$

### G.3 PROOF OF THEOREM 6

**Lemma 6.** *Consider Algorithm 2 with $\theta = (p\eta^{-1} - 1)/(\beta\eta^{-1} 1 - 1) < 1$. Then for any $y^t = \kappa x_f^t + (1 - \kappa)x^t \in \text{conv}\left\{ x_f^t, x^t \right\}$ for any $s < t$ exist constants $\alpha_f^s, \alpha^s \geq 0$ and $c_r \geq 0$ such that*

$$y^t = \widetilde{y}^s - p\gamma \sum_{r=s}^{t-1} c_r g^r = \alpha_f^s x_f^s + \alpha^s x^s - p\gamma \sum_{r=s}^{t-1} c_r g^r.$$

*And $\alpha_f^s + \alpha^s = 1$ for any $s < t$. If $(1 - \kappa)\eta \leq 1$, then $c_r \leq t - s + 2$, otherwise we can only use the estimate $c_r \leq \eta$.*

*Proof.* We start by writing out lines 3 and 10 of Algorithm 2:

$$x_f^s = x_g^{s-1} - p\gamma g^{s-1} = \theta x_f^{s-1} + (1 - \theta)x^{s-1} - p\gamma g^{s-1}. \tag{27}$$

Now let us handle expression $\eta x_g^k + (p - \eta)x_f^k + (1 - p)(1 - \beta)x^k + (1 - p)\beta x_g^k - x^*$ for a while. Taking into account the choice of $\theta$ such that $\theta = (p\eta^{-1} - 1)/(\beta p\eta^{-1} - 1)$ (in particular, $(p\eta^{-1} - 1) = (\beta p\eta^{-1} - 1)\theta$ and $\eta(1 - \beta p\eta^{-1})(1 - \theta) = p(1 - \beta)$), we get

$$\eta x_g^k + (p - \eta)x_f^k + (1 - p)(1 - \beta)x^k + (1 - p)\beta x_g^k$$

$$= (\eta + (1-p)\beta)x_g^k + (p - \eta)x_f^k + (1-p)(1-\beta)x^k$$

$$= (\eta + (1-p)\beta)x_g^k + \eta(p\eta^{-1} - 1)x_f^k + (1-p)(1-\beta)x^k$$

$$= (\eta + (1-p)\beta)x_g^k + \eta(\beta p\eta^{-1} - 1)\theta x_f^k + (1-p)(1-\beta)x^k$$

$$= (\eta + (1-p)\beta)x_g^k + \eta(\beta p\eta^{-1} - 1)(x_g^k - (1-\theta)x^k) + (1-p)(1-\beta)x^k$$

$$= (\eta + (1-p)\beta)x_g^k + \eta(\beta p\eta^{-1} - 1)(x_g^k - (1-\theta)x^k) + (1-p)(1-\beta)x^k$$

$$= \beta x_g^k - \eta(\beta p\eta^{-1} - 1)(1-\theta)x^k + (1-p)(1-\beta)x^k$$

$$= \beta x_g^k + p(1-\beta)x^k + (1-p)(1-\beta)x^k$$

$$= \beta x_g^k + (1-\beta)x^k\,.$$

Now we write out line 11 of Algorithm 2:

$$x^s = \beta x_g^{s-1} + (1-\beta)x^{s-1} - \eta x_g^{s-1} + \eta x_f^s = \beta x_g^{s-1} + (1-\beta)x^{s-1} - \eta p\gamma g^{s-1}$$

$$= \beta(\theta x_f^{s-1} + (1-\theta)x^{s-1}) + (1-\beta)x^{s-1} - \eta p\gamma g^{s-1} \tag{28}$$

$$= \beta\theta x_f^{s-1} + (1-\beta\theta)x^{s-1} - \eta p\gamma g^{s-1}.$$

Now we use induction. $x_f^t = \theta x_f^{s-1} + (1-\theta)x^{s-1} - p\gamma g^{s-1}$, then $\alpha_f^{t-1} = \theta \geq 0$, $\alpha^{t-1} = 1 - \theta \geq 0$, $c_r = 1 \leq \eta$ and $\alpha_f^{t-1} + \alpha^{t-1} = 1$, therefore base step is fulfilled. If $x_f^t = \alpha_f^s x_f^s + \alpha^s x^s - p\gamma \sum_{r=s}^{t-1} c_r g^r$ for some $s < t$, when with help of (27) and (28) we can write out

$$x_f^t = \alpha_f^s \left( \theta x_f^{s-1} + (1-\theta)x^{s-1} - p\gamma g^{s-1} \right)$$

$$+ \alpha^s \left( \beta\theta x_f^{s-1} + (1-\beta\theta)x^{s-1} - \eta p\gamma g^{s-1} \right) - p\gamma \sum_{r=s}^{t-1} c_r g^r.$$

Therefore $\alpha_f^{s-1} = \alpha_f^s \theta + \alpha^s \beta\theta \geq 0$, $\alpha^{s-1} = \alpha_f^s(1-\theta) + \alpha^s(1-\beta\theta) \geq 0$ and $c_{s-1} = \alpha_f^s + \eta\alpha^s \leq \eta$. Then, the step of the induction is fulfilled, since $\alpha_f^{s-1} + \alpha^{s-1} = 1$. Therefore results of this Lemma are true for $y^t = x_f^t \in \text{conv}\left\{ x_f^t, x^t \right\}$.

Consider $y^t = x^t \in \text{conv}\left\{ x_f^t, x^t \right\}$. Form (28) follows that $\alpha_f^{t-1} = \beta\theta$ and $\alpha^{t-1} = 1 - \beta\theta$, therefore base step is fulfilled. The step of the induction will be the same as in $y^t = x_f^t$. Therefore results of this Lemma are true for $y^t = x^t$. Then, they are true for any $y^t \in \text{conv}\left\{ x_f^t, x^t \right\}$.

If $y^t = \kappa x_f^t + (1-\kappa)x^t$, then $\alpha^s(y) = \kappa\alpha^s(x_f^t) + (1-\kappa)\alpha^s(x^t)$. Since $(1-\theta)\eta \leq 1$, then $\alpha^{t-1}(x_f^t)\eta \leq 1 = t - (t-1)$. Therefore $\alpha^s(x_f^t)\eta \leq t - s$ by induction, since $\alpha^{s-1}(x_f^t)\eta = \alpha_f^s(x_f^t)(1-\theta)\eta + (1-\beta\theta)\alpha^s(x_f^t)\eta \leq \alpha_f^s(x_f^t) + (1-\beta\theta)(t-s) \leq t - s + 1$.

Then, if $(1-\kappa)\eta \leq 1$, then $\alpha^s(y^t)\eta = \kappa\alpha^s(x_f^t)\eta + (1-\kappa)\eta\alpha^s(x^t) \leq \kappa(t-s) + \alpha^s(x^t) \leq t - s + 1$. Now we consider $c_s(y^t)$. $c_s(y^t) = \alpha_f^s(y^t) + \alpha^s(y^t)\eta \leq \alpha_f^s(y^t) + t - s + 1 \leq t - s + 2$.

$\square$

**Lemma 7.** *Assume 1, 2 and 4. Then for iterates of Algorithm 2 with $\theta = (p\eta^{-}1 - 1)/(\beta p\eta^{-}1 - 1), \theta > 0, \eta \geq 1$, it holds that*

$$\mathbb{E}\left\| x^{t+1} - x^* \right\|^2$$

$$\leq (1-\beta)(1 + \frac{\beta}{4})\mathbb{E}\left\| x^t - x^* \right\|^2 + \beta(1 + \frac{\beta}{4})\mathbb{E}\left\| x_g^t - x^* \right\|^2 + (\beta^2 - \beta)\mathbb{E}\left\| x^t - x_g^t \right\|^2$$

$$+ 10\frac{d^2}{m^2}(\delta^2 + 1)p^2\gamma^2\eta^2\mathbb{E}\left\| \nabla f(x_g^t) \right\|^2 + p^2\gamma^2\eta^2\tau\left( 32\frac{\tau^2 d^2 L^2 p^2\gamma^2}{m^2\beta} + \frac{5}{4} \right)\sum_{r=t-\tau}^{t-1}\left\| g^r \right\|^2$$

$$+ 3\varepsilon p\gamma\eta L\frac{d}{m}\sqrt{\delta^2+1}\mathbb{E}\left[\left\|x^{t-\tau}-x^*\right\|^2\right] + 3\varepsilon p\gamma\eta L\frac{d}{m}\sqrt{\delta^2+1}\mathbb{E}\left[\left\|x_f^{t-\tau}-x^*\right\|^2\right] \quad (29)$$

$$- 2\gamma\eta^2\,\mathbb{E}\left\langle\nabla f(x_g^t), x_g^t + (p\eta^{-1}-1)x_f^t - p\eta^{-1}x^*\right\rangle + 2p\gamma\eta\left(\frac{\varepsilon d}{m\sqrt{\delta^2+1}L} + 4p\gamma\eta\frac{d^2}{m^2}\right)\sigma^2.$$

*Proof.* Using lines 10 and 11 of Algorithm 2, we get

$$\mathbb{E}\left\|x^{t+1}-x^*\right\|^2 = \mathbb{E}\left\|\eta x_f^{t+1} + (p-\eta)x_f^t + (1-p)(1-\beta)x^t + (1-p)\beta x_g^t - x^*\right\|^2$$

$$= \mathbb{E}\left\|\eta x_g^t - p\gamma\eta g^t + (p-\eta)x_f^t + (1-p)(1-\beta)x^t + (1-p)\beta x_g^t - x^*\right\|^2$$

$$= \mathbb{E}\left\|\eta x_g^t + (p-\eta)x_f^t + (1-p)(1-\beta)x^t + (1-p)\beta x_g^t - x^*\right\|^2 + p^2\gamma^2\eta^2\,\mathbb{E}\left\|g^t\right\|^2$$

$$\quad - 2p\gamma\eta\,\mathbb{E}\left\langle g^t, \eta x_g^t + (p-\eta)x_f^t + (1-p)(1-\beta)x^t + (1-p)\beta x_g^t - x^*\right\rangle$$

$$= \underbrace{\mathbb{E}\left\|\eta x_g^t + (p-\eta)x_f^t + (1-p)(1-\beta)x^t + (1-p)\beta x_g^t - x^*\right\|^2}_{①} + \underbrace{p^2\gamma^2\eta^2\,\mathbb{E}\left\|g^t\right\|^2}_{②}$$

$$\underbrace{- 2p\gamma\eta\,\mathbb{E}\left\langle g^t - \nabla f(x_g^t), \eta x_g^t + (p-\eta)x_f^t + (1-p)(1-\beta)x^t + (1-p)\beta x_g^t - x^*\right\rangle}_{③}$$

$$\underbrace{- 2p\gamma\eta\,\mathbb{E}\left\langle \nabla f(x_g^t), \eta x_g^t + (p-\eta)x_f^t + (1-p)(1-\beta)x^t + (1-p)\beta x_g^t - x^*\right\rangle}_{④}.$$

Consider ①. From Lemma 6, we know that

$$\eta x_g^t + (p-\eta)x_f^t + (1-p)(1-\beta)x^t + (1-p)\beta x_g^t = \beta x_g^t + (1-\beta)x^t.$$

It implies

$$\left\|\eta x_g^t + (p-\eta)x_f^t + (1-p)(1-\beta)x^t + (1-p)\beta x_g^t - x^*\right\|^2$$

$$= \left\|\beta x_g^t + (1-\beta)x^t - x^*\right\|^2$$

$$= \left\|\beta(x_g^t - x^t) + x^t - x^*\right\|^2$$

$$= \left\|x^t - x^*\right\|^2 + 2\beta\left\langle x^t - x^*, x_g^t - x^t\right\rangle + \beta^2\left\|x_g^t - x^t\right\|^2 \quad (30)$$

$$= \left\|x^t - x^*\right\|^2 + \beta(\left\|x_g^t - x^*\right\|^2 - \left\|x^t - x^*\right\|^2 - \left\|x_g^t - x^t\right\|^2) + \beta^2\left\|x_g^t - x^t\right\|^2$$

$$= (1-\beta)\left\|x^t - x^*\right\|^2 + \beta\left\|x_g^t - x^*\right\|^2 + (\beta^2 - \beta)\left\|x^t - x_g^t\right\|^2.$$

Consider ②. Using convexity of squared Euclidean norm and Lemma 4, one can obtain

$$p^2\gamma^2\eta^2\,\mathbb{E}\left\|g^t\right\|^2 = p^2\gamma^2\eta^2\,\mathbb{E}\left\|\frac{1}{n}\sum_{i=1}^{n}Q_t^i(\nabla f_i(x_g^t))\right\|^2$$

$$\leq p^2\gamma^2\eta^2\frac{1}{n}\sum_{i=1}^{n}\mathbb{E}\left\|Q_t^i(\nabla f_i(x_g^t))\right\|^2$$

$$\overset{(4)}{\leq} p^2\gamma^2\eta^2\frac{d^2}{m^2}\frac{1}{n}\sum_{i=1}^{n}\mathbb{E}\left\|\nabla f_i(x_g^t)\right\|^2 \quad (31)$$

$$\overset{(5)}{\leq} 2p^2\gamma^2\eta^2\frac{d^2}{m^2}(\delta^2+1)\,\mathbb{E}\left\|\nabla f(x_g^t)\right\|^2 + 2p^2\gamma^2\eta^2\frac{d^2}{m^2}\sigma^2,$$

where in the last inequality we used Lemma 5.

Consider ③. We first use Lemma 6 twice

$$x_g^t = \theta x_f^t + (1-\theta)x^t = \alpha_f^{t-\tau}x_f^{t-\tau} + \alpha^{t-\tau}x^{t-\tau} - p\gamma\sum_{r=t-\tau}^{t-1}c_r g^r$$

$$\eta x_g^t + (p - \eta)x_f^t + (1-p)(1-\beta)x^t + (1-p)\beta x_g^t = \beta x_g^t + (1-\beta)x^t$$
$$= \beta\theta x_f^t + (1 - \beta\theta)x^t$$
$$= \hat\alpha_f^{t-\tau} x_f^{t-\tau} + \hat\alpha^{t-\tau} x^{t-\tau} - p\gamma \sum_{r=t-\tau}^{t-1} \hat c_r g^r.$$

Next, we apply Corollary 3 with $\hat a^{t-\tau} = \nabla f_i(\widetilde x_g^{t-\tau})$, where $\widetilde x_g^{t-\tau} = \alpha_f^{t-\tau} x_f^{t-\tau} + \alpha^{t-\tau} x^{t-\tau}$, and $\hat b^{t-\tau} = \hat\alpha_f^{t-\tau} x_f^{t-\tau} + \hat\alpha^{t-\tau} x^{t-\tau} - x^*$, leading us to

$$- 2p\gamma\eta\, \mathbb{E}\left\langle g^t - \nabla f(x_g^t), \eta x_g^t + (p-\eta)x_f^t + (1-p)(1-\beta)x^t + (1-p)\beta x_g^t - x^* \right\rangle$$

$$= -2p\gamma\eta \frac{1}{n}\sum_{i=1}^n \mathbb{E}\left\langle Q_t^i(\nabla f_i(x_g^t)) - \nabla f_i(x_g^t), \eta x_g^t + (p-\eta)x_f^t + (1-p)(1-\beta)x^t \right.$$
$$\left. + (1-p)\beta x_g^t - x^* \right\rangle$$

$$\leq \frac{\varepsilon d}{m\beta_0} p\gamma\eta \frac{1}{n}\sum_{i=1}^n \mathbb{E}\left[\left\|\nabla f_i(\widetilde x_g^{t-\tau})\right\|^2\right] + \frac{\varepsilon d\beta_0}{m} p\gamma\eta \mathbb{E}\left[\left\|\hat\alpha_f^{t-\tau} x_f^{t-\tau} + \hat\alpha^{t-\tau} x^{t-\tau} - x^*\right\|^2\right]$$

$$+ 4\frac{d^2}{m^2} p\gamma\eta\,(\beta_1 + \beta_2)\frac{1}{n}\sum_{i=1}^n \mathbb{E}\left[\left\|\nabla f_i(x_g^t) - \nabla f_i(\widetilde x_g^{t-\tau})\right\|^2\right]$$

$$+ p\gamma\eta\left(\frac{1}{\beta_1} + \frac{1}{\beta_3}\right)\mathbb{E}\left[\left\|-p\gamma\sum_{r=t-\tau}^{t-1} \hat c_r g^r\right\|^2\right]$$

$$+ 4\frac{d^2}{m^2} p\gamma\eta\beta_3 \frac{1}{n}\sum_{i=1}^n \mathbb{E}\left[\left\|\nabla f_i(x_g^t)\right\|^2\right] + \frac{p\gamma\eta}{\beta_2}\mathbb{E}\left[\left\|\beta x_g^t + (1-\beta)x^t - x^*\right\|^2\right].$$

Using Assumption 1 and Lemma 5 with $c_r \leq \tau \leq 2\tau$ and $\hat c_r \leq \eta$ one might obtain

$$- 2p\gamma\eta\, \mathbb{E}\left\langle g^t - \nabla f(x_g^t), \eta x_g^t + (p-\eta)x_f^t + (1-p)(1-\beta)x^t + (1-p)\beta x_g^t - x^* \right\rangle$$

$$\leq \frac{2\varepsilon d}{m\beta_0} p\gamma\eta(\delta^2 + 1)\mathbb{E}\left[\left\|\nabla f(\widetilde x_g^{t-\tau})\right\|^2\right] + \frac{\varepsilon d\beta_0}{m} p\gamma\eta\mathbb{E}\left[\left\|\hat\alpha_f^{t-\tau} x_f^{t-\tau} + \hat\alpha^{t-\tau} x^{t-\tau} - x^*\right\|^2\right]$$

$$+ 4\frac{d^2 L^2}{m^2} p\gamma\eta\,(\beta_1 + \beta_2)\mathbb{E}\left[\left\|-p\gamma\sum_{r=t-\tau}^{t-1} c_r g^r\right\|^2\right] + p\gamma\eta\left(\frac{1}{\beta_1} + \frac{1}{\beta_3}\right)\mathbb{E}\left[\left\|-p\gamma\sum_{r=t-\tau}^{t-1} \hat c_r g^r\right\|^2\right]$$

$$+ 8\frac{d^2}{m^2}(\delta^2 + 1)p\gamma\eta\beta_3\mathbb{E}\left[\left\|\nabla f(x_g^t)\right\|^2\right] + \frac{p\gamma\eta}{\beta_2}\mathbb{E}\left[\left\|\beta x_g^t + (1-\beta)x^t - x^*\right\|^2\right]$$

$$+ 2p\gamma\eta(\frac{\varepsilon d}{m\beta_0} + 4\frac{d^2\beta_3}{m^2})\sigma^2 \tag{32}$$

$$\leq \frac{\varepsilon d}{m} p\gamma\eta\left(2(\delta^2 + 1)L^2\alpha_f^{t-\tau}\frac{1}{\beta_0} + \beta_0\hat\alpha_f^{t-\tau}\right)\mathbb{E}\left[\left\|x_f^{t-\tau} - x^*\right\|^2\right]$$

$$+ \frac{\varepsilon d}{m} p\gamma\eta\left(2(\delta^2 + 1)L^2\alpha^{t-\tau}\frac{1}{\beta_0} + \beta_0\hat\alpha^{t-\tau}\right)\mathbb{E}\left[\left\|x^{t-\tau} - x^*\right\|^2\right]$$

$$+ p^3\gamma^3\eta\tau\left(4\frac{\tau^2 d^2 L^2}{m^2}(\beta_1 + \beta_2) + \eta^2\left(\frac{1}{\beta_1} + \frac{1}{\beta_3}\right)\right)\sum_{r=t-\tau}^{t-1} \|g^r\|^2$$

$$+ 8\frac{d^2}{m^2}(\delta^2 + 1)p\gamma\eta\beta_3\mathbb{E}\left[\left\|\nabla f(x_g^t)\right\|^2\right]$$

$$+ \frac{p\gamma\eta}{\beta_2}\beta\mathbb{E}\left[\left\|x_g^t - x^*\right\|^2\right] + \frac{p\gamma\eta}{\beta_2}(1-\beta)\mathbb{E}\left[\left\|x^t - x^*\right\|^2\right] + 2p\gamma\eta(\frac{\varepsilon d}{m\beta_0} + 4\frac{d^2\beta_3}{m^2})\sigma^2.$$

Consider ④. Taking into account line 4 and the choice of $\theta$ such that $\theta = (p\eta^{-1} - 1)/(\beta p\eta^{-1} - 1)$, one can note

$$\eta x_g^k + (p - \eta)x_f^k + (1 - p)(1 - \beta)x^k + (1 - p)\beta x_g^k - x^*$$

$$= (\eta + (1 - p)\beta)x_g^k + (p - \eta)x_f^k + (1 - p)(1 - \beta)x^k - x^*$$

$$= \eta p^{-1} \left((p + (1 - p)p^{-1}\eta\beta)x_g^k + (p\eta^{-1} - 1)px_f^k + (1 - p)(1 - \beta)p\eta^{-1}x^k - \eta^{-1}px^*\right)$$

$$= \eta p^{-1} \left((p + (1 - p)p^{-1}\eta\beta)x_g^k + (p\eta^{-1} - 1)px_f^k + (1 - p)(1 - \beta p\eta^{-1})(1 - \theta)x^k - \eta^{-1}px^*\right)$$

$$= \eta p^{-1} \left((p + (1 - p)p^{-1}\eta\beta)x_g^k + (p\eta^{-1} - 1)px_f^k + (1 - p)(1 - \beta p\eta^{-1})(x_g^k - \theta x_f^k) - \eta^{-1}px^*\right)$$

$$= \eta p^{-1} \left(x_g^k + (p\eta^{-1} - 1)px_f^k - (1 - p)(1 - \beta p\eta^{-1})\theta x_f^k - \eta^{-1}px^*\right)$$

$$= \eta p^{-1} \left(x_g^k + (p\eta^{-1} - 1)px_f^k - (1 - p)(p\eta^{-1} - 1)x_f^k - \eta^{-1}px^*\right)$$

$$= \eta p^{-1} \left(x_g^k + (p\eta^{-1} - 1)x_f^k - \eta^{-1}px^*\right). \tag{33}$$

Using that, we get

$$-2p\gamma\eta \, \mathbb{E} \left\langle \nabla f(x_g^t), \eta x_g^t + (p - \eta)x_f^t + (1 - p)(1 - \beta)x^t + (1 - p)\beta x_g^t - x^* \right\rangle$$

$$= -2\gamma\eta^2 \, \mathbb{E} \left\langle \nabla f(x_g^t), x_g^t + (p\eta^{-1} - 1)x_f^t - p\eta^{-1}x^* \right\rangle. \tag{34}$$

Summing (30), (31), (32) and (34) with $\beta_0 = \sqrt{\delta^2 + 1}L$, $\beta_1 = \beta_2 = \frac{4p\gamma\eta}{\beta}$ and $\beta_3 = p\gamma\eta$ we finish the proof. $\square$

**Lemma 8.** *Assume 1, 2 and 4. Then for iterates of Algorithm 2 and for any $u \in \mathbb{R}^d$ it holds that*

$$\mathbb{E}\left[f(x_f^{t+1})\right] \le \mathbb{E}[f(u)] - \mathbb{E}\left[\langle \nabla f(x_g^t), u - x_g^t \rangle\right] - \frac{\mu}{2}\|u - x_g^t\| - \frac{p\gamma}{2}\mathbb{E}\left[\|\nabla f(x_g^t)\|^2\right]$$

$$+ 2\varepsilon\gamma\mathbb{E}\left[\|\nabla f(\widetilde{x}_g^{t-\tau})\|^2\right] + 20\frac{L^2 d^3 \gamma^3 p^2 \tau^3 (\delta^2 + 1)}{m^3} \sum_{s=t-\tau}^{t-1} \mathbb{E}\left[\|\nabla f(x_g^s)\|^2\right] + 23\frac{L^2 d^3 \gamma^3 p^2 \tau^4}{m^3}\sigma^2,$$

*where*

$$\gamma \le \frac{1}{L} \quad and \quad p \le \frac{m^2}{12(\delta^2 + 1)d^2}.$$

*Proof.* Using 1 with $x = x_f^{t+1}$, $y = x_g^t$ and line 3 of Algorithm 2 we get

$$\mathbb{E}\left[f(x_f^{t+1})\right] \le \mathbb{E}\left[f(x_g^t)\right] + \mathbb{E}\left[\left\langle \nabla f(x_g^t), x_f^{t+1} - x_g^t \right\rangle\right] + \frac{L}{2}\mathbb{E}\left[\left\|x_f^{t+1} - x_g^t\right\|^2\right]$$

$$= \mathbb{E}\left[f(x_g^t)\right] - p\gamma\mathbb{E}\left[\langle \nabla f(x_g^t), g^t \rangle\right] + \frac{Lp^2\gamma^2}{2}\mathbb{E}\left[\|g^t\|^2\right] \tag{35}$$

$$= \mathbb{E}\left[f(x_g^t)\right] - p\gamma\mathbb{E}\left[\langle \nabla f(x_g^t), \nabla f(x_g^t) \rangle\right] - p\gamma\mathbb{E}\left[\langle \nabla f(x_g^t), g^k - \nabla f(x_g^t) \rangle\right]$$

$$+ \frac{Lp^2\gamma^2}{2}\mathbb{E}\left[\|g^t\|^2\right].$$

Consider $\mathbb{E}\left[\langle \nabla f(x_g^t), g^k - \nabla f(x_g^t) \rangle\right]$. Using Corollary 3 with $a^t = \nabla f_i(x_g^t)$, $b^t = \nabla f(x_g^t)$, $\hat{a}^{t-\tau} = \nabla f_i(\widetilde{x}_g^{t-\tau})$, $\hat{b}^{t-\tau} = \nabla f(\widetilde{x}_g^{t-\tau})$, where $x_g^t \in \text{conv}\left\{x_f^t, x^t\right\} = \widetilde{x}_g^{t-\tau} - p\gamma\sum_{s=t-\tau}^{t-1} c_s g^s$ from Lemma 6. Using Assumption 1 we obtain

$$2\left|\mathbb{E}\left[\langle \nabla f(x_g^t), g^k - \nabla f(x_g^t) \rangle\right]\right| \le \frac{\varepsilon d}{m\beta_0}\mathbb{E}\left[\frac{1}{n}\sum_{i=1}^n \|\nabla f_i(\widetilde{x}_g^{t-\tau})\|^2\right] + \frac{\varepsilon d\beta_0}{m}\mathbb{E}\left[\|\nabla f(\widetilde{x}_g^{t-\tau})\|^2\right]$$

$$+ 4\frac{d^2 L^2}{m^2}(\beta_1 + \beta_2)\mathbb{E}\left[\|x_g^t - \widetilde{x}_g^{t-\tau}\|^2\right] + L^2\left(\frac{1}{\beta_1} + \frac{1}{\beta_3}\right)\mathbb{E}\left[\|x_g^t - \widetilde{x}_g^{t-\tau}\|^2\right]$$

$$+ 4\frac{d^2}{m^2}\beta_3 \mathbb{E}\left[\frac{1}{n}\sum_{i=1}^n \left\|\nabla f_i(x_g^t)\right\|^2\right] + \frac{1}{\beta_2}\mathbb{E}\left[\left\|\nabla f(x_g^t)\right\|^2\right].$$

Taking $\beta_0 = \sqrt{\delta^2 + 1}$, $\beta_1 = m/d$, $\beta_2 = m/(dp)$, $\beta_3 = pm/d$ and using results from Lemma 5 we obtain

$$2\left|\mathbb{E}\left[\langle\nabla f(x_g^t), g^k - \nabla f(x_g^t)\rangle\right]\right| \leq \frac{2\varepsilon d}{m}\left(\sqrt{\delta^2 + 1}\mathbb{E}\left[\left\|\nabla f(\widetilde{x}_g^{t-\tau})\right\|^2\right] + \frac{\sigma^2}{\sqrt{\delta^2 + 1}}\right)$$

$$+ \frac{dp}{m}\mathbb{E}\left[\left\|\nabla f(x_g^t)\right\|^2\right] + 10\frac{L^2 d}{pm}\mathbb{E}\left[\left\|-p\gamma\sum_{s=t-\tau}^{t-1} c_s\frac{1}{n}\sum_{i=1}^n Q_s^i(\nabla f_i(x_g^s))\right\|^2\right]$$

$$+ \frac{8dp}{m}\left((\delta^2 + 1)\mathbb{E}\left[\left\|\nabla f(x_g^t)\right\|^2\right] + \sigma^2\right) + \frac{\varepsilon d\sqrt{\delta^2 + 1}}{m}\mathbb{E}\left[\left\|\nabla f(\widetilde{x}_g^{t-\tau})\right\|^2\right].$$

Using Lemma 4 and 5, convexity of the squared norm and the fact that $c_s \leq t - s + 2 \leq \tau + 2 \leq 2\tau$ we obtain

$$2\left|\mathbb{E}\left[\langle\nabla f(x_g^t), g^k - \nabla f(x_g^t)\rangle\right]\right| \leq \frac{3\varepsilon d\sqrt{\delta^2 + 1}}{m}\mathbb{E}\left[\left\|\nabla f(\widetilde{x}_g^{t-\tau})\right\|^2\right] +$$

$$+ 40\frac{L^2 d^3 \gamma^2 p\tau^3}{m^3}\sum_{s=t-\tau}^{t-1}\mathbb{E}\left[(\delta^2 + 1)\left\|\nabla f(x_g^s)\right\|^2 + \sigma^2\right]$$

$$+ \frac{9dp(\delta^2 + 1)}{m}\mathbb{E}\left[\left\|\nabla f(x_g^t)\right\|^2\right] + \frac{2d}{m}\left(\frac{\varepsilon}{\sqrt{\delta^2 + 1}} + p\right)\sigma^2.$$

Using the fact that $L^2\gamma^2 d^2/m^2\tau^4\eta^2 \geq 1$ and $\varepsilon \leq \sqrt{\delta^2 + 1}p$ we obtain

$$2\left|\mathbb{E}\left[\langle\nabla f(x_g^t), g^k - \nabla f(x_g^t)\rangle\right]\right| \leq \frac{3\varepsilon d\sqrt{\delta^2 + 1}}{m}\mathbb{E}\left[\left\|\nabla f(\widetilde{x}_g^{t-\tau})\right\|^2\right] + 44\frac{L^2 d^3 \gamma^2 p\eta^2\tau^4}{m^3}\sigma^2$$

$$+ 40\frac{L^2 d^3 \gamma^2 p\tau^3(\delta^2 + 1)}{m^3}\sum_{s=t-\tau}^{t-1}\mathbb{E}\left[\left\|\nabla f(x_g^s)\right\|^2\right] + \frac{9dp(\delta^2 + 1)}{m}\mathbb{E}\left[\left\|\nabla f(x_g^t)\right\|^2\right].$$

Using this result, Lemmas 4 and 5 we can estimate (35):

$$\mathbb{E}\left[f(x_f^{t+1})\right] = \mathbb{E}\left[f(x_g^t)\right] - p\gamma\mathbb{E}\left[\left\|\nabla f(x_g^t)\right\|^2\right]$$

$$- p\gamma\mathbb{E}\left[\langle\nabla f(x_g^t), g^k - \nabla f(x_g^t)\rangle\right] + \frac{L}{2}\mathbb{E}\left[\left\|g^t\right\|^2\right]$$

$$\leq \mathbb{E}\left[f(x_g^t)\right] - p\gamma\mathbb{E}\left[\left\|\nabla f(x_g^t)\right\|^2\right] + \frac{2\varepsilon p\gamma d\sqrt{\delta^2 + 1}}{m}\mathbb{E}\left[\left\|\nabla f(\widetilde{x}_g^{t-\tau})\right\|^2\right] +$$

$$+ 20\frac{L^2 d^3 \gamma^3 p^2\tau^3(\delta^2 + 1)}{m^3}\sum_{s=t-\tau}^{t-1}\mathbb{E}\left[\left\|\nabla f(x_g^s)\right\|^2\right] + \frac{5d\gamma p^2(\delta^2 + 1)}{m}\mathbb{E}\left[\left\|\nabla f(x_g^t)\right\|^2\right]$$

$$+ 22\frac{L^2 d^3 \gamma^3 p^2\tau^4}{m^3}\sigma^2 + \frac{Lp^2\gamma^2 d^2}{m^2}(\delta^2 + 1)\mathbb{E}\left[\left\|\nabla f(x_g^t)\right\|^2\right] + \frac{Lp^2\gamma^2 d^2}{m^2}\sigma^2.$$

Taking

$$\gamma \leq \frac{1}{L} \quad \text{and} \quad p \leq \frac{m^2}{12(\delta^2 + 1)d^2},$$

we obtain

$$\mathbb{E}\left[f(x_f^{t+1})\right] \leq \mathbb{E}\left[f(x_g^t)\right] - \frac{p\gamma}{2}\mathbb{E}\left[\left\|\nabla f(x_g^t)\right\|^2\right] + 2\varepsilon\gamma\mathbb{E}\left[\left\|\nabla f(\widetilde{x}_g^{t-\tau})\right\|^2\right] +$$
$$+ 20\frac{L^2 d^3 \gamma^3 p^2 \tau^3 (\delta^2+1)}{m^3} \sum_{s=t-\tau}^{t-1} \mathbb{E}\left[\left\|\nabla f(x_g^s)\right\|^2\right] + 23\frac{L^2 d^3 \gamma^3 p^2 \tau^4}{m^3}\sigma^2.$$

Using 2 with $x = u$ and $y = x_g^t$, one can conclude that for any $u \in \mathbb{R}^d$ it holds

$$\mathbb{E}\left[f(x_f^{t+1})\right] \leq \mathbb{E}\left[f(u)\right] - \mathbb{E}\left[\langle\nabla f(x_g^t), u - x_g^t\rangle\right] - \frac{\mu}{2}\left\|u - x_g^t\right\|$$
$$- \frac{p\gamma}{2}\mathbb{E}\left[\left\|\nabla f(x_g^t)\right\|^2\right] + 2\varepsilon\gamma\mathbb{E}\left[\left\|\nabla f(\widetilde{x}_g^{t-\tau})\right\|^2\right] +$$
$$+ 20\frac{L^2 d^3 \gamma^3 p^2 \tau^3 (\delta^2+1)}{m^3} \sum_{s=t-\tau}^{t-1} \mathbb{E}\left[\left\|\nabla f(x_g^s)\right\|^2\right] + 23\frac{L^2 d^3 \gamma^3 p^2 \tau^4}{m^3}\sigma^2.$$

This finishes the proof. $\qquad\square$

**Theorem 7** (Theorem 3). *Consider Assumptions 1, 2 and 4. Let problem* (1) *be solved by Algorithm 2. Then for any* $\gamma > 0, \varepsilon > 0, \tau > \tau_{mix}(\varepsilon), T > \tau$ *and* $\beta, \theta, \eta, p$ *satisfying*

$$\gamma \leq \frac{\mu^{\frac{1}{3}} m^{\frac{1}{2}}}{2\tau L^{\frac{4}{3}} d^{\frac{1}{2}}}, \quad \varepsilon \leq \min\left\{\frac{m^{\frac{7}{4}}}{6d^{\frac{7}{4}}\tau^{\frac{5}{4}}L(\delta^2+1)}; \frac{m^{\frac{5}{4}}}{\sqrt{2}\tau^{\frac{3}{4}}\mu^{\frac{1}{3}}L^{\frac{2}{3}}d^{\frac{5}{4}}}; \frac{m^{\frac{15}{4}}}{6d^{\frac{15}{4}}\tau^{\frac{13}{4}}(\delta^2+1)^2}\right\},$$

$$p \leq \frac{m^2}{13d^2(\delta^2+1)\tau^2}, \quad \beta = \sqrt{\frac{2p^2\mu\gamma}{3}}, \quad \eta = \sqrt{\frac{3}{2\mu\gamma}}, \quad \theta = \frac{p\eta^{-1}-1}{\beta p\eta^{-1}-1}.$$

*it holds that*

$$\mathbb{E}[\|x^{T+1} - x^*\|^2 + \frac{3}{\mu}(f(x_f^{T+1}) - f(x^*))] \leq \exp\left(-(T-\tau)\sqrt{\frac{2p^2\mu\gamma}{3}}\right)F_\tau$$
$$+ \exp\left(-T\sqrt{\frac{2p^2\mu\gamma}{3}}\right)\Delta_\tau + \frac{45\gamma}{\mu}\sigma^2,$$

*where* $F_\tau := \mathbb{E}[\|x^\tau - x^*\|^2 + \frac{3}{\mu}(f(x_f^\tau) - f(x^*))]$ *and* $\Delta_\tau \leq \frac{\sqrt{\gamma}}{\tau^{\frac{4}{3}}\mu^{\frac{1}{3}}}\sum_{t=0}^{\tau}\left(\mathbb{E}\left\|\nabla f(x_g^t)\right\| +\right.$
$$\left.\mathbb{E}\left\|x^t - x^*\right\|^2 + \mathbb{E}[f(x_f^t) - f(x^*)]\right).$$

*Proof.* We start by using Lemma 8 with $u = x^*$ and $u = x_f^t$

$$\mathbb{E}\left[f(x_f^{t+1})\right] \leq \mathbb{E}\left[f(x^*)\right] - \mathbb{E}\left[\langle\nabla f(x_g^t), x^* - x_g^t\rangle\right] - \frac{\mu}{2}\left\|x^* - x_g^t\right\| - \frac{p\gamma}{2}\mathbb{E}\left[\left\|\nabla f(x_g^t)\right\|^2\right]$$
$$+ 2\varepsilon\gamma\mathbb{E}\left[\left\|\nabla f(\widetilde{x}_g^{t-\tau})\right\|^2\right] + 20\frac{L^2 d^3 \gamma^3 p^2 \tau^3 (\delta^2+1)}{m^3} \sum_{s=t-\tau}^{t-1} \mathbb{E}\left[\left\|\nabla f(x_g^s)\right\|^2\right] + 23\frac{L^2 d^3 \gamma^3 p^2 \tau^4}{m^3}\sigma^2,$$

$$\mathbb{E}\left[f(x_f^{t+1})\right] \leq \mathbb{E}\left[f(x_f^t)\right] - \mathbb{E}\left[\langle\nabla f(x_g^t), x_f^t - x_g^t\rangle\right] - \frac{\mu}{2}\left\|x_f^t - x_g^t\right\| - \frac{p\gamma}{2}\mathbb{E}\left[\left\|\nabla f(x_g^t)\right\|^2\right]$$
$$+ 2\varepsilon\gamma\mathbb{E}\left[\left\|\nabla f(\widetilde{x}_g^{t-\tau})\right\|^2\right] + 20\frac{L^2 d^3 \gamma^3 p^2 \tau^3 (\delta^2+1)}{m^3} \sum_{s=t-\tau}^{t-1} \mathbb{E}\left[\left\|\nabla f(x_g^s)\right\|^2\right] + 23\frac{L^2 d^3 \gamma^3 p^2 \tau^4}{m^3}\sigma^2.$$

Summing the first inequality with coefficient $2p\gamma\eta$, the second with coefficient $2p\gamma\eta(\eta - p)$ and (29), we get

$$\mathbb{E}[\|x^{t+1} - x^*\|^2 + 2\gamma\eta^2 f(x_f^{t+1})]$$

$$\leq (1-\beta)(1+\frac{\beta}{4}) \, \mathbb{E} \left\| x^t - x^* \right\|^2 + \beta(1+\frac{\beta}{4}) \, \mathbb{E} \left\| x_g^t - x^* \right\|^2 + (\beta^2 - \beta) \, \mathbb{E} \left\| x^t - x_g^t \right\|^2$$

$$+ 10\frac{d^2}{m^2}(\delta^2+1)p^2\gamma^2\eta^2 \, \mathbb{E} \left\| \nabla f(x_g^t) \right\|^2 + p^2\gamma^2\eta^2\tau \left( 32\frac{\tau^2 d^2 L^2 p^2 \gamma^2}{m^2\beta} + \frac{5}{4} \right) \sum_{r=t-\tau}^{t-1} \| g^r \|^2$$

$$+ 3\varepsilon p\gamma\eta L\frac{d}{m}\sqrt{\delta^2+1}\mathbb{E}\left[ \left\| x^{t-\tau} - x^* \right\|^2 \right] + 3\varepsilon p\gamma\eta L\frac{d}{m}\sqrt{\delta^2+1}\mathbb{E}\left[ \left\| x_f^{t-\tau} - x^* \right\|^2 \right]$$

$$- 2\gamma\eta^2 \, \mathbb{E} \left\langle \nabla f(x_g^t), x_g^t + (p\eta^{-1}-1)x_f^t - p\eta^{-1}x^* \right\rangle + 2p\gamma\eta \left( \frac{\varepsilon d}{m\sqrt{\delta^2+1}L} + 4p\gamma\eta\frac{d^2}{m^2} \right)\sigma^2$$

$$+ 2p\gamma\eta \Bigg( \mathbb{E} \left[ f(x^*) \right] - \mathbb{E} \left[ \langle \nabla f(x_g^t), x^* - x_g^t \rangle \right] - \frac{\mu}{2} \left\| x^* - x_g^t \right\| - \frac{p\gamma}{2}\mathbb{E}\left[ \left\| \nabla f(x_g^t) \right\|^2 \right]$$

$$+ 2\varepsilon\gamma\mathbb{E}\left[ \left\| \nabla f(\widetilde{x}_g^{t-\tau}) \right\|^2 \right] + 20\frac{L^2 d^3 \gamma^3 p^2 \tau^3 (\delta^2+1)}{m^3} \sum_{s=t-\tau}^{t-1} \mathbb{E}\left[ \left\| \nabla f(x_g^s) \right\|^2 \right]$$

$$+ 23\frac{L^2 d^3 \gamma^3 p^2 \tau^4}{m^3}\sigma^2 \Bigg)$$

$$+ 2\gamma\eta(\eta-p) \Bigg( \mathbb{E} \left[ f(x_f^t) \right] - \mathbb{E} \left[ \langle \nabla f(x_g^t), x_f^t - x_g^t \rangle \right] - \frac{\mu}{2} \left\| x_f^t - x_g^t \right\| - \frac{p\gamma}{2}\mathbb{E}\left[ \left\| \nabla f(x_g^t) \right\|^2 \right]$$

$$+ 2\varepsilon\gamma\mathbb{E}\left[ \left\| \nabla f(\widetilde{x}_g^{t-\tau}) \right\|^2 \right] + 20\frac{L^2 d^3 \gamma^3 p^2 \tau^3 (\delta^2+1)}{m^3} \sum_{s=t-\tau}^{t-1} \mathbb{E}\left[ \left\| \nabla f(x_g^s) \right\|^2 \right]$$

$$+ 23\frac{L^2 d^3 \gamma^3 p^2 \tau^4}{m^3}\sigma^2 \Bigg)$$

$$\leq (1-\beta)(1+\frac{\beta}{4}) \, \mathbb{E} \left\| x^t - x^* \right\|^2 + (\beta + \frac{\beta^2}{4} - p\gamma\eta\mu) \, \mathbb{E} \left\| x_g^t - x^* \right\|^2 + (\beta^2 - \beta) \, \mathbb{E} \left\| x^t - x_g^t \right\|^2$$

$$+ p^2\gamma^2\eta^2 \left( 10\frac{d^2}{m^2}(\delta^2+1) - \frac{1}{p} \right) \mathbb{E} \left\| \nabla f(x_g^t) \right\| + 2p\gamma\eta \, \mathbb{E} \, f(x^*) + 2\gamma\eta(\eta-p) \, \mathbb{E} \, f(x_f^t)$$

$$+ p^2\gamma^2\eta^2\tau(\delta^2+1)\frac{d^2}{m^2}\left( 32\frac{\tau^2 d^2 L^2 p^2 \gamma^2}{m^2\beta} + \frac{5}{4} \right) \sum_{r=t-\tau}^{t-1} \mathbb{E} \left\| \nabla f(x_g^r) \right\|$$

$$+ \varepsilon\gamma\eta L(3p\frac{d}{m}\sqrt{\delta^2+1} + 2\gamma\eta L)\mathbb{E}\left[ \left\| x^{t-\tau} - x^* \right\|^2 \right]$$

$$+ \varepsilon\gamma\eta L(3p\frac{d}{m}\sqrt{\delta^2+1} + 2\gamma\eta L)\mathbb{E}\left[ \left\| x_f^{t-\tau} - x^* \right\|^2 \right]$$

$$+ 2p\gamma\eta \Bigg( \frac{\varepsilon d}{m\sqrt{\delta^2+1}L} + 4p\gamma\eta\frac{d^2}{m^2}$$

$$+ 23p\gamma^3\eta\tau^4\frac{d^3}{m^3}L^2 + p\gamma\eta\tau^2\frac{d^2}{m^2}\left( 16\frac{\tau^2 d^2 L^2 p^2 \gamma^2}{m^2\beta} + \frac{5}{8} \right) \Bigg)\sigma^2,$$

where in the last inequality we used Lemma 5 and Assumption 1. Since $\beta < 1$, the choice of $p\gamma\eta\mu = \frac{3\beta}{2}$ gives

$$(1-\beta)(1+\frac{\beta}{4}) \leq 1 - \frac{3\beta}{4},$$

$$\beta + \frac{\beta^2}{4} - p\gamma\eta\mu \leq \frac{3\beta}{2} - p\gamma\eta\mu \leq 0,$$

$$\beta^2 - \beta \leq 0.$$

This lead us to

$$\mathbb{E}[\|x^{t+1} - x^*\|^2 + 2\gamma\eta^2(f(x_f^{t+1}) - f(x^*))]$$

$$\leq (1 - \frac{3\beta}{4})\mathbb{E}\left\|x^t - x^*\right\|^2 + 2p\gamma\eta^2(1 - \frac{p}{\eta})\mathbb{E}[f(x_f^t) - f(x^*)]$$

$$+ p^2\gamma^2\eta^2\left(10\frac{d^2}{m^2}(\delta^2 + 1) - \frac{1}{p}\right)\mathbb{E}\left\|\nabla f(x_g^t)\right\|$$

$$+ p^2\gamma^2\eta^2\tau(\delta^2 + 1)\frac{d^2}{m^2}\left(32\frac{\tau^2 d^2 L^2 p^2 \gamma^2}{m^2\beta} + \frac{5}{4}\right)\sum_{r=t-\tau}^{t-1}\mathbb{E}\left\|\nabla f(x_g^r)\right\| \qquad (36)$$

$$+ \varepsilon\gamma\eta L(3p\frac{d}{m}\sqrt{\delta^2 + 1} + 2\gamma\eta L)\mathbb{E}\left[\left\|x^{t-\tau} - x^*\right\|^2\right]$$

$$+ \varepsilon\gamma\eta L(3p\frac{d}{m}\sqrt{\delta^2 + 1} + 2\gamma\eta L)\frac{2}{\mu}\mathbb{E}[f(x_f^{t-\tau}) - f(x^*)]$$

$$+ 2p\gamma\eta\left(\frac{\varepsilon d}{m\sqrt{\delta^2 + 1}L} + 4p\gamma\eta\frac{d^2}{m^2}\right.$$

$$\left. + 23p\gamma^3\eta\tau^4\frac{d^3}{m^3}L^2 + p\gamma\eta\tau^2\frac{d^2}{m^2}\left(16\frac{\tau^2 d^2 L^2 p^2 \gamma^2}{m^2\beta} + \frac{5}{8}\right)\right)\sigma^2,$$

where we also used Assumption 2 and subtracted $2\gamma\eta^2 f(x^*)$ from both sides. Next, we perform the summation from $t = \tau$ to $t = T > \tau$ of equations (36) with coefficients $p_t$:

$$\sum_{t=\tau}^{T} p_t\, \mathbb{E}[\|x^{t+1} - x^*\|^2 + 2\gamma\eta^2(f(x_f^{t+1}) - f(x^*))]$$

$$\leq \sum_{t=\tau}^{T} p_t(1 - \frac{3\beta}{4})\mathbb{E}\left\|x^t - x^*\right\|^2$$

$$+ \sum_{t=\tau}^{T} p_t 2p\gamma\eta^2(1 - \frac{p}{\eta})\mathbb{E}[f(x_f^t) - f(x^*)] + \sum_{t=\tau}^{T} p_t p^2\gamma^2\eta^2\left(10\frac{d^2}{m^2}(\delta^2 + 1) - \frac{1}{p}\right)\mathbb{E}\left\|\nabla f(x_g^t)\right\|$$

$$+ \sum_{t=\tau}^{T} p_t p^2\gamma^2\eta^2\tau(\delta^2 + 1)\frac{d^2}{m^2}\left(32\frac{\tau^2 d^2 L^2 p^2 \gamma^2}{m^2\beta} + \frac{5}{4}\right)\sum_{r=t-\tau}^{t-1}\mathbb{E}\left\|\nabla f(x_g^r)\right\|$$

$$+ \sum_{t=\tau}^{T} p_t\varepsilon\gamma\eta L(3p\frac{d}{m}\sqrt{\delta^2 + 1} + 2\gamma\eta L)\mathbb{E}\left[\left\|x^{t-\tau} - x^*\right\|^2\right]$$

$$+ \sum_{t=\tau}^{T} p_t\varepsilon\gamma\eta L(3p\frac{d}{m}\sqrt{\delta^2 + 1} + 2\gamma\eta L)\frac{2}{\mu}\mathbb{E}[f(x_f^{t-\tau}) - f(x^*)]$$

$$+ \sum_{t=\tau}^{T} p_t 2p\gamma\eta\left(\frac{\varepsilon d}{m\sqrt{\delta^2 + 1}L} + 4p\gamma\eta\frac{d^2}{m^2}\right.$$

$$\left. + 23p\gamma^3\eta\tau^4\frac{d^3}{m^3}L^2 + p\gamma\eta\tau^2\frac{d^2}{m^2}\left(16\frac{\tau^2 d^2 L^2 p^2 \gamma^2}{m^2\beta} + \frac{5}{8}\right)\right)\sigma^2.$$

Similar as in Theorem 5 we take $p_t = p^t$, $p = (1 - \frac{\beta}{2})^{-1}$, it implies $p_\tau \leq 6$ and therefore

$$\sum_{t=\tau}^{T} p_t\, \mathbb{E}[\|x^{t+1} - x^*\|^2 + 2\gamma\eta^2(f(x_f^{t+1}) - f(x^*))]$$

$$\leq \sum_{t=\tau}^{T} p_t\left(1 - \frac{3\beta}{4} + 6\varepsilon\gamma\eta L\left(3p\frac{d}{m}\sqrt{\delta^2 + 1} + 2\gamma\eta L\right)\right)\mathbb{E}\left\|x^t - x^*\right\|^2$$

$$+ \sum_{t=\tau}^{T} p_t \left( 2p\gamma\eta^2(1 - \frac{p}{\eta}) + 12\frac{\varepsilon\gamma\eta L}{\mu}\left(3p\frac{d}{m}\sqrt{\delta^2+1} + 2\gamma\eta L\right) \right) \mathbb{E}[f(x_f^t) - f(x^*)]$$

$$+ \sum_{t=\tau}^{T} p_t p^2\gamma^2\eta^2 \left( 10\frac{d^2}{m^2}(\delta^2+1) - \frac{1}{p} + \tau^2(\delta^2+1)\frac{d^2}{m^2}\left(32\frac{\tau^2 d^2 L^2 p^2 \gamma^2}{m^2\beta} + \frac{5}{4}\right) \right) \mathbb{E}\left\| \nabla f(x_g^t) \right\|$$

$$+ \sum_{t=0}^{\tau} p_{t+\tau} 8p^2\gamma^4\eta^2(\delta^2+1)\frac{d^3}{m^3}\tau^3 L^2\left(\frac{2p^2 d}{m\beta} + 5\right) \sum_{r=t-\tau}^{t-1} \mathbb{E}\left\| \nabla f(x_g^r) \right\|$$

$$+ \sum_{t=0}^{\tau} p_{t+\tau}\varepsilon\gamma\eta L(3p\frac{d}{m}\sqrt{\delta^2+1} + 2\gamma\eta L)\mathbb{E}\left[ \left\| x^t - x^* \right\|^2 \right]$$

$$+ \sum_{t=0}^{\tau} p_{t+\tau}\varepsilon\gamma\eta L(3p\frac{d}{m}\sqrt{\delta^2+1} + 2\gamma\eta L)\frac{2}{\mu}\mathbb{E}[f(x_f^t) - f(x^*)]$$

$$+ \sum_{t=\tau}^{T} p_t 2p\gamma\eta \left( \frac{\varepsilon d}{m\sqrt{\delta^2+1}L} + 4p\gamma\eta\frac{d^2}{m^2} + 23p\gamma^3\eta\tau^4\frac{d^3}{m^3}L^2 \right.$$

$$\left. + p\gamma\eta\tau^2\frac{d^2}{m^2}\left(16\frac{\tau^2 d^2 L^2 p^2 \gamma^2}{m^2\beta} + \frac{5}{8}\right) \right)\sigma^2.$$

Taking

$$\gamma \leq \frac{\mu^{\frac{1}{3}}m^{\frac{1}{2}}}{2\tau L^{\frac{4}{3}}d^{\frac{1}{2}}}, \quad p \leq \frac{m^2}{13d^2(\delta^2+1)\tau^2},$$

$$\varepsilon \leq \min\left\{ \frac{m^{\frac{7}{4}}}{6d^{\frac{7}{4}}\tau^{\frac{5}{4}}L(\delta^2+1)}; \frac{m^{\frac{5}{4}}}{\sqrt{2}\tau^{\frac{3}{4}}\mu^{\frac{1}{3}}L^{\frac{2}{3}}d^{\frac{5}{4}}}; \frac{m^{\frac{15}{4}}}{6d^{\frac{15}{4}}\tau^{\frac{13}{4}}(\delta^2+1)^2} \right\},$$

we get

$$10\frac{d^2}{m^2}(\delta^2+1) - \frac{1}{p} + \tau^2(\delta^2+1)\frac{d^2}{m^2}\left(32\frac{\tau^2 d^2 L^2 p^2 \gamma^2}{m^2\beta} + \frac{5}{4}\right) \leq 0,$$

$$6\varepsilon\gamma\eta L\left(3p\frac{d}{m}\sqrt{\delta^2+1} + 2\gamma\eta L\right) \leq \frac{\beta}{4},$$

$$12\frac{\varepsilon\gamma\eta L}{\mu}(3p\frac{d}{m}\sqrt{\delta^2+1} + 2\gamma\eta L) \leq 2p\gamma\eta^2\frac{p}{2\eta},$$

and therefore with $\beta = \frac{p}{\eta}$

$$\sum_{t=\tau}^{T} p_t \mathbb{E}[\|x^{t+1} - x^*\|^2 + 2\gamma\eta^2(f(x_f^{t+1}) - f(x^*))]$$

$$\leq \sum_{t=\tau}^{T} p_t\left(1 - \frac{\beta}{2}\right) \mathbb{E}[\|x^t - x^*\|^2 + 2\gamma\eta^2(f(x_f^t) - f(x^*))]$$

$$+ \sum_{t=0}^{\tau} p_{t+\tau} 8p^2\gamma^4\eta^2(\delta^2+1)\frac{d^3}{m^3}\tau^3 L^2\left(\frac{2p^2 d}{m\beta} + 5\right) \sum_{r=t-\tau}^{t-1} \mathbb{E}\left\| \nabla f(x_g^r) \right\|$$

$$+ \sum_{t=0}^{\tau} p_{t+\tau}\varepsilon\gamma\eta L(3p\frac{d}{m}\sqrt{\delta^2+1} + 2\gamma\eta L)\mathbb{E}\left[ \left\| x^t - x^* \right\|^2 \right]$$

$$+ \sum_{t=0}^{\tau} p_{t+\tau}\varepsilon\gamma\eta L(3p\frac{d}{m}\sqrt{\delta^2+1} + 2\gamma\eta L)\frac{2}{\mu}\mathbb{E}[f(x_f^t) - f(x^*)]$$

$$+ \sum_{t=\tau}^{T} p_t 2p\gamma\eta \left( \frac{\varepsilon d}{m\sqrt{\delta^2+1}L} + 4p\gamma\eta\frac{d^2}{m^2} \right.$$

$$+ 23p\gamma^3\eta\tau^4\frac{d^3}{m^3}L^2 + p\gamma\eta\tau\frac{d^2}{m^2}\left(16\frac{\tau^2 d^2 L^2 p^2 \gamma^2}{m^2\beta} + \frac{5}{8}\right)\bigg)\sigma^2.$$

Assume the following notation

$$\Delta_\tau := \sum_{t=0}^{\tau} p_{t+\tau} 8p^2\gamma^4\eta^2(\delta^2+1)\frac{d^3}{m^3}\tau^3 L^2\left(\frac{2p^2 d}{m\beta} + 5\right)\sum_{r=t-\tau}^{t-1}\mathbb{E}\left\|\nabla f(x_g^r)\right\|$$

$$+ \sum_{t=0}^{\tau} p_{t+\tau}\varepsilon\gamma\eta L(3p\frac{d}{m}\sqrt{\delta^2+1} + 2\gamma\eta L)\mathbb{E}\left[\left\|x^t - x^*\right\|^2\right]$$

$$+ \sum_{t=0}^{\tau} p_{t+\tau}\varepsilon\gamma\eta L(3p\frac{d}{m}\sqrt{\delta^2+1} + 2\gamma\eta L)\frac{2}{\mu}\mathbb{E}[f(x_f^t) - f(x^*)]$$

$$\le \frac{\sqrt{\gamma}}{\tau^{\frac{4}{3}}\mu^{\frac{1}{3}}}\sum_{t=0}^{\tau}\left(\mathbb{E}\left\|\nabla f(x_g^t)\right\| + \mathbb{E}\left\|x^t - x^*\right\|^2 + \mathbb{E}[f(x_f^t) - f(x^*)]\right)$$

Now we substitute $p_t$, this lead us to

$$\sum_{t=\tau}^{T}\left(1 - \frac{\beta}{2}\right)^{-t}\mathbb{E}[\|x^{t+1} - x^*\|^2 + 2\gamma\eta^2(f(x_f^{t+1}) - f(x^*))]$$

$$\le \sum_{t=\tau}^{T}\left(1 - \frac{\beta}{2}\right)^{-t+1}\mathbb{E}[\|x^t - x^*\|^2 + 2\gamma\eta^2(f(x_f^t) - f(x^*))] + \Delta_\tau$$

$$+ \sum_{t=\tau}^{T}\left(1 - \frac{\beta}{2}\right)^{-t}2p\gamma\eta\bigg(\frac{\varepsilon d}{m\sqrt{\delta^2+1}L} + 4p\gamma\eta\frac{d^2}{m^2}$$

$$+ 23p\gamma^3\eta\tau^4\frac{d^3}{m^3}L^2 + p\gamma\eta\tau\frac{d^2}{m^2}\left(16\frac{\tau^2 d^2 L^2 p^2 \gamma^2}{m^2\beta} + \frac{5}{8}\right)\bigg)\sigma^2.$$

This implies

$$\left(1 - \frac{\beta}{2}\right)^{-T}\mathbb{E}[\|x^{T+1} - x^*\|^2 + 2\gamma\eta^2(f(x_f^{T+1}) - f(x^*))] \le \left(1 - \frac{\beta}{2}\right)^{\tau}\mathbb{E}[\|x^\tau - x^*\|^2$$

$$+ 2\gamma\eta^2(f(x_f^\tau) - f(x^*))] + \Delta_\tau$$

$$+ \sum_{t=\tau}^{T}\left(1 - \frac{\beta}{2}\right)^{-t}2p\gamma\eta\bigg(\frac{\varepsilon d}{m\sqrt{\delta^2+1}L} + 4p\gamma\eta\frac{d^2}{m^2}$$

$$+ 23p\gamma^3\eta\tau^4\frac{d^3}{m^3}L^2 + p\gamma\eta\tau\frac{d^2}{m^2}\left(16\frac{\tau^2 d^2 L^2 p^2 \gamma^2}{m^2\beta} + \frac{5}{8}\right)\bigg)\sigma^2.$$

Rearranging this inequality and taking $\varepsilon \le \frac{\sqrt{\gamma}m}{\sqrt{\mu}d}$ we obtain

$$\mathbb{E}[\|x^{T+1} - x^*\|^2 + 2\gamma\eta^2(f(x_f^{T+1}) - f(x^*))]$$

$$\le \left(1 - \frac{\beta}{2}\right)^{T-\tau}\mathbb{E}[\|x^\tau - x^*\|^2 + 2\gamma\eta^2(f(x_f^\tau) - f(x^*))] + \left(1 - \frac{\beta}{2}\right)^{T}\Delta_\tau + 6\sqrt{\frac{\gamma}{\mu}}\sigma^2.$$

This finishes the proof. □

## H    EXPERIMENTS

This section provides description of the experiment setup, presents and analyses results of logistic regression experiments on LIBSVM datasets, studies dependence of history size over convergence. Moreover, experiments with neural networks optimization for data-parallelism and model-parallelism are presented and discussed.

## H.1 Technical details

Our implementation of compression operators and algorithms is written in Python 3.10, with the use of PyTorch optimization library. We implement a simulation of distributed optimization system on a single machine, which is equivalent in terms of convergence analysis. Our server is AMD Ryzen Threadripper 2950X 16-Core Processor @ 2.2 GHz CPU and x2 NVIDIA GeForce GTX 1080 Ti GPU. We use Weights&Biases Biewald (2020) for experiments tracking and hyperparameters tuning.

## H.2 Logistic Regression experiments

We conduct experiments on classification with logistic regression on four datasets: Mushrooms, A9A, W8A, MNIST. We apply the following optimization algorithms: proposed MQSGD and its accelerated version AMQSGD, and also use Markovian compressors with popular DIANA Mishchenko et al. (2019) algorithm. In all of our experiments, we do not utilize the steps of the optimizer, but rather the information that is transmitted by each worker at the current timestamp $t$. This implies that there are $n$ workers, with each worker sending $m$ coordinates at each iteration of the optimization step. Consequently, the $x$-axis displays numbers of the form $mn \cdot 1, mn \cdot 2, \ldots, mn \cdot t, \ldots, mn \cdot T$. This allows us to understand the performance of compressors with varying values of $m$ and $n$.

We use convex logistic regression loss with a regularization term $\lambda = 0.05$. Each dataset is split horizontally (by rows) equally between $N = 10$ clients. The feature dimension is denoted as $d$ in the figures, varying from hundreds to almost a thousand between datasets. The underlying sparsification compressors in Rand-10% for all logistic regression experiments. Learning rate initial value and decay rate are fine-tuned for each problem and compressor. Additionally, Markovian-specific parameters such as history size $K$, forgetting rate $b$ are also fine-tuned. Table 2 provides hyperparameters grid for the tuning. We obtain optimal solution $x^*$ for each problem with scipy.optimize method in order to use this value for the graphics.

Table 2: Hyperparameters values used for tuning in the experiments.

| Hyperparameters | Values List |
|---|---|
| Learning rate | $[0.01, 0.03, 0.05, 0.1, 0.3, 0.5, 1]$ |
| Learning rate decay rate | $[0.5, 0.8, 1]$ |
| History size $K$ | $[1 \ldots 40]$ |
| Forgetting rate $B$ | $[1, 10, 15, 20, 30, 50]$ |

Figures 5, 6 and 7 present relative distance to the optimum and gradient norm for the best runs on MQSGD, AMQSGD and DIANA, respectively. We observe that Markovian compressors consistently outperform the Rand-10% baseline in all scenarios, as the diverging trend can be seen. Only in some experiments with DIANA (MNIST) the advantage is negligible although present. We also observe that simpler and computational-effective BanLast compressor is often enough to achieve substantial convergence improvement. Notably, fine-tuned hyperparameters are similar across datasets and algorithms: for example, BanLast tends to perform best with largest possible values of history size $K$, and KAWASAKI forgetting rate $b$ is large. Notice that BanLast compressor with largest $K$ turns into round-robin compressor with (almost) no stochasticity in coordinates choice.

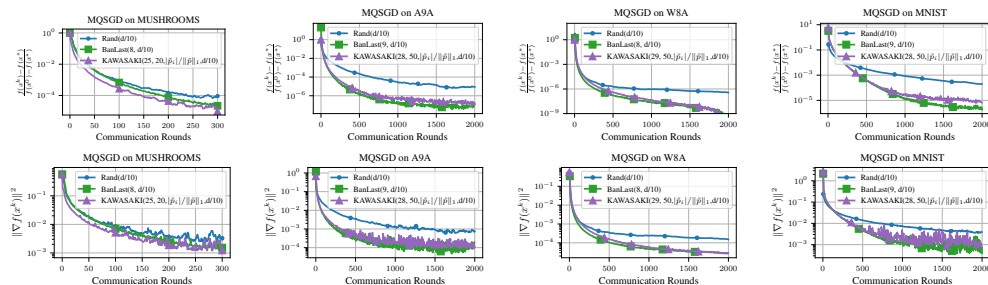

Figure 5: `MQSGD` LIBSVM logistic regression experiments. Best run after hyperparameters tuning is displayed for each method.

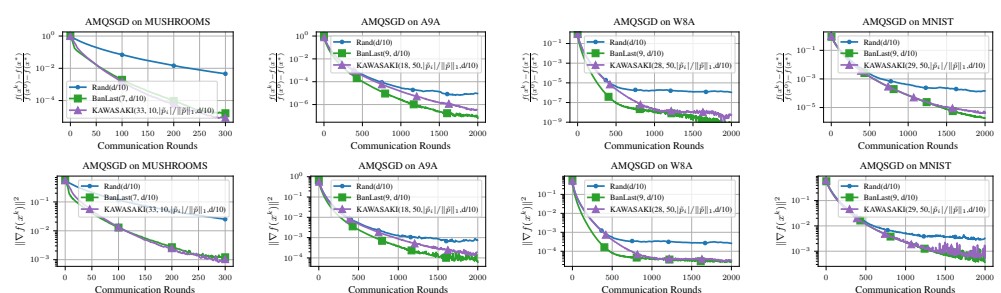

Figure 6: `AMQSGD` LIBSVM logistic regression experiments. Best run after hyperparameters tuning is displayed for each method.

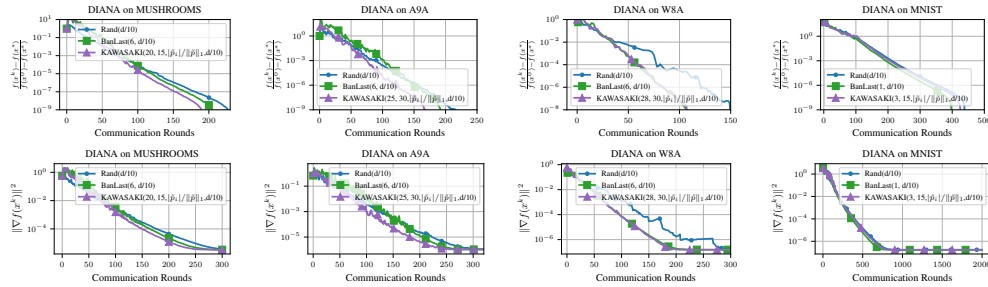

Figure 7: `DIANA` LIBSVM logistic regression experiments. Best run after hyperparameters tuning is displayed for each method.

### H.3 DEPENDENCE ON SIZE HISTORY

As a part of hyperparameter tuning, we additionally analyze how history size $K$ affects the convergence of Markovian compression-based methods. Figure 8 presents dependence of distance to optimum metric on history size for logistic regression experiments. We observe that `BanLast` performs better around larger values of $K = 8$ or $K = 9$. In such case for Rand10% used along with `BanLast(9)`, the compression procedure resembles a permutation: for each 10 iterations, no indices are repeated, and the transmission cycle repeats after that. `KAWASAKI` history size seems to have periodical spikes and drops, achieving minimum at around $K = 25$. However, statistics for `DIANA` differ drastically, indicating that history size should be adjusted for each problem independently.

### H.4 COMPARISON WITH PERMUTATION & NATURAL COMPRESSION

In this section, we provide empirical comparison of the proposed compressors with other complex compression schemes.

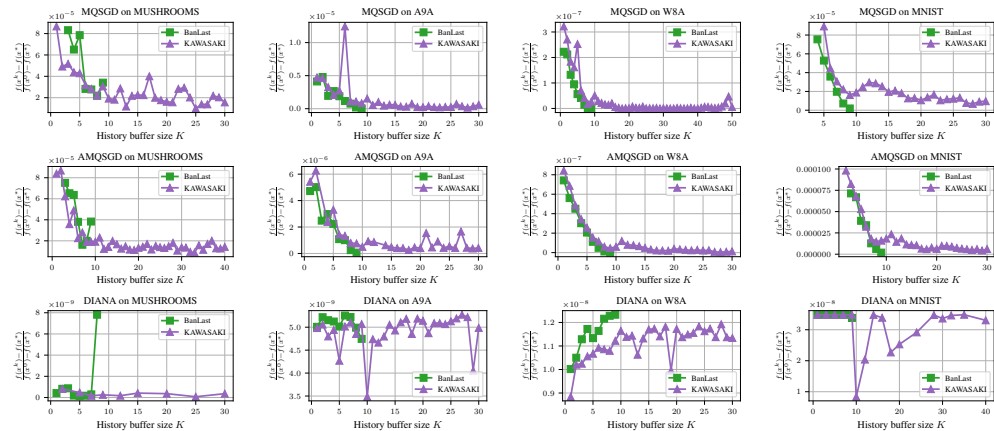

Figure 8: Convergence of Markovian-based algorithms on history size $K$

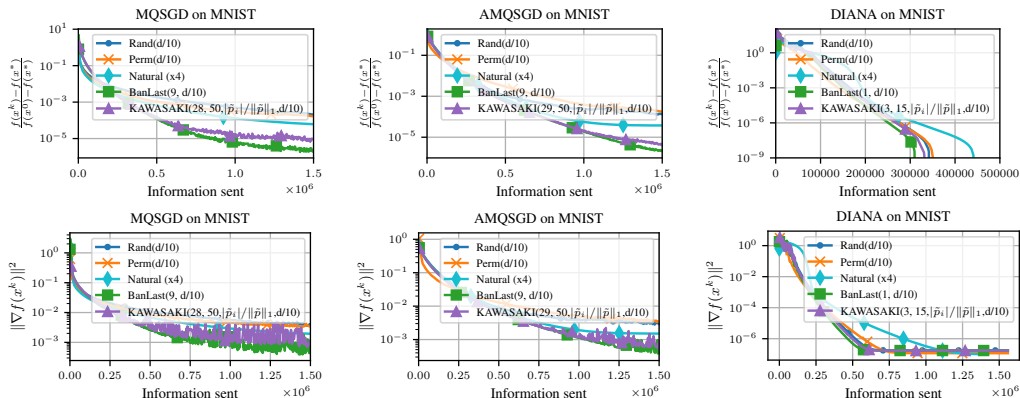

Figure 9: Comparison with PermK compressor and Natural compression. PermK compression factor is 10, Natural compression factor is 4. Logistic regression with L2 regularization on MNIST dataset for MQSGD, AMQSGD and DIANA algorithms on $N = 5$ clients. Best run is shown after fine-tuning learning rate, its decay, and Markovian compression parameters. X axis represent amount of information communicated.

Markovian compressors proposed in the paper compress vector coordinates dependently over optimization epochs. A similar idea of distributed compression is proposed in PermK Szlendak et al. (2021), where coordinates are arranged between workers at each iteration. Another compressor in the consideration is Natural compression Horvath et al. (2022), an unbiased randomized compressor.

Results of comparison of these compressors on MNIST dataset are presented in Figure 9. The results justify that Markovian compressors tend to converge faster than the competitors, allowing larger learning rates.

### H.5 COMBINATION WITH OTHER COMPRESSORS

Although markovian compressors are initially targeted to work with sparsification-based compressors, refining coordinates selection probabilities, they are fully compatible with other compressors afterwards. To illustrate this, and to conduct additional comparison with PermK compressor, we setup experiments combined with Natural Compression . Precisely, we compare RandK+Natural, PermK+Natural, BanLast+Natural and KAWASAKI+Natural compressors on logistic regression on MNIST dataset.

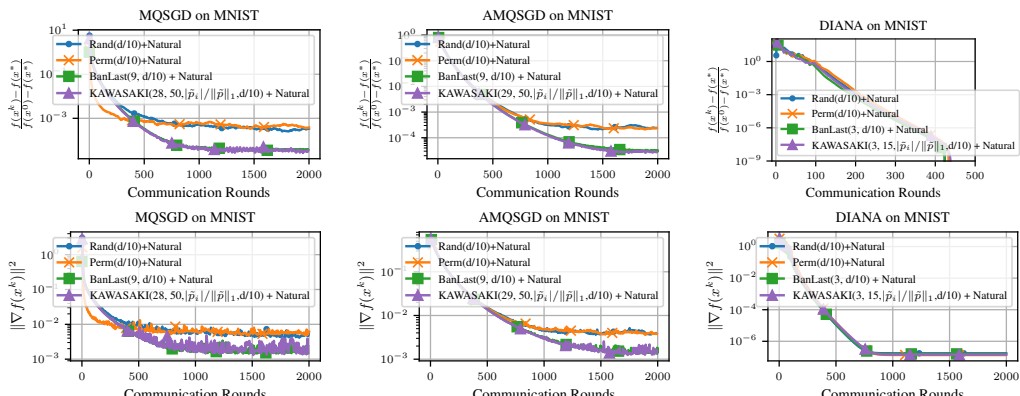

Figure 10: Experiments with Natural compression, MNIST logistic regression experiments. Best run after hyperparameters tuning is displayed for each method.

Figure 10 shows results of combination of mentioned sparsification compressors with natural compression.

## H.6 NEURAL NETWORKS EXPERIMENTS: DATA PARALLELISM CASE

To adopt Markovian compression to a more complex task, we perform image classification on CIFAR-10 Krizhevsky et al. (2009) with Resnet-18 He et al. (2016) convolutional neural network. We split the training set of size $50,000$ equally between $N = 5$ clients. We use SGD optimizer with momentum $0.9$ and weight decay $5 \cdot 10^{-4}$. Hyperparameters such as batch size and learning rate are fine-tuned. Markovian compresors hyperparameters, such as history size $K$ and forgetting rate $b$ are fine-tuned, while activation function is set to ordinary normalization. Experiments are conducted with several sparsification compressors, such as Rand-5%, Rand-7%, and Rand-10%, with number of epochs adjusted for each case.

Figures 11, 12 and 13 present train loss, gradient norm and test accuracy for each baseline method and Markovian compressors for Rand-5%, Rand-7% and Rand-10% scenarios, respectively. Summary on best test accuracy is presented in Table 3, and extended numerical results for Rand-5% compressor were presented in main experiments Table 1. We observe that in such complex, batched optimization problem only KAWASAKI obtains a substantial convergence improvement, as opposed to simpler logistic regression. Nevertheless, BanLast still performs the best when used with large history size, while both history size and forgetting rate are low for KAWASAKI. In terms of achieved test set accuracy, methods differ significantly only on higher compression rates like Rand-5%. This may imply that Markovian compression tolerates stronger compression, which is useful in practice. To summarize, Markovian compressors can be successfully applied in neural networks training, with KAWASAKI compressor significantly improving convergence.

Finally, we also conduct the comparison with Permutatino and Natural compression, both independently and in combination. Figure 14 shows learning curves for training with $N = 20$ clients. KAWASAKI compressor appears to have best convergence in both independently and in combination with Natural compression againt Permutation compressor.

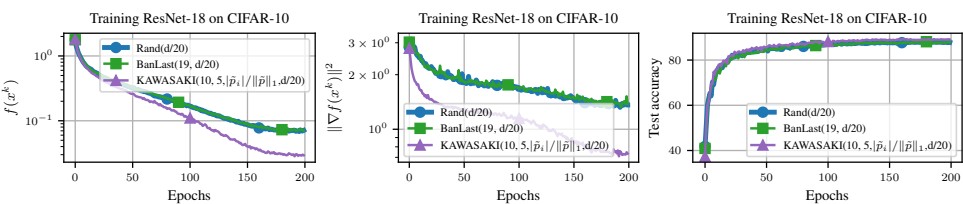

Figure 11: Resnet-18 on CIFAR-10 training results for Rand-5% sparsification.

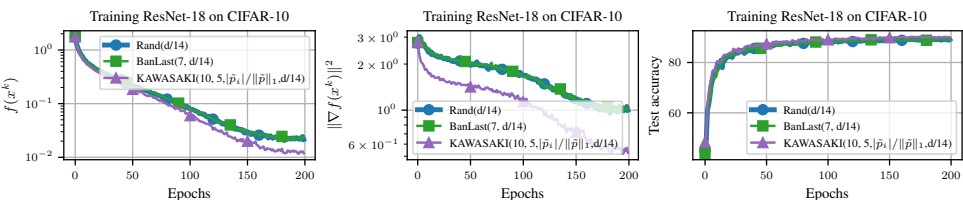

Figure 12: Resnet-18 on CIFAR-10 training results for Rand-7% sparsification.

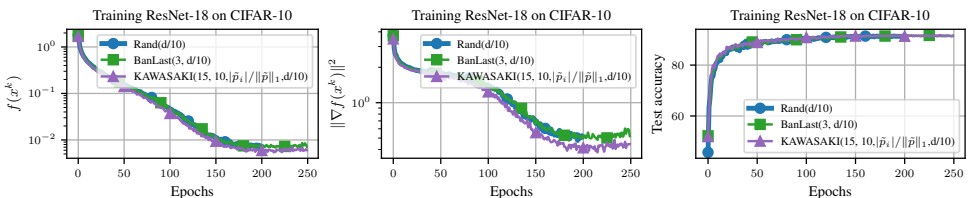

Figure 13: Resnet-18 on CIFAR-10 training results for Rand-10% sparsification.

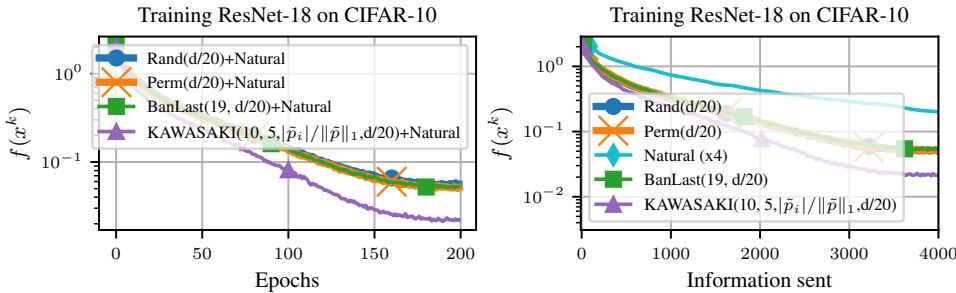

Figure 14: Comparison with other compressors on Resnet-18 training on CIFAR-10 dataset for Rand-5% sparsification on $N = 20$ clients. Natural compression factor is 4. Left figure is sequential combination with Natural compression. Right figure is comparison against PermK and Natural compressors independently, with information sent on x-axis.

Table 3: Best test accuracy % of training ResNet-18 on CIFAR-10 with different compressors

|  | Rand-K% | Banlast | KAWASAKI |
|---|---|---|---|
| Rand-5% | 88.03 | 88.1 | **89.27** |
| Rand-7% | 89.31 | 89.38 | **90.28** |
| Rand-10% | **91.46** | **91.72** | **91.78** |

## H.7 NEURAL NETWORKS EXPERIMENTS: MODEL PARALLELISM CASE

As opposed to data-parallel setting, model parallelism is paradigm which splits the model (typically a deep neural network) to a pipeline of layers between workers. Such distributed scenario is especially relevant for large language models (LLM), which consist of billions of trainable parameters. As communication is a typical bottleneck in such systems Diskin et al. (2021), various compression techniques are applied to layer activations and their respective gradients that are transferred between adjacent pipeline workers. Such techniques include quantization and sparsification Dettmers et al. (2022); Bian et al. (2023), as well as low-rank compression Song et al. (2023) techniques.

We perform training of Resnet-18 He et al. (2016) convolutional neural network on CIFAR-10 dataset Krizhevsky et al. (2009). We split the ResNet onto 4 workers by resnet blocks, simulated on a single device with compression of activations and their respective gradients in the places of communication. We apply Markovian compressors only to gradients in model-parallel setup, using

same RandK compression for both activations and gradients independently for each compression block.

Table 4: Best test accuracy % for model parallelism experiments with Resnet-18 classification of CIFAR-10

| Compressor | Compression ON | Compression OFF |
|---|---|---|
| No compression | 92.8 | 92.8 |
| Rand10% | 84.6 | 86.1 |
| BanLastK+Rand10% | 85.2 | 86.4 |
| KAWASAKI(simplex projection)+Rand10% | 84.5 | 85.0 |
| KAWASAKI(normalize)+Rand10% | 85.2 | **86.8** |
| KAWASAKI(softmax)+Rand10% | 85.3 | **87.3** |

Table 4 presents best test set accuracy achieved for training with different compressors. While compression indeed decreases accuracy for Rand-10%, application of Markov compressors, especially KAWASAKI with normalization and softmax activation functions, favours the final test accuracy on a whole one percent. Note that compression is not applied during inference, only on training phase. This case illustrates potential of Markov compressors beyond data-parallelism setup considered in theory. In practical training of large neural networks, where both data-parallelism and model-parallelism are often applied simultaneously, Markov compressors could also be useful, as per shown efficiency on both these setups in separate.

### H.8 FINE-TUNING DEBERTAV3-BASE ON GLUE DEVELOPMENT SET

In this series of experiments, we examine a distributed approach to fine-tuning language models using LoRA (Hu et al., 2021). This method is based on freezing the model weights that are pre-trained on a large dataset, and add a low rank adapter with matrices $A \in \mathbb{R}^{n \times r}$ and $B \in \mathbb{R}^{r \times m}$ to some selected layers $W_{\text{old}} \in \mathbb{R}^{n \times m}$ of this model, such that $W_{\text{new}} = W_{\text{old}} + A \cdot B$. Since in practice the parameter $r$ is chosen to be much smaller than $n$ and $m$, the new model has much fewer trainable parameters and can be efficiently trained on downsteram tasks.

In our experiments, we apply LoRA adapters with fixed rank $r = 8$ to the attention layers of the DeBERTaV3-base model (He et al., 2021). The downsteram task is the classical GLUE benchmark for natural language understanding (Wang et al., 2019). We consider only random sparsification compressors (Definition 4) with $25\%$ compression rate, due to the large computational cost of this experiment. Figure 15 shows learning curves for training with $N = 10$ clients. Our Markovian compressors appears to have best convergence against independent Rand$m$ compressor.

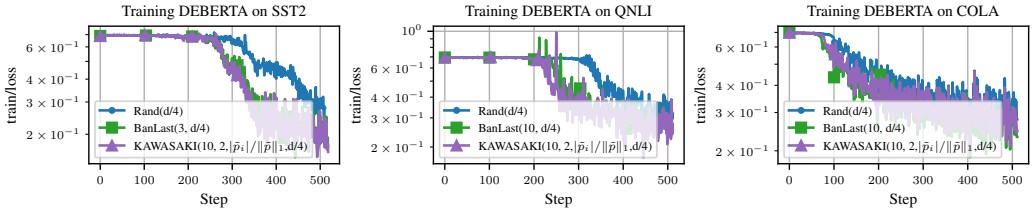

Figure 15: Comparison with other compressors on fine-tuning task on GLUE benchmark on $N = 10$ clients. We performed experiments on SST2, QNLI and COLA tasks, they are arranged from left to right.

