# OpenReview forum: "Markovian Compression: Looking to the Past Helps Accelerate the Future"
_ICLR.cc/2025/Conference — Submitted to ICLR 2025_

### Official Review · Reviewer_iwaF · 2024-10-24

**Soundness:** 3
**Presentation:** 3
**Contribution:** 2
**Rating:** 3
**Confidence:** 2

**Summary:**

This paper proposes a method for improving the performance of distributed optimization mitigating the communication bottleneck. The key idea is to use Markovian compressors in the vanilla Quantized Stochastic Gradient Descent (QSGD) algorithm. In addition, the momentum acceleration method is also used. The proof of convergence is provided and the usefulness of the proposed method is confirmed by experiments.

**Strengths:**

The algorithm is simple and looks easy to implement. Convergence of the algorithm is guaranteed by proofs under several assumptions. Practical usefulness is shown experimentally.

**Weaknesses:**

The key idea of "random sparsification" has already been proposed in the vanilla QSGD in an earlier paper. The contribution of this paper is to introduce Markovian randomness to it. The significance of the contribution is smaller than the prior work although the guarantee of its convergence may be non-trivial to prove.

**Questions:**

There are various hyperparameters. Although some details on their tuning are provided in Appendix H, please describe the guiding principles in more detail.

---

> ### Author Response · Authors · 2024-11-16
>
> Thank you for the review! Please find attached an improved version of the paper, which incorporates all comments received. The blue color of the text indicates changes made to the previous version.
>
> > __The key idea of "random sparsification" has already been proposed in the vanilla QSGD in an earlier paper.__
>
> This paper indeed proposes an improvement to the already existing concept of «random sparsification». However, as can be seen from experiments (“Practical usefulness is shown experimentally”), our modifications, which are quite easy to implement in practice, give strong gains to the performance of machine learning models. Also, as the reviewer mentioned, “the theoretical analysis is non-trivial”. It is worth noting that many papers published in well-known ML journals and ML conferences on the topic of distributed optimization also provide the results with close contribution:
>
>
> [1] Szlendak et al., Permutation Compressors for Provably Faster Distributed Nonconvex Optimization, __ICLR__
>
> In this paper authors consider random sparsification too, however they permute send coordinates across all workers, therefore they can accumulate true gradient on the server, if functions $f_i$ are «similar».
>
> [2] Beznosikov et al., On biased compression for distributed learning, __JMLR__
>
> Here authors uniformly analyze all existing compression methods (sparcifications and quantizations) and existing error feedback techniques.
>
> [3] Mishchenko et al., IntSGD: Adaptive floatless compression of stochastic gradients, __ICLR__
>
> IntSGD is a method that rounds all coordinates of a worker’s gradient to the integers, therefore it becomes more memory efficient in terms of computational efficiency. Thus, intSGD can be thought of as quantising the gradient vector to integers. This method already existed in the literature before the Mishchenko et al. article, in this paper the authors refined this method and presented its theoretical analysis.
>
> [4] Bernstein et al., signSGD: Compressed optimization for non-convex problems, __ICML__
>
> Authors in this paper take the sign function of the gradient at each step, which can be considered as quantization of the input vector.
>
> [5] Karimireddy et al., Error feedback fixes sign sgd and other gradient compression schemes, __ICML__
>
> This is a combination of existing sign SGD and existing error feedback techniques.
>
> [6] Horvath et al., Natural compression for distributed deep learning, __MCML__
>
> This paper provides a compression method, based on the rounding coordinates of the gradient to the power of two. This is a particular case of an existing quantization technique.
>
> > __There are various hyperparameters. Although some details on their tuning are provided in Appendix H, please describe the guiding principles in more detail.__
>
> Thank you for the fair remark! We have added further details regarding the experimental setup and hyperparameters into the updated version of the paper. Please refer to the amendments made to the Experiments section (lines 2305-2318), specifically Table 2 with hyperparameters values used for gridsearch.

---

> > ### Comment · Reviewer_iwaF · 2024-11-19
> >
> > Thank you for your response to my comments.
> > However, I still find it difficult to see sufficient value in this paper from the following perspectives:
> >
> > 1. The proposed method includes the naive QSGD as a special case. Therefore, it seems evident that the performance would improve over QSGD by simply broadening the range of parameter search.
> >
> > 2. The increase in parameters adds a cost to finding the optimal settings. According to your response, there seems to be no theoretical guidance for this search, and it appears that one has to rely on numerical exploration for each specific problem. The effectiveness of the proposed method should be demonstrated considering this cost as well. However, the paper lacks such discussion, making it difficult to determine whether this method is practically useful.

---

> ### Author Response · Authors · 2024-11-19
>
> Thanks for your comment! Indeed, in the previous version of the paper, in Example 1 we show how we can automatically choose the value $K$ for the particular $d/m = 0.1$. Now, based on Example 1, we expanded this result and derived the optimal value of $K$ for any fixed $d/m$ – see Appendix B. We find the full theoretical expression and, moreover, find the simplification (via approximation) of this expression: $K^* = 0.73 \cdot d / m$. It can be considered as an automatic selection for $K$ without tuning.  One can also look at Figure 7, in which we varied $K$ for $d/m = 10$. These plots show that the fixed value of $K^* = 7 \approx 0.73 \cdot 10$ is close to, and sometimes coincides with, the optimal value, which only confirms our theoretical results. We describe in more detail why this particular formula is optimal for choosing $K^*$ in Appendix B in the new version of the paper. One can see that this value $K^*$ is sufficient, and the additional tuning provides improvements, but not dramatic ones – see Figure 7. For completeness we do tuning, but it is optional for the general user.
>
> Moreover, experiments from Figure 7 show that careful selection of additional hyperparameters (e.g. $K$) is not required. It is enough to take $K \approx d/m$ (as mentioned above). Moreover, if we take $K$ even larger, it does not give any problems in terms of convergence.

---

> > ### Author Response · Authors · 2024-11-29
> >
> > Dear Reviewer iwaF,
> >
> > With the discussion deadline approaching, we are writing to kindly continue the discussion. We have added details on the selection of the new hyperparameters and reflected this in the previous comment and the new version of the paper. Also we have conducted additional experiments (see Section H.8).
> >
> > Given these improvements and clarifications, we would kindly request the reviewer to reconsider the score.

---

### Official Review · Reviewer_omNy · 2024-10-28

**Soundness:** 3
**Presentation:** 2
**Contribution:** 2
**Rating:** 5
**Confidence:** 3

**Summary:**

This paper proposed two sparsification compressors for distributed optimization problems, where the communications between different devices are first compressed before transmission. Almost all previous compressors do not consider what have been transmitted during the previous iterations. While the paper proposed compressors that consider previous iterations in a Markovian fashion. The paper further proves convergence bounds for the proposed compressors. Some experimental results and numerical examples are presented to show that the proposed compressors result in performance improvements compared with previous approaches.

**Strengths:**

The paper present convergence bounds for the proposed compressors and distributed optimization approaches. There bounds can be valuable for analyzing similar optimization approaches.

**Weaknesses:**

* The presentation of the proof part need to be improved. In its current form, the proof is difficult to follow. It is suggested that for each inequality, the author may provide a clue that which fact(s) has been used to get the inequality. It may also be helpful that the author provide more proof sketches.
* It seems that the convergence bounds hold for any unbiased compressors or asymptotic unbiased compressors. Then, how the author justify that BanLast and KAWASAKI are near-optimal among all Markovian type compressors.
* For the experiment part, in my opinion, more experimental results are needed for justifying that the proposed compressors truly outperform the previous random compressors. It is recommend that more experiments with different types of neural networks and more numbers of devices. The distributed optimization approaches are interesting, typically when people want to train large models on massive datasets. So it would be interesting to see that the proposed compressors outperform other approaches on larger training jobs.

**Questions:**

* In line 850, the notation is quite confusing for me. The left-hand side is a matrix and the right hand side sounds like a norm for me. Are there any typos here?

---

> ### Author Response · Authors · 2024-11-16
>
> We are grateful to the reviewer for providing comprehensive feedback on our paper. Please find attached a revised version of the article, in which we tried to address all of the suggestions provided in a comprehensive manner. All corrections are marked in the blue color.
>
> > __The presentation of the proof part need to be improved.__
>
> Thank you for the fair remark. In an effort to provide insight into the proofs, additional clues have been added into Appendix. For example, please direct your attention to lines 1166, 1227, 1346, 1446, 1634, 1651, and 1753 of the revised version of the paper.
>
> > __It seems that the convergence bounds hold for any unbiased compressors or asymptotic unbiased compressors.__
>
> It is true that the convergence results presented in the paper are applicable to any asymptotic unbiased compressor. Nevertheless, it would appear that there has been a certain degree of miscommunication. We did not claim in our paper that BanLastK and KAWASAKI are the optimal ones. We provided them as an illustrative example of Markovian type compressors. The optimality of compressors in any class is an extremely complex issue that requires a separate, in-depth discussion.
>
> The question of optimality of compression is open even for unbiased and independent compressions. In [1], the authors try to make an attempt to construct such a theory using the uncertainty principle. But the work is far from an exhaustive analysis. This is due to the fact the uncertainty principle is not the only and not the most obvious way to describe optimality. Moreover, no current compression (including the one proposed by the authors) is optimal - see Fig.1 of [1]. And the algorithm proposed by the authors is costly and computationally complex because it uses a greedy procedure inside, but this is not taken into account in terms of time losses compared to simpler compression approaches. An attempt was also made to study the lower bounds and optimality in the convergence theory of distributed optimization with compression [2], but in obtaining these estimates the authors limited themselves to independent random selection of coordinates. The question of considering a general class of unbiased operators remained uncovered.
> Summarizing, the advanced results of the community on optimality of compression operators are far from the final ones.
>
>
> [1] Safaryan et al., Uncertainty principle for communication compression in distributed and federated learning and the search for an optimal compressor, __Information and Inference: A Journal of the IMA__
>
> [2] He et al., Lower bounds and accelerated algorithms in distributed stochastic optimization with communication compression, __arxiv__
>
> > __For the experiment part__
>
> We added NLP experiments - see Appendix H.8. In these experiments, our approach performs just as well.
>
> > __In line 850, the notation is quite confusing for me.__
>
> The right-hand side of this expression is just a notation of the elements of the matrix $P$: $p_{ij}$, $i$ and $j$ are from 1 to $n$. In the new version of the paper we changed the notation $|| \cdot ||$ to ( ) to avoid potential misunderstandings (see line 850).

---

> > ### Author Response · Authors · 2024-11-29
> >
> > Dear Reviewer omNy,
> >
> > With the discussion deadline approaching, we are writing to kindly request your feedback on our rebuttal response. We have  made the proof more detailed, conducted additional experiments.
> >
> > Given these improvements and clarifications, we would kindly request the reviewer to reconsider the score.

---

### Official Review · Reviewer_oiFi · 2024-10-29

**Soundness:** 3
**Presentation:** 2
**Contribution:** 3
**Rating:** 5
**Confidence:** 2

**Summary:**

The paper consider a generic distributed optimisation problem, and proposes a solution to the information bottleneck due to the necessary communication between computing nodes working in parallel. The solution is based on two stochastic compression operators that go beyond simple sparsifiers. Theoretical guarantees are provided on convergences rates in terms of distance to the minimum of the objective. The effect of momentum is also analysed. Numerical tests are then presented on distributed logistic regression and neural networks training.

**Strengths:**

Let me start by stating that I am not *at all* an expert in optimisation, even less in distributed setting. I will therefore not judge the novelty of the paper as I have no clue.
Given this, I have found the paper interesting. The proposed idea is very natural and quite simple. The theoretical guarantees are strong and provide an clear view on convergence properties (I think).
The numerics are convincing.

**Weaknesses:**

I was disappointed by the quality of presentation. E.g., the text is sometimes quite cryptic and there are many linking words missing. Some definitions are missing too, and I had to guess a number of times what the authors meant. I recommend making an effort in presentation.
In spite I have found the problem in interesting in itself, it is not clear for me that ICLR is the right place to publish this paper. But the AC and authors probably know more about this than me, this is not a strong view.

**Questions:**

+ strange referencing in the text: I don't know if this is the standard ICLR referencing, but a lot of text space is lost due to the refs everywhere, that cut also the reading flow.
+ def 1: L-smooth -> L_i-smooth
+ Randm never defined
+ F_tau not defined in Theorem 2. Is this just a constant? That depends on what?
+ what is "variance reduction" discussed in a whole paragraph?
+ what is the « information sent » axis in fig 1?
+ many linking words missing

---

> ### Author Response · Authors · 2024-11-16
>
> We thank the reviewer for the comprehensive comments towards our paper. Please find attached a revised version of the paper, in which we tried to address all of the feedback provided. Some sections of the text have been revised to enhance clarity and coherence. We also added linking words, making the proof clearer. All corrections are marked in the blue color.
>
> > __Strange referencing in the text:__
>
> We would like to point out that the standard .sty file of the ICLR 2025 conference was used. Therefore, all references are formatted in the correct style.
>
> > __def 1: L-smooth -> L_i-smooth__
>
> Thank you for identifying the typo. We fixed it in line 153 of the updated version of the paper.
>
> > __Randm never defined__
>
> One of the most important definitions in our paper is Definition 4 (lines 193-194). Subsequently, in lines 195-197, we claim that ' the classical random sparsification operator fits Definition 4'. However, as you correctly observed, we did not designate it precisely as Rand$m$, although it is identical. To prevent further misunderstandings, we have clarified this in the revised version of the paper (see line 196).
>
> > __F_tau not defined in Theorem 2__
>
> In the initial version of the paper, we stated at the end of Theorem 2 (line 285) that we employ the notation $F_t := E[f(x^t) - f(x^*)]$. In other words, this signifies that $F_\tau := E[f(x^\tau) - f(x^*)]$. Nevertheless, to prevent any potential confusion, we have included a separate definition for $F_\tau$ (see line 285) in the updated version of the paper.
>
> > __What is "variance reduction" discussed in a whole paragraph?__
>
> The “variance reduction” represents a well-established technique in finite-sum minimization problems of the form $\min_{\mathbf{x}} \frac{1}{N}\sum\limits_{i = 1}^{N} f_i(\mathbf{x}) =: f(\mathbf{x})$. Instead of using a random sample $g_k = \nabla f_i(\mathbf{x}_k)$ as a gradient estimator, variance reduction methods use $g_k = \nabla f(\mathbf{w}_k) + \nabla f_i(\mathbf{x}_k) - \nabla f_i(\mathbf{w}_k)$. A proper choice of $\mathbf{w}_k$ decreases the “variance” $\mathbb{E}\|\|g_k - \nabla f(\mathbf{x}_k)\|\|^2$ compared to $\mathbb{E}\|\|\nabla f_i(\mathbf{x}_k) - \nabla f(\mathbf{x}_k)\|\|^2$. This approach yields enhanced stability and accelerated convergence. In the context of compression operators, this technique has been employed to develop error compensation methods [1] that compress the error rather than the pure gradient, as the name implies. For an introduction to these methods, we would suggest the articles [2, 3, 4, 5] we cited at the beginning of the discussion about this technique at line 441.
>
> [1] Frank Seide, Hao Fu, Jasha Droppo, Gang Li, and Dong Yu. 1-bit stochastic gradient descent and its application to data-parallel distributed training of speech DNNs, Annual Conference of the International Speech Communication Association
>
> [2] Mishchenko et al., Distributed learning with compressed gradient differences, Optimization Methods and Software
>
> [3] Gorbunov et al., MARINA: Faster non-convex distributed learning with compression, International Conference on Machine Learning
>
> [4] Tyurin and Richtarik, DASHA: Distributed Nonconvex Optimization with Communication Compression and Optimal Oracle Complexity, International Conference on Learning Representations
>
> [5] Richtarik et al., EF21: A New, Simpler, Theoretically Better, and Practically Faster Error Feedback, Advances in Neural Information Processing Systems
>
>
> > __What is the « information sent » axis in fig 1?__
>
> In all of our experiments, we do not utilize the steps of the optimizer, but rather the information that is transmitted by each worker at the current timestamp $t$. This implies that there are $n$ workers, with each worker sending $m$ coordinates at each iteration of the optimization step. Consequently, the $x$-axis displays numbers of the form $m n \cdot 1, mn \cdot 2, \dots, m n \cdot t, \dots , m n \cdot T$. This allows us to understand the performance of compressors with varying values of $m$ and $n$. We have incorporated this description into the revised version of the paper, which can be found on lines 2302-2306.
>
> > __Many linking words missing__
>
> The latest iteration of the paper incorporates a greater number of linking words. For further examples, please see lines 31, 35, 38, 40, 44, 93, 103 and 106.

---

> ### Author Response · Authors · 2024-11-29
>
> Dear Reviewer oiFi,
>
> With the discussion deadline approaching, we are writing to kindly request your feedback on our rebuttal response. We have made the work with the text (linking words, typos), added more explanations. Also we have conducted additional experiments (see Section H.8).
>
> Given these improvements and clarifications, we would kindly request the reviewer to reconsider the score.

---

### Official Review · Reviewer_B4dG · 2024-11-04

**Soundness:** 3
**Presentation:** 4
**Contribution:** 3
**Rating:** 8
**Confidence:** 3

**Summary:**

This paper proposed a possible improvement of the random sparsification method in the context of distributed optimization, which is in turn a compression technique aimed at reducing the communication cost. Briefly speaking, random sparsification compresses the data by selecting a subset of coordinates uniformly at random and keeping only these selected coordinates of data. The key idea of the paper is that, instead of using i.i.d. selection of subset, it may help to select the subset with a Markov chain. In this way, more diversity is encouraged by setting a smaller probability to select those coordinates which was just selected in the previous few iterations.

**Strengths:**

I consider Markovian compressors to be a novel idea, but it is possible that I have missed some related literature.

The proposed algorithm is easy to implement and fairly understandable, which might broaden its interest to general readers.

The math looks correct to me, though I have not checked every detail.

**Weaknesses:**

My main concern is that the theoretical analysis does not seem to touch the real advantage of using a Markovian compressor. It appears that the paper resorted to stationarity analysis, but the stationary distribution of the Markov chain simply falls back to the standard random sparsification. It is thus unclear what is gained by a Markovian compressor. Although a better analysis could be challenging, as argued in the paper, the significance of the proposed algorithm would be greatly strengthened if more theoretical insights were provided.

**Questions:**

Please refer to the Weaknesses part.

---

> ### Author Response · Authors · 2024-11-16
>
> We thank the reviewer for the great feedback! We attached an updated version of the paper, in which we have tried to take into account the comments of every reviewer. All alterations to the text are indicated by blue highlighting.
> > __My main concern is that the theoretical analysis does not seem to touch the real advantage of using a Markovian compressor.__
>
> Thank you for the fair remark! We did not attempt to disguise the issue that had been raised, and we tried to discuss it in Section 2.4, lines 420-430, of the original paper. It is indeed the case that, from a theoretical standpoint, it is optimal to take a relatively small value of $\tau$, as this will result in a more rapid convergence of the chain towards a uniform distribution. Nevertheless, as the idea of the paper and empirical evidence indicate, incorporating some history (i.e., utilizing $\tau > 1$) yields superior results compared to the conventional RandM compressor (that is, not incorporating any past information).

---

> > ### Author Response · Authors · 2024-11-29
> >
> > Dear Reviewer B4dG,
> >
> > With the discussion deadline approaching, we are writing to kindly request your feedback on our rebuttal response. Note that we have improved the paper according to the rest of the reviews.

---

### Meta-Review · Area_Chair_bRky · 2024-12-21

**Metareview:**

This paper considered the distributed optimization problems proposed a family of compression schemes where operators transform vectors fed to their input according to a Markov chain. The authors also analyzed the method theoretically by considering non-convex, Polyak-Lojasiewicz, and strongly convex settings. Moreover,  the efficiency of the method was verified through experiments on distributed optimization tasks.

Strengths:

1. Integrating Markovian processes into compressors for distributed optimization is an interesting idea.

2. The authors provided some  convergence analysis.

Weaknesses:

1. Given existing methods like QSGD, the main contribution is an incremental improvement, leading to marginal novelty.

2. Some concerns were raised regarding the intense hyper-parameter finetuning.

3. The experiments are not sufficient and,  to fully validate the approach,  it was suggested to perform some larger-scale training tasks and more various architectures by the reviewers.

4. The writing of this paper is not good enough.

Overall, I have to recommend rejection but I sincerely encourage the authors to improve it.

**Additional Comments On Reviewer Discussion:**

In the rebuttal, some concerns were addressed but some  not, for example the incremental novelty and insufficient experiments.  Moreover, Reviewer also concerned about the hyper-parameter tuning; omNy requested additional experiments with other neural network types and larger-scale tasks.

---

### Decision · Program_Chairs · 2025-01-22

Reject